# Transparent artificial intelligence-enabled interpretable and interactive sleep apnea assessment across flexible monitoring scenarios

Shuaicong Hu [1], Jian Liu [1], Yanan Wang [1], Cong Fu[2,3,4], Jichu Zhu[2], Huan Yu[2,3,4] ✉ & Cuiwei Yang [1,5] ✉

Early detection of widespread undiagnosed sleep apnea is crucial for preventing its severe health complications. However, large-scale diagnosis faces inaccessible monitoring and trust barriers in automated analysis, particularly due to the absence of transparent artificial intelligence frameworks capable of monitoring adaptation. Here, we develop Apnea Interact Xplainer, a transparent system enabling sleep apnea diagnosis through flexible channel analysis across clinical and home settings. Analyzing 15,807 polysomnography recordings from seven independent multi-ethnic cohorts, our system achieves accuracies of 0.738-0.810 for four-level severity classification, with 99.8% accuracy within one severity grade and R-squared of 0.92-0.96 for apnea-hypopnea index prediction on external test cohorts. The system provides multi-level expert-logic interpretable visualization of respiratory patterns enabling transparent collaborative decision-making. Notably, it achieves a sensitivity of 0.970 for early sleep apnea detection using only oximetry signals, while providing nightly risk assessment and intelligent monitoring reports. This study establishes a paradigm shift in advancing early and cost-effective sleep apnea diagnosis through transparent artificial intelligence.

At least 936 million (95% CI: 903–970) adults aged 30–69 globally suffer from mild to severe sleep apnea (SA)[1], a common sleep disorder characterized by recurrent breathing interruptions during sleep. These repetitive episodes of hypoxia-reoxygenation trigger oxidative stress, sympathetic activation, and systemic inflammation[2], leading to serious health complications including neurocognitive impairment[3], cardiovascular diseases[4], and metabolic disorders[5]. Early diagnosis and intervention are crucial as these pathophysiological changes can be reversed in early stages[6]. Despite advanced healthcare systems in developed countries like the United States, an estimated 75–90% of SA patients remain undiagnosed, incurring annual economic costs exceeding $150 billion[7]. The complex and expensive nature of sleep monitoring (approximately $1500 per test[8]) and time-consuming manual analysis (average 70.7 min per recording[9]) has led to widespread underdiagnosis, particularly in developing countries where healthcare resources are limited[10]. This challenge is further intensified by the expanding patient demographics beyond traditionally affected aging populations to include younger adults due to modern lifestyle changes[11].

[1]Department of Biomedical Engineering, College of Biomedical Engineering, Fudan University, Shanghai, China. [2]Sleep and Wake Disorders Center of Fudan University, Shanghai, China. [3]Department of Neurology, Huashan Hospital, Fudan University, Shanghai, China. [4]National Center for Neurological Disorders, Huashan Hospital, Fudan University, Shanghai, China. [5]Key Laboratory of Medical Imaging Computing and Computer Assisted Intervention of Shanghai, Shanghai, China. ✉e-mail: yuhuan@fudan.edu.cn; yangcw@fudan.edu.cn

Polysomnography (PSG), the gold standard for SA diagnosis, requires comprehensive overnight monitoring in specialized sleep laboratories with multiple physiological parameters and trained technicians. While PSG provides detailed sleep analysis, its limitations in specialized facility requirements, complex setup procedures, and potential sleep alterations due to the monitoring environment have restricted its widespread application[12]. To balance the cost and accessibility of sleep monitoring, home sleep apnea testing (HSAT)[13] and portable devices[14] have gained attention as alternative methods for diagnosing SA. These approaches primarily collect channel-specific information related to sleep breathing, improving the cost-effectiveness of SA diagnosis and helping to expand screening coverage, thereby increasing early detection rates of SA. Moreover, studies have shown encouraging results using single-channel signals, such as electrocardiogram (ECG)[15,16] and blood oxygen saturation ($SpO_2$), as alternative methods for diagnosing SA, which can be easily achieved with simple sensors[17,18]. Although these simplified monitoring approaches show promise, several fundamental challenges persist: labor-intensive data analysis, lack of standardized interpretation, and limited ability to capture comprehensive sleep events.

Given the rapid development of artificial intelligence (AI)[19–21] algorithms in time-consuming tasks such as sleep analysis[22–24], recent studies have explored AI-based approaches for SA diagnosis. However, existing methods have several critical limitations: (1) Most ECG-based studies have only performed binary event classification without apnea-hypopnea index (AHI) estimation and have been limited by small training datasets (<100 subjects)[15]; (2) Although some work has demonstrated the value of configurable channel combinations[18,25], their black-box decision processes hinder clinical adoption; (3) While large-scale oximetry analysis has achieved promising AHI estimation[17], it lacks detailed respiratory event traceability for clinical review. Moreover, none of these approaches has provided deployable AI tools for real-world practice. Developing such tools for variable monitoring environments faces major challenges in maintaining stability across different populations, devices, and available channel configurations. Furthermore, emerging AI healthcare technologies require evidence of model decision-making logic[26,27]. This trust deficit is particularly problematic in sleep medicine, where clinical decisions rely on complex pattern recognition and interpretation of multiple physiological signals. The inability to understand AI's reasoning process not only limits clinicians' confidence in automated diagnoses but also prevents effective human-AI collaboration in identifying subtle or atypical sleep disorders. Traditional black-box AI approaches fail to leverage clinicians' expertise and experience, which are crucial for handling edge cases and ensuring patient safety. These limitations highlight the critical need for a transparent AI solution that combines interpretable decision-making with efficient expert review capabilities[28].

In this study, we develop Apnea Interact Xplainer (AIX), a transparent interactive framework featuring AI-based decision making, flexible sensor configuration, and efficient human-AI collaboration through event-level logic tracebacks (Fig. 1). The AIX system emphasizes explaining the reasoning behind the model's predictions of abnormal respiratory events through a transparent scale diffusion mechanism that uniquely enables multi-level interpretation of the AI decision process, from individual respiratory events to overnight diagnostic conclusions. We analyzed 15,807 PSG recordings from seven independent multi-ethnic cohorts to validate AIX's performance across diverse demographics, including ethnicity, gender, age, and BMI. AIX processes three key signals, including nasal airflow (Flow), respiratory effort from thoracic movement (Chest), and $SpO_2$, in clinical settings, while enabling reliable screening through $SpO_2$-only measurements in home-based settings, significantly improving accessibility. By enabling flexible monitoring scenarios and transparent AI-guided decision support, AIX offers a scalable solution that could potentially transform SA diagnosis from a specialized laboratory procedure to an accessible population screening tool, particularly beneficial for improving early detection rates across both clinical and home-based settings, while maintaining the clinical rigor necessary for reliable diagnosis.

## Results
### Overview of experimental procedures
This work uses seven heterogeneous cohorts across ethnic groups (Asian, Black or African American, Hispanic, White, and Others), including five from the National Sleep Research Resource (NSRR): Sleep Heart Health Study (SHHS) 1 and 2 (longitudinal studies)[29], Multi-Ethnic Study of Atherosclerosis (MESA)[30], Osteoporotic Fractures in Men Study (MROS)[31], and Cleveland Family Study (CFS)[32]. The remaining two cohorts are from FDU-HSH Sleep Research, comprising a retrospective cohort ($n = 350$) and a prospective cohort ($n = 297$), used to evaluate the AIX system's performance with portable monitoring configurations using real-world oximetry signals. These cohorts include 15,807 PSG recordings totaling over 130,000 h of overnight digital signals from multiple physiological sensors. We focus on channels common in HSAT[33] and portable devices: Flow, Chest, and $SpO_2$ signals. All signals are resampled to a uniform rate. Recordings are excluded if they lack required channels, have sleep durations under 4 h, or miss AHI annotations. Figures 2 and 3 summarize the distribution of gender, age, BMI, AHI, and recording duration across cohorts.

AHI annotation follows the American Academy of Sleep Medicine (AASM) standards[34] using *ahi_aOh3a* variable. Apnea is defined as Flow reduction ≥90%, hypopnea as Flow reduction ≥30%, both requiring ≥10 s duration and ≥3% oxygen desaturation. We combine apnea (obstructive, central, mixed) and hypopnea timestamps as respiratory event labels, as both contribute equally to AHI. Segments without these labels are normal, and $SpO_2$ artifact segments are excluded. AASM guidelines[34] classify subjects by AHI as healthy (<5 events/hour), mild (≥5 to <15 events/hour), moderate (≥15 to <30 events/hour), and severe (≥30 events/hour).

To ensure unbiased evaluation and assess temporal stability, we employ two distinct training strategies for the SHHS cohort: (1) Non-overlap SHHS model (training subjects: $n = 2789$, Supplementary Fig. 1), which excludes overlapping subjects between SHHS1 and SHHS2 for training and evaluation, ensuring no data leakage across cohorts. (2) All-Sub SHHS model (training subjects: $n = 5255$, Fig. 2), trained on all available SHHS1 subjects, maximizing the use of training data for performance optimization. The Non-overlap SHHS model is primarily used for SHHS2 evaluation to avoid bias and can also be applied to assess temporal performance between SHHS1 and SHHS2. Meanwhile, the All-Sub SHHS model serves as a reference for external test cohorts. Detailed results from both models are reported to provide a comprehensive view of AIX performance across different cohorts and channel configurations.

### AIX for AHI prediction and transparent granular traceability
We develop the AIX system with dual-model architecture supporting flexible monitoring configurations (Supplementary Fig. 11). Following clinical sleep staging conventions[34], the granularity prediction model (Model 1) makes event-level decisions by classifying each 30-s window, while incorporating 120-s preceding and following signals to form a 270-s contextual segment. Since severe respiratory events typically persist for 30–45 s[17] and require temporal context for accurate detection, this design ensures that while the classification target remains the central 30-s window, the model has sufficient context to detect respiratory events that may influence the target window. The model provides granular explanations through attention visualization within each 30-s window, highlighting specific regions of abnormal breathing patterns. As demonstrated in Supplementary Fig. 24a, the attention heatmaps directly highlight regions of interest in both Flow

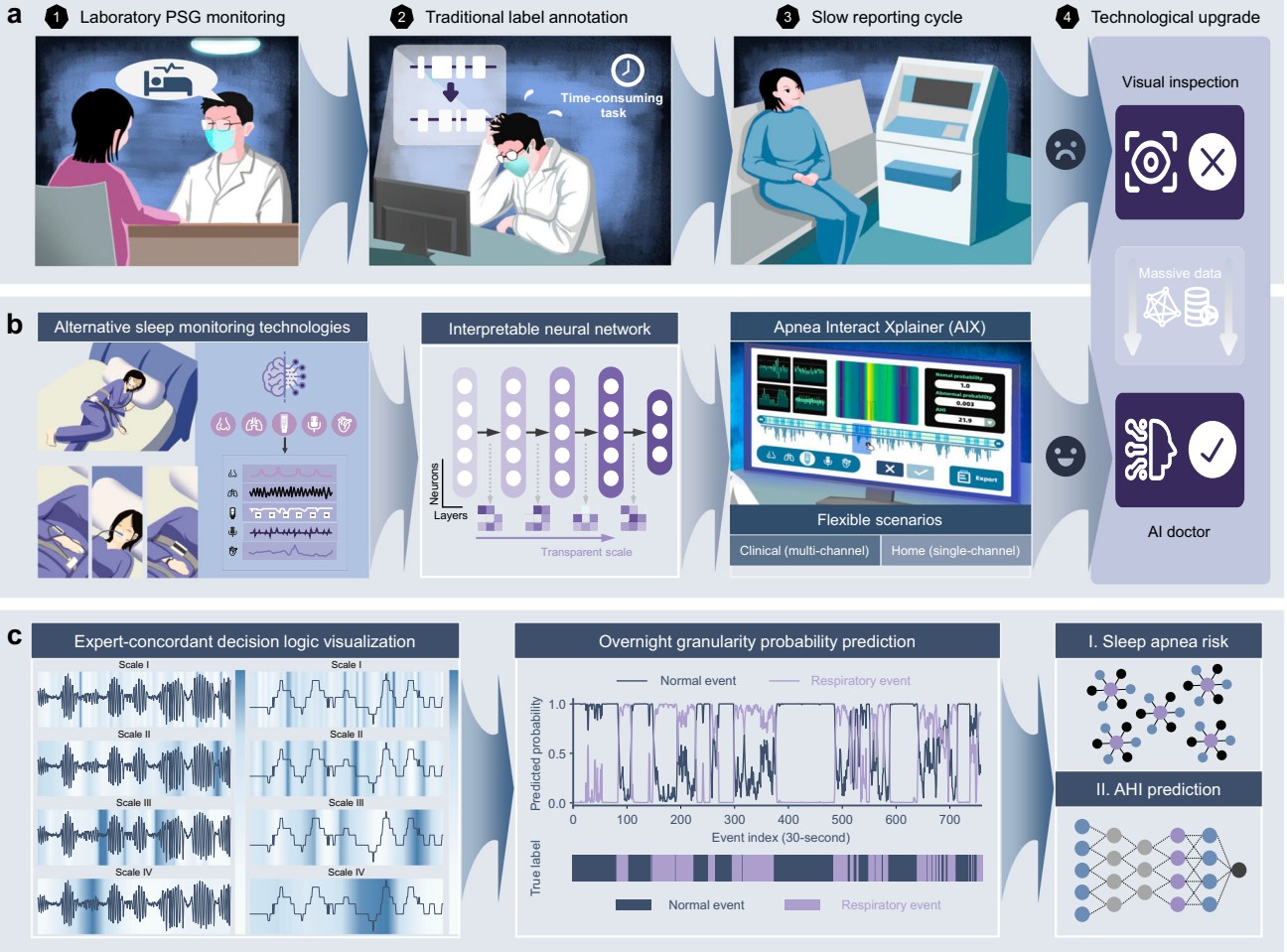

**Fig. 1 | The overall workflow of the study.** Our Apnea Interact Xplainer (AIX) system introduces transparent artificial intelligence (AI) technology that achieves expert-concordant decision visualization, enables seamless human-machine interaction, and supports flexible monitoring configurations across different clinical scenarios. **a** Traditional polysomnography (PSG) monitoring involves complex sensor setups, imposing a burden on subjects and relying on manual annotations by clinicians, which reduces diagnostic efficiency and limits screening rates. **b** The AIX system's flexible monitoring configurations adapt to both simplified multi-channel analysis in home sleep apnea testing (HSAT) and portable device-based home sleep monitoring as alternative approaches to traditional PSG. These diverse inputs are processed through an interpretable neural network model and a user-friendly interface design, facilitating transparent AI-guided human-machine interaction during diagnosis review. **c** The analytical functions of the AIX system demonstrate expert-concordant decision logic through a transparent scale diffusion mechanism, along with granularity prediction of overnight respiratory abnormality probability, overnight SA risk assessment, and AHI regression modeling.

and $SpO_2$ signals that correspond to expert-annotated respiratory events, providing traceable event-level interpretation of the model's decision process.

For single-channel scenarios (Flow, Chest, or $SpO_2$), Model 1 processes input signals of shape $[B,1,256]$ through our transparent scale diffusion network (TSD-Net) to enable expert-logic traceability through a transparent scale diffusion mechanism that visualizes multi-level attention patterns for AI decision interpretation, while outputting probability values for respiratory event detection. These 30-s probability values are concatenated to form the overnight probability sequence. For multi-channel configurations, Model 1 deploys parallel TSD-Nets for each channel, with each network serving as a feature extractor. Through average pooling along the channel dimension and flattening the features from the output of stage 4 (shape $[B,512,8]$, Supplementary Table 12), each TSD-Net outputs 8 abstract features (shape $[B,8]$). These features are concatenated to form a 24-dimensional feature vector (8 features×3 channels), which is then processed by LightGBM[35] to produce the overnight probability sequence.

The AHI regression model (Model 2) processes these overnight probability sequences for comprehensive analysis, whether from single-channel direct output or multi-channel integration. This model generates dual outputs: AHI prediction for clinical severity classification and sleep apnea risk index (SARI) calculation as an interpretable risk metric. SARI complements AHI by considering both event frequency and duration characteristics within the probability estimates.

This adaptive architecture enables AIX to maintain consistent analysis capabilities across different monitoring configurations while providing expert-concordant decision interpretation at the granular event level, supporting both accurate diagnosis and interpretable assessment.

## Flexible channel configuration performance in AHI regression and SA severity classification

The Non-overlap SHHS model is trained on a subset of SHHS1 subjects ($n = 2789$) without overlap with SHHS2 subjects, and tested on the remaining SHHS1 recordings and all SHHS2 recordings. In contrast, the

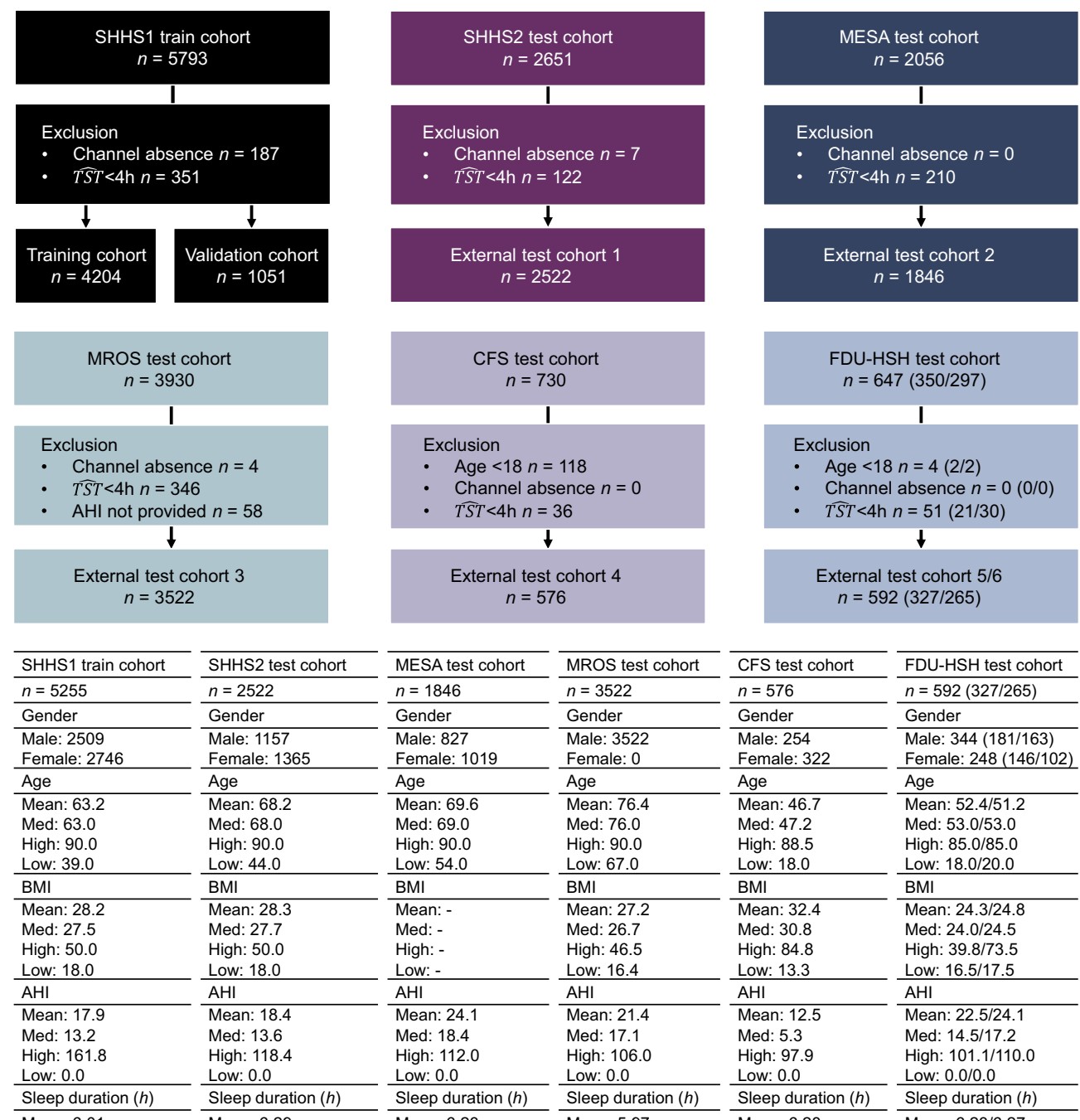

**Fig. 2 | Clinical characteristics of multicenter NSRR cohorts and real-world cohorts.** SHHS, Sleep Heart Health Study. MESA, Multi-Ethnic Study of Atherosclerosis. MROS, Men Study of Osteoporotic Fractures. CFS, Cleveland Family Study. FDU-HSH (retrospective/prospective), Sleep Research at Fudan University-Huashan Hospital.

All-Sub SHHS model is trained on all SHHS1 subjects ($n = 5255$) and evaluated on external cohorts (MESA, MROS, and CFS). To evaluate the system's performance across flexible monitoring scenarios, we examine the three-channel integration (Gold) results as a clinical reference standard. For AHI regression, the Gold configuration demonstrates excellent performance with R-squared ($R^2$) and intra-class correlation coefficient (ICC) values consistently above 0.90 across all test cohorts: SHHS1 ($R^2 = 0.94$, ICC = 0.94), SHHS2 ($R^2 = 0.96$, ICC = 0.96), MESA ($R^2 = 0.92$, ICC = 0.91), MROS ($R^2 = 0.96$, ICC = 0.96), and CFS ($R^2 = 0.93$, ICC = 0.93). More importantly for clinical application, we thoroughly evaluate the system's performance in four-level SA severity classification (normal, mild, moderate, severe). The Gold

configuration achieves robust classification accuracy across cohorts: SHHS1 (ACC = 0.78, Macro F1 = 0.78), SHHS2 (ACC = 0.81, Macro F1 = 0.82), MESA (ACC = 0.74, Macro F1 = 0.74), MROS (ACC = 0.81, Macro F1 = 0.80), and CFS (ACC = 0.80, Macro F1 = 0.79). Analysis of confusion matrices (Fig. 4) reveals that classification errors predominantly occur between adjacent severity categories, which is clinically acceptable given the continuous nature of AHI measurements. This pattern aligns with clinical practice, where misclassifications primarily occur for cases with AHI values near the categorical boundaries, having a limited impact on clinical decision making. The detailed performance comparison across different cohorts is provided in Supplementary Fig. 4 and Supplementary Tables 2, 3.

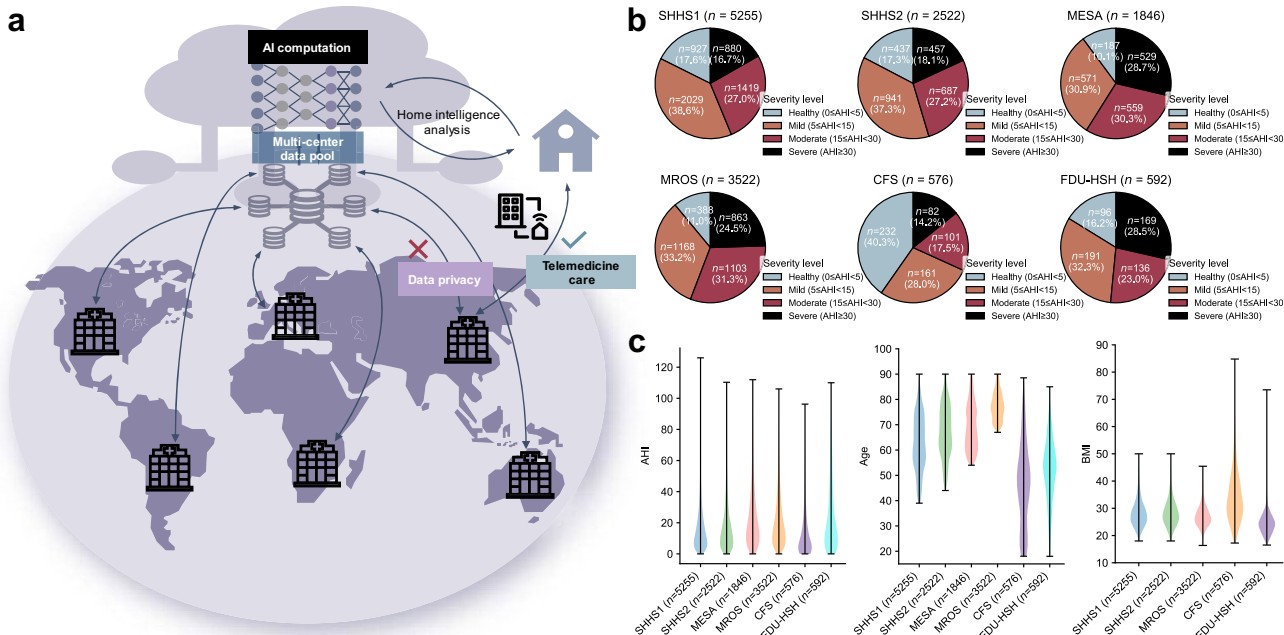

**Fig. 3 | Distribution of ethnic characteristics across regions. a** AI models are trained using diverse ethnic populations from multiple regions. This process aims to develop AI computational services with high generalization capabilities, offering pathways for the implementation of intelligent remote healthcare services and home self-monitoring management. **b** The composition of SA disease severity is analyzed using data from different centers. **c** AHI, age, and BMI are visualized using kernel density estimation (KDE) in the form of violin plots. Violin plots display the probability density of the data at different values, with the width of each violin representing the frequency distribution. Thin lines extend to the minimum and maximum values of each cohort. For AHI and age, the lower portion of the data was truncated during preprocessing (AHI ≥ 0, age ≥ 18).

To validate the system's adaptability to simplified monitoring configurations, we further evaluate the performance of individual channels (Supplementary Figs. 5–7). Notably, the $SpO_2$-only configuration maintains strong severity classification capability with only modest degradation from the Gold standard: SHHS1 (ACC = 0.79, Macro F1 = 0.79), SHHS2 (ACC = 0.78, Macro F1 = 0.78), MESA (ACC = 0.75, Macro F1 = 0.74), MROS (ACC = 0.79, Macro F1 = 0.77), and CFS (ACC = 0.77, Macro F1 = 0.78). The preservation of accurate severity classification, even with single-channel monitoring, is particularly significant for expanding screening accessibility. In contrast, the Flow-only configuration shows moderate performance (SHHS1: ACC = 0.62; SHHS2: ACC = 0.61; MESA: ACC = 0.48; MROS: ACC = 0.59; CFS: ACC = 0.49), while Chest-only configuration exhibits similar patterns (SHHS1: ACC = 0.62; SHHS2: ACC = 0.60; MESA: ACC = 0.48; MROS: ACC = 0.63; CFS: ACC = 0.55).

To further validate clinical applicability across severity levels, we analyze classification performance using different AHI cutoffs (Supplementary Tables 7–9). The results demonstrate consistent performance across severity transitions. Particularly for early SA detection (AHI cutoff at 5), the $SpO_2$-only configuration achieves excellent Macro F1 scores across all test cohorts (SHHS2: 0.842, MESA: 0.827, MROS: 0.805, CFS: 0.877), closely matching the Gold standard (SHHS2: 0.874, MESA: 0.829, MROS: 0.860, CFS: 0.901). This is particularly significant for population-level screening and early intervention. However, while the $SpO_2$-only configuration performs well overall, specific challenges remain in certain cohorts.

Based on the single-channel $SpO_2$ configuration, MROS exhibits low sensitivity in the healthy category (Supplementary Fig. 7), consistent with literature[17], possibly because its healthy subjects display characteristics similar to abnormal breathing events. To address this, the Gold configuration incorporates additional respiratory channels, providing complementary information to improve sensitivity in such challenging scenarios. However, the Gold configuration

underperforms the single-channel $SpO_2$ configuration in MESA, which can be attributed to differences in breathing patterns across racial groups, leading to distributional shifts. Additionally, the respiratory features captured by the additional channels may not generalize well across diverse groups, further reducing its effectiveness in MESA. For AHI prediction, reported under the Gold configuration, SHHS2 achieves the lowest MAE at 2.92, explaining its superior classification performance, while MESA shows the highest MAE at 4.76 due to racial variability and the aforementioned challenges with the Gold configuration. In comparison, MROS and CFS also benefit from the Gold configuration, achieving MAEs of 3.23 and 3.44, respectively, reflecting their improved performance over the single-channel $SpO_2$ configuration in these cohorts.

We further evaluate the Non-overlap SHHS model on MESA, MROS, and CFS cohorts (Supplementary Fig. 2 and Supplementary Table 3), where the number of training subjects decreases significantly compared to the All-Sub SHHS model. Results (Supplementary Fig. 3) show no significant differences between the two models across most cohorts, including SHHS1, SHHS2, MESA, and MROS ($P > 0.05$). For instance, in the MESA cohort, the Gold configuration achieves Macro F1 scores of 0.7406 (Non-overlap SHHS model) versus 0.7351 (All-Sub SHHS model, $P = 0.3053$). However, in the CFS cohort, the p-value is 0.0348 (Macro F1: 0.7798 versus 0.7860), indicating slight statistical significance but not a particularly strong difference. These results suggest that our method maintains strong performance despite reduced training subjects in most cases, showcasing its robustness and adaptability.

**Cross-population stability assessment of AIX**

We conduct comprehensive stability analysis across demographic and ethnic groups. For demographic subgroups, we evaluate two representative configurations (Gold and $SpO_2$) across gender, age, and BMI categories using data from SHHS2, MESA, MROS, and CFS (Fig. 4d).

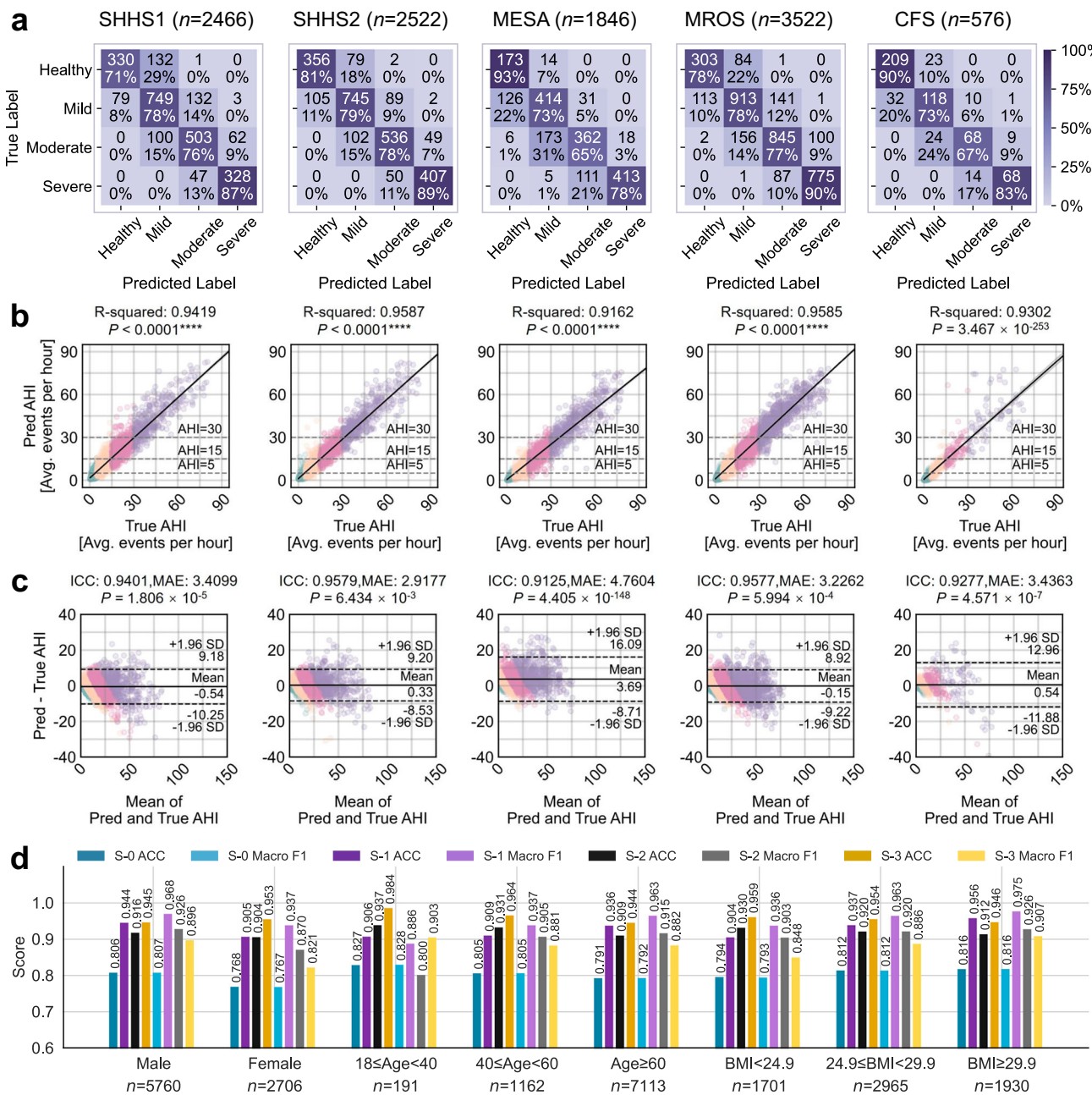

**Fig. 4 | External test cohorts (SHHS1, SHHS2, MESA, MROS, CFS) are evaluated for performance, with the model trained on SHHS1 data. a** The confusion matrix shows the classification into four severity levels. **b** The scatter plot illustrates the correlation between manual AHI and predicted AHI. The R-squared ($R^2$) value is calculated, along with the diagonal representing the linear regression model, the 95% CI, and the two-sided $p$-value. **c** The Bland-Altman plot displays the comparison of manual and predicted AHI, with error lines positioned at ±1.96 standard deviation (SD). The $p$-value is computed using the two-sided Wilcoxon signed-rank test, and the intraclass correlation coefficient (ICC) is provided, along with the mean of the mean absolute error (MAE) of predicted AHI for all subjects. Green, yellow, pink,

and purple scatter points represent healthy individuals, mild SA, moderate SA, and severe SA patients, respectively. **d** Performance of AIX under Gold channel configuration across different gender, age, and BMI groups. S−0, S−1, S−2, and S−3 represent four classification scenarios: four-level SA classification, AHI < 5 versus AHI ≥ 5, AHI < 15 versus AHI ≥ 15, and AHI < 30 versus AHI ≥ 30. *Note: Results for SHHS1 and SHHS2 are obtained using the Non-overlap SHHS model, which excludes overlapping subjects between the two cohorts to ensure unbiased evaluation and temporal performance assessment. Results for MESA, MROS, and CFS are based on the All-Sub SHHS model, which utilizes all available SHHS1 subjects to maximize training data.

The Gold configuration maintains stable performance across different groups. For gender, it achieves a Macro F1 score of 0.81 for males ($n = 5760$) and 0.77 for females ($n = 2706$). Across age groups, performance remains consistent, with scores of 0.83 for young adults ($18 \leq$ age $< 40$, $n = 191$), 0.81 for middle-aged adults ($40 \leq$ age $< 60$, $n = 1162$), and 0.79 for elderly populations (age $\geq 60$, $n = 7113$). Similarly, for BMI categories, the configuration demonstrates reliability,

achieving scores of 0.79 for normal-weight individuals (BMI < 24.9, $n = 1701$), 0.81 for overweight individuals ($24.9 \leq$ BMI < 29.9, $n = 2965$), and 0.82 for obese individuals (BMI ≥ 29.9, $n = 1930$). Minor variations are observed in females, elderly subjects, and those with lower BMI, potentially due to physiological differences. Importantly, similar patterns of stability are observed under the $SpO_2$ configuration, reinforcing these findings (Supplementary Fig. 25).

For ethnic diversity validation, we analyze system performance across different racial groups using SHHS2, MESA, and CFS cohorts, which include substantial populations of diverse backgrounds (Supplementary Table 5). The ethnic distribution varies significantly across cohorts, with MESA showing the highest diversity (Asian 12.08%, Black 26.60%, White 37.70%, Hispanic 23.62%), while SHHS predominantly comprises White subjects ($>85\%$). Results demonstrate consistent performance across racial groups under both Gold and $SpO_2$ configurations (Supplementary Table 6). In MESA cohort, $SpO_2$ configuration achieves comparable accuracy across Asian (ACC = 0.731, $R^2 = 0.950$), Black (ACC = 0.760, $R^2 = 0.930$), White (ACC = 0.741, $R^2 = 0.916$), and Hispanic (ACC = 0.766, $R^2 = 0.935$) populations. Similar consistency is observed in SHHS2, where performance remains stable across Black (ACC = 0.760, $R^2 = 0.951$) and White (ACC = 0.783, $R^2 = 0.955$) populations. For Asian population validation, we combine both retrospective ($n = 327$, ACC = 0.783, $R^2 = 0.953$) and prospectively collected ($n = 265$, ACC = 0.789, $R^2 = 0.921$) cohorts from FDU-HSH, which together validate the system's racial generalizability with a single-channel $SpO_2$ configuration. These results demonstrate AIX's robustness in addressing racial distribution shifts while maintaining stable performance across diverse populations.

Moreover, to assess temporal stability and potential training bias, we conduct a longitudinal analysis using 2647 subjects who appear in both SHHS1 and SHHS2 cohorts (approximately 5 years apart). We specifically focus on the $SpO_2$-only configuration given its optimal balance between performance and accessibility. The model trained on non-overlapping SHHS1 subjects ($n = 3146$) demonstrates remarkable temporal stability when tested on the overlapping subjects at both time points (Supplementary Figs. 1, 8 and Supplementary Table 4). For SHHS1 data ($n = 2466$), it achieves ACC = 0.785, Macro F1 = 0.791, with $R^2 = 0.9491$ and ICC = 0.9477 (MAE = 3.20 ± 3.42) for AHI prediction. The performance remains consistent when tested on SHHS2 data ($n = 2,518$, ACC = 0.778, Macro F1 = 0.783, $R^2 = 0.9535$, ICC = 0.9524, MAE = 3.28 ± 3.62), with confusion matrices showing stable classification patterns, particularly for severe cases (SHHS1: SEN = 0.888, SHHS2: SEN = 0.910). Bland-Altman analysis further confirms the model's temporal stability with comparable mean differences and limits of agreement between time points. This temporal robustness suggests the model's potential for reliable longitudinal monitoring of SA progression.

## Transparent scale diffusion for granular-level respiratory pattern visualization

We enhance the model's interpretability of respiratory event patterns by performing what we call transparent scale diffusion visualizations on external test cohorts. We average pool attention matrices with different levels of granularity output by the model along the feature dimension and project them onto the signal length, finally presenting them in the form of heatmaps. We publicly disclose the model's attention regions at different transparent scales (Supplementary Figs. 20 and 21) for Flow and $SpO_2$ signals (Fig. 5b, c), which provide key diagnostic criteria for SA[34]. In the Flow channel, the model highlights regions where airflow is reduced ≥90% (apnea) or ≥30% (hypopnea) with duration ≥10 s. In the $SpO_2$ channel, the model focuses on desaturation events with ≥3% reduction from baseline. The visualization results demonstrate that model attention aligns with standard AASM scoring guidelines for respiratory event identification[34]. This validates the significant advantages of next-generation transparent AI technology, enabling operators to understand the reasoning logic of AI models and enhancing decision-making confidence[36]. Quantitative comparisons of the transparent AI framework are detailed in the Methods section (Details and advantages of transparent scale diffusion technology).

## Feature separability and error analysis across channel configurations

Through t-Distributed Stochastic Neighbor Embedding (t-SNE) visualization[37], we illustrate the segment-level event features learned by the AI model based on four channel configurations (Fig. 5a). The original signal features exhibit highly overlapping distributions between respiratory and normal events, indicating the raw signals are not directly distinguishable. Among single channels, $SpO_2$ demonstrates better class separation compared to Flow and Chest signals, providing evidence for the feasibility of simplified home monitoring based on $SpO_2$. The Gold channel, integrating three channels, exhibits the most distinct class boundaries, indicating that the richness of information among different channel types contributes to improved feature discrimination. This is further validated by quantitative evaluations across test cohorts (Supplementary Tables 1–3, Supplementary Figs. 5–7 and 14). Flow (ACC = 0.482–0.621, Macro F1 = 0.390–0.607) and Chest (ACC = 0.484–0.625, Macro F1 = 0.485–0.614) show limited performance, while $SpO_2$ achieves more reliable results (ACC = 0.751–0.785, Macro F1 = 0.746–0.791), and Gold channel demonstrates the best performance (ACC = 0.738–0.810, Macro F1 = 0.737–0.817). The ROC analysis of binary event detection further confirms this trend, with AUC values ranging from 0.739–0.823 for Flow, 0.723–0.869 for Chest, 0.725–0.868 for $SpO_2$, and 0.750–0.880 for Gold channel (Supplementary Fig. 14).

Additionally, the performance of different channel types across varying SA severity levels is displayed. Figure 5f shows line plots depicting the MAE trends of model predictions for subjects with different severity levels across SHHS1, SHHS2, MESA, MROS, and CFS. The Gold configuration demonstrates consistently lower MAE compared to single channels, with overall MAE of 3.41 ± 3.64, 2.92 ± 3.47, 4.76 ± 5.56, 3.23 ± 3.32, and 3.44 ± 5.34 for SHHS1, SHHS2, MESA, MROS, and CFS, respectively. As severity increases, prediction errors show greater variation, with MAE increasing from 1.30 ± 1.54 in healthy subjects to 5.75 ± 5.63 in severe cases under Gold configuration in SHHS2, while $SpO_2$ shows a similar trend from 2.02 ± 2.00 to 5.46 ± 5.29. Single channels exhibit higher errors, with Flow showing MAE of 5.77 ± 5.99 and Chest showing 6.07 ± 6.58 overall in SHHS2 (Supplementary Tables 2 and 3). This suggests that different MAE tolerances are required for correctly classifying the severity of AHI predictions. Given the nonlinear variation of AHI thresholds with increasing SA severity, the AI model's ability to reasonably allocate MAE for subjects of different severity levels is crucial for improving classification performance. Lower and more stable MAE values indicate the general advantage of multi-channel configuration across various severity levels.

## Human-AI collaboration feasibility of AIX

We explore the feasibility of AI model interactions through analyzing prediction confidence patterns. Clustering analysis of segment-level predictions shows that incorrect predictions concentrate in boundary regions between clusters (Fig. 5d). Statistical analysis (two-tailed Mann-Whitney U test) examines the distribution of absolute probability differences between normal and respiratory events (Fig. 5e). The distribution of these differences shows distinct patterns. The median values for both true negatives (TN, $n = 302$) and true positives (TP, $n = 376$) exceed 0.9 (0.944 and 0.994, respectively), while incorrect predictions (FN + FP, $n = 52$) exhibit substantially smaller median value of 0.276. These distinctions are highly significant ($P = 7.07 \times 10^{-17}$ between TN and FN + FP, $P = 2.07 \times 10^{-19}$ between TP and FN + FP).

This quantifiable uncertainty pattern serves as the foundation of our trust assessment framework, providing users with reliable indicators for identifying cases requiring verification. The significant separation between confident and uncertain predictions enables a data-driven approach to building trust in the system's decision-making process. To systematically evaluate the system's trust-building

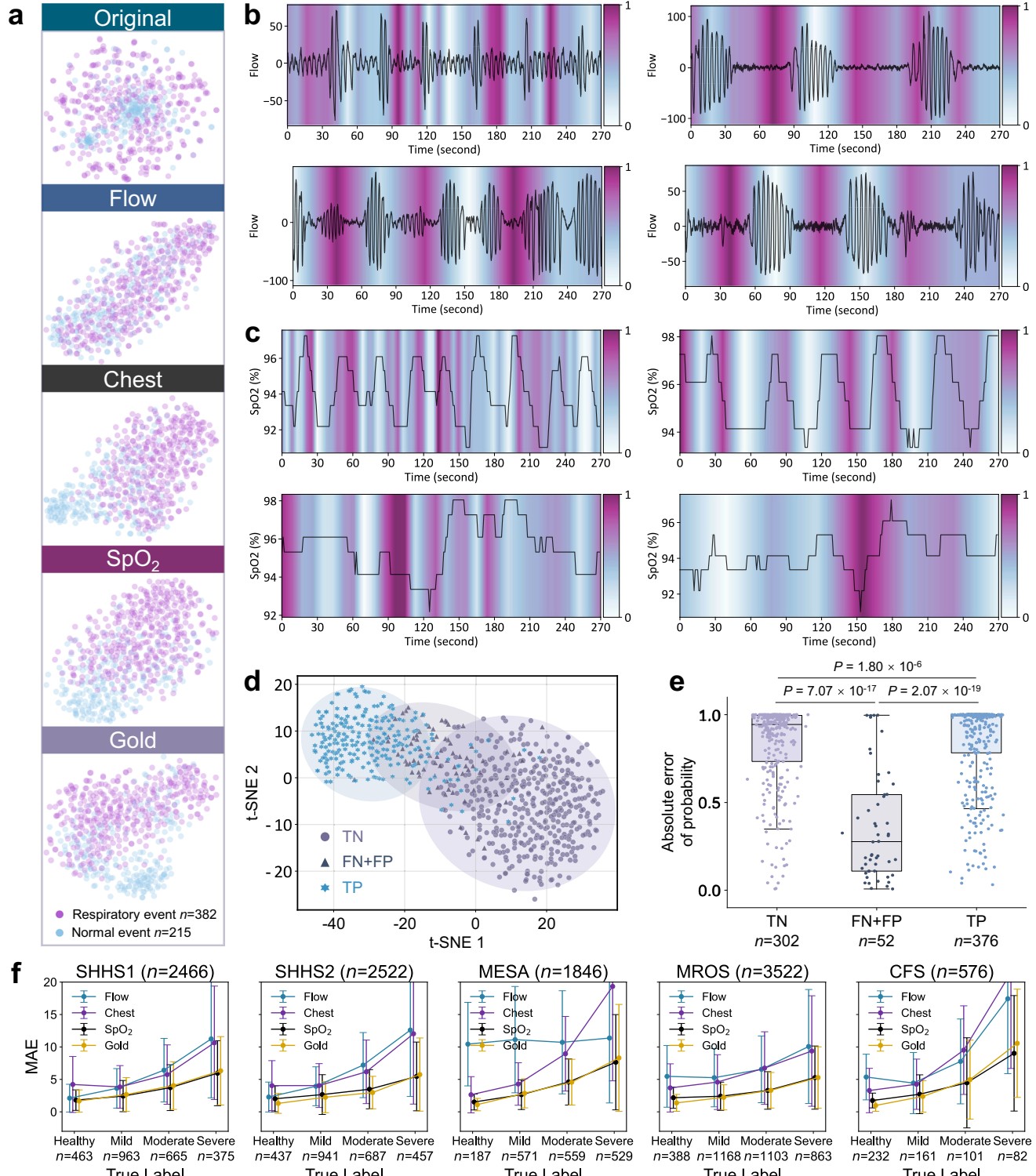

**Fig. 5 | Model interpretability and interaction feasibility. a** t-SNE visualization of original data and AI-extracted features from one subject across four channel types (Flow, Chest, SpO₂, and Gold), showing respiratory events ($n = 382$ epochs) and normal events ($n = 215$ epochs). **b**, **c** Model attention maps highlighting important regions in Flow and SpO₂ signals at different temporal scales, providing transparent decision logic. **d** t-SNE visualization shows separable clustering of true negative (TN), combined false negative and false positive (FN + FP), and true positive (TP). **e** Prediction confidence scores for TN ($n = 302$ epochs), FN + FP ($n = 52$ epochs), and TP ($n = 376$ epochs) from one subject's overnight consecutive 30-s epochs. Incorrect predictions show significantly lower confidence scores (two-tailed Mann-Whitney U test, $P = 7.07 \times 10^{-17}$, $P = 2.07 \times 10^{-19}$). Box plots display median values (central line), with box boundaries representing the 25th percentile (bottom) and 75th percentile (top), and whiskers extending to 1.5 times the interquartile range. **f** MAE trends (mean ± SD) across SA severity levels for SHHS1 ($n = 2466$ subjects), SHHS2 ($n = 2522$ subjects), MESA ($n = 1846$ subjects), MROS ($n = 3522$ subjects), and CFS ($n = 576$ subjects) under different channel configurations.

capabilities through interactive review, we conduct experiments on both FDU-HSH retrospective ($n = 327$) and prospective ($n = 265$) cohorts using SpO$_2$ monitoring (Supplementary Fig. 35, Supplementary Tables 10 and 11). Taking the FDU-HSH retrospective cohort as an example, we examine the relationship between interaction threshold intervals and error rates (Fig. 6a–c) to understand their trade-off patterns. For instance, with an interaction threshold interval of 0.1-0.2, we observe that 3.66% of samples are identified for review, with corresponding FP rate of 0.96% and FN rate of 0.22% (Supplementary Table 10). This illustrates how interaction thresholds can be leveraged to balance review efficiency and error control.

To validate the effectiveness of this trust-building framework, we conduct progressive performance validation experiments through selective review. The results demonstrate substantial improvements in diagnostic reliability. When reviewing 10.53% of cases (interaction threshold: 0.00–0.10), the binary event classification metrics improve from ACC = 0.867, SEN = 0.914, SPE = 0.851 to ACC = 0.911, SEN = 0.945, SPE = 0.899. Increasing review coverage to 20.40% (interaction threshold: 0.00-0.45) achieves even higher performance levels of ACC = 0.939, SEN = 0.966, SPE = 0.929 (Supplementary Table 11). This comprehensive trust assessment framework, combining uncertainty quantification, efficient review targeting, and progressive performance validation, enables users to build trust through transparent interaction while maintaining workflow efficiency. The framework provides concrete evidence of the AIX system's capability to achieve higher reliability through strategic human-AI collaboration, as demonstrated by the quantifiable performance improvements under different manual review coverage ratios.

### AIX real-world testing and SA risk assessment

AIX supports the use of single-channel SpO$_2$ data, enabling accessible and cost-effective SA monitoring in home settings. We validate its practical applicability through external testing on both FDU-HSH retrospective ($n = 327$) and prospective ($n = 265$) cohorts (Fig. 6d–h, Supplementary Fig. 28 and Supplementary Fig. 35). The prospective validation demonstrates excellent measurement agreement (ICC = 0.9174, MAE = 5.8330) and strong correlation ($R^2 = 0.9207$, $P < 0.0001$) in AHI prediction. The confusion matrix shows high sensitivity for healthy (100%) and severe (87%) cases, confirming the system's robust diagnostic performance. The system is implemented as a portable application supporting Bluetooth data transmission from wearable SpO$_2$ sensors[38] (Supplementary Fig. 30), demonstrating its potential for widespread deployment in home monitoring scenarios.

To complement AHI's event frequency measurement, we develop an overnight SA risk index (SARI) as a digital biomarker. SARI integrates both frequency and duration characteristics of respiratory events by averaging probabilities over 30-s windows throughout the night. Analysis of probability distributions within these windows reveals that SARI effectively captures two key aspects of respiratory events[39] (Fig. 6d): (1) longer event durations correlate with higher probability scores (Kruskal-Wallis $P = 8.96 \times 10^{-94}$ and $P = 7.28 \times 10^{-256}$, respectively), and (2) apnea windows consistently receive higher predictions than hypopnea windows across all duration ranges, with mean probabilities increasing from $0.704 \pm 0.414$ for hypopnea to $0.918 \pm 0.252$ for apnea events. These characteristics enable SARI to serve as a digital risk marker that quantifies both the occurrence and severity of respiratory events (Supplementary Figs. 26 and 27).

Through overnight monitoring examples (Fig. 6e and Supplementary Fig. 19), we demonstrate SARI's strong correlation with AHI (Spearman's $\rho = 0.960$, $P = 9.78 \times 10^{-182}$) and its ability to differentiate between severity levels (Fig. 6f, g, two-tailed Mann-Whitney $U$ test). SARI exhibits significant variation across AHI sub-intervals both with and without sleep status consideration (Kruskal-Wallis $P = 4.46 \times 10^{-53}$ and $P = 8.25 \times 10^{-54}$, respectively). The strong correlation between the two assessment approaches (Spearman's $\rho = 0.994$, $P = 1.94 \times 10^{-315}$)

demonstrates that SARI maintains consistent increasing trends with AHI regardless of sleep status consideration, validating its role as a robust severity indicator. These findings are further validated in the prospective cohort, where SARI maintains a strong correlation with AHI (Spearman's $\rho = 0.952$, $P = 3.80 \times 10^{-137}$) and demonstrates significant differences between adjacent severity levels (two-tailed Mann-Whitney $U$ test: $P = 1.47 \times 10^{-17}$ for healthy versus mild, $P = 1.27 \times 10^{-16}$ for mild versus moderate, and $P = 2.57 \times 10^{-19}$ for moderate versus severe).

## Discussion

Large-scale early detection and assessment of SA remain challenging due to the lack of accurate and user-friendly AI tools suitable for both clinical and home environments. Previous research has explored various approaches for automated SA diagnosis. Early studies focused on ECG-based methods due to their potential for wearable applications. A comprehensive review by Salari et al.[15] showed that most machine learning (ML) and deep learning (DL) models achieved accuracies over 70% in detecting SA episodes from ECG segments. A systematic evaluation by Bahrami et al.[16] demonstrated that hybrid DL models could achieve 88.13% accuracy in SA detection using a single-lead ECG. However, these approaches were primarily evaluated on the Apnea-ECG dataset[40] with only 70 subjects, and were limited to binary classification tasks at segment-level events or subject-level SA diagnosis. While segment-level detection allows rough estimation of AHI values, this may not fully reflect the clinical severity assessment that considers additional physiological and temporal factors. Using SpO$_2$ signals, Levy et al.[17] developed an innovative DL model called OxiNet based on 12,923 PSG recordings from six independent cohorts. Their model achieved ICC values of 0.92–0.96 and Macro F1 scores of 0.75–0.84 across multiple external test cohorts, demonstrating the potential of single-channel oximetry for SA diagnosis. Their work established strong foundations for future exploration of real-time respiratory event detection and model interpretability. Meanwhile, multi-channel approaches have demonstrated promising results in home settings. Wang et al.[41] proposed a wearable smart ring device using photoplethysmogram (PPG) sensors to collect pulse wave signals and SpO$_2$ synchronously. Using multiscale entropy and random forest algorithms, they validated their system on 10 subjects and achieved an accuracy of $85.99 \pm 2.26\%$ in identifying SA episodes compared to PSG, though further validation on larger cohorts would strengthen these findings. Retamales et al.[18] recently introduced DRIVEN using 10,643 PSG recordings from MESA, MROS and SHHS cohorts, correctly classifying 72.4% of test subjects into AHI severity classes using abdominal movement and SpO$_2$ sensors. Our recent work on IPCT-Net[25] leveraged 1603 HSAT recordings with a channel fusion framework. By comprehensively analyzing Flow, Chest, SpO$_2$, pulse rate (PR) and snoring (Snore) signals, our model achieved $R^2$ of 0.939–0.959 and ICC of 0.937–0.958 in AHI estimation, while demonstrating robust performance in four-level SA severity classification (ACC = 0.722–0.764, Macro F1 = 0.677–0.717), showing comparable effectiveness between simplified and multi-channel combinations. Despite these encouraging advances, current approaches face inherent limitations in clinical adoption. The lack of transparency in model decisions makes it difficult for clinicians to validate diagnostic reasoning. Models often struggle to adapt across different monitoring configurations and equipment types. Additionally, quantitative evidence supporting optimal channel selection in clinical practice remains sparse. To address these challenges, we develop and validate AIX, a transparent, explainable AI interactive framework that introduces expert-logic consistency in SA diagnosis. The system's robust performance in four-level SA severity classification (ACC = 0.74−0.81) and binary screening (AHI cutoff at 5) across multiple external test cohorts (ACC = 0.905-0.943 for Gold configuration, ACC = 0.885-0.933 for single-channel SpO$_2$ configuration) supports reliable clinical

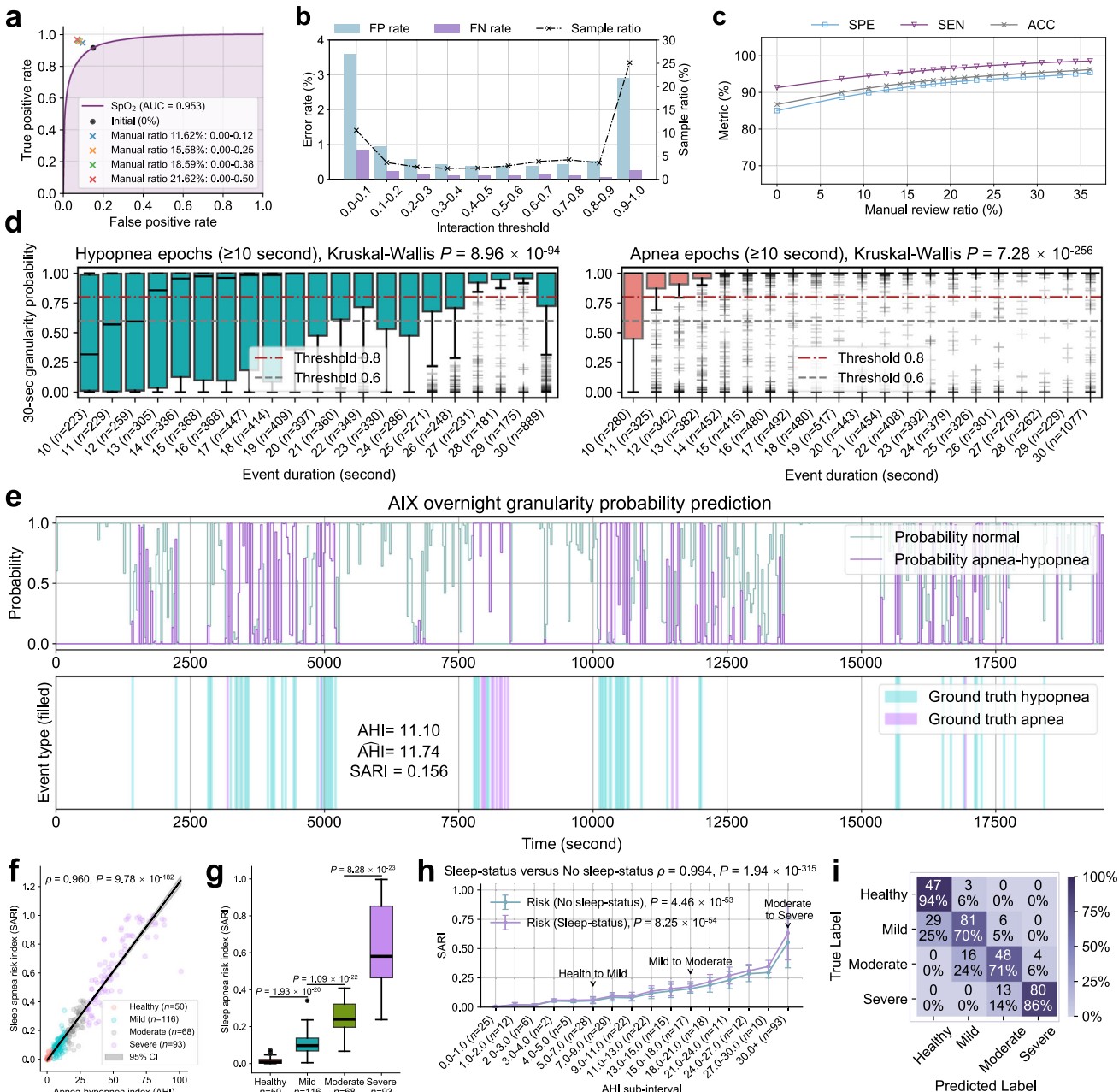

**Fig. 6 | AIX real-world scenario validation and risk assessment (FDU-HSH retrospective cohort, *n* = 327 subjects). a** The ROC curve for binary respiratory event classification by AIX based on single-channel SpO₂ signals. **b** The ternary relationship between FP rate, FN rate, and sample proportion retrieved by AIX based on different interaction threshold intervals. **c** The impact of different manual review proportions on the performance of binary respiratory event classification. **d** The probability prediction distribution of different hypopnea/apnea durations in 30-s epochs across subjects, displaying the predicted probability boxplot. Hypopnea epochs (*n* = 7075 epochs) show moderate probability scores (mean ± SD: 0.704 ± 0.414) with higher variability, while apnea epochs (*n* = 8715 epochs) demonstrate consistently high probability scores (mean ± SD: 0.918 ± 0.252). Kruskal-Wallis tests reveal significant differences across duration groups for both hypopnea (*P* = 8.96 × 10⁻⁹⁴) and apnea epochs (*P* = 7.28 × 10⁻²⁵⁶). **e** AIX overnight granularity prediction for one subject, showing predicted probabilities for normal breathing and apnea-hypopnea events, ground truth event annotations, along with

calculated AHI and sleep apnea risk index (SARI) values. **f** The correlation between SARI and AHI (two-sided Spearman's rank correlation test). The gray shaded area represents the 95% CI for the predicted mean values. **g** SARI values across SA severity groups (*n* = 327 subjects). Two-tailed Mann-Whitney U tests: healthy versus mild (*P* = 1.93 × 10⁻²⁰), mild versus moderate (*P* = 1.09 × 10⁻²²), moderate versus severe (*P* = 8.28 × 10⁻²³). **h** Comparison of SARI distributions calculated with and without sleep status across AHI sub-intervals (*n* = 327 subjects). Data are presented as mean ± SD. Kruskal-Wallis tests show significant differences across AHI sub-intervals for both conditions (No sleep-status: *P* = 4.46 × 10⁻⁵³, Sleep-status: *P* = 8.25 × 10⁻⁵⁴). Two-sided Spearman's rank correlation test shows strong agreement between the two approaches (*ρ* = 0.994, *P* = 1.94 × 10⁻³¹⁵). **i** Confusion matrix of four-level SA severity classification by AIX on the FDU-HSH retrospective cohort. Box plots in (**d**, **g**) show median values (central line), with box boundaries representing the 25th percentile (bottom) and 75th percentile (top), and whiskers extending to 1.5 times the interquartile range.

stratification while maintaining accessibility through simplified monitoring configurations. The significance of this work surpasses traditional diagnostic tools in several important aspects.

Most importantly, AIX, built upon a transparent AI interactive framework, represents a paradigm shift in clinical AI applications. While previous research has primarily focused on improving prediction accuracy[15,16,18,25], clinical adoption of AI has been hampered by physicians' distrust of black-box decisions. Our investigation reveals that AI transparency not only preserves performance but also enhances diagnostic effectiveness. Across five large external test cohorts ($n = 10,932$), the AIX system achieves ACC values of 0.74–0.81 and $R^2$ values of 0.92–0.96 in four-level severity classification, with 99.8% of subjects either correctly classified or differing by only one severity grade. More significantly, our real-world validation in home environments demonstrates AIX's robust performance using simplified monitoring configurations. In the retrospective cohort ($n = 327$), AIX maintains an ACC of 0.78 in four-level severity classification using nocturnal oximetry signals alone, with binary event classification accuracy and sensitivity improving from 0.867 and 0.914 to 0.939 and 0.966, respectively, through transparent AI-guided interactive review of approximately 20% ambiguous cases. These findings are further validated in our prospective cohort ($n = 265$), where the system maintains excellent performance metrics (AUC = 0.943 for respiratory event detection, ACC = 0.789 for four-level severity classification, $R^2 = 0.921$ and ICC = 0.917 for AHI prediction) when analyzing single-channel $SpO_2$ signals. The SARI metric demonstrates significant discriminative power between adjacent severity categories ($P < 0.0001$ for all transitions), supporting its utility in risk stratification. These comprehensive validation results from both cohorts provide strong evidence for AIX's reliability in simplified monitoring scenarios while maintaining high diagnostic standards. The transparent scale diffusion mechanism introduced by AIX uniquely illuminates the reasoning process underlying abnormal respiratory event detection. In contrast to previous black-box approaches that merely provide final predictions, AIX's attention mechanism enables multi-level decision interpretation from individual respiratory events to whole-night conclusions. This transparency allows both experts and non-professionals to comprehend and validate system decisions, proving crucial for clinical adoption and trust establishment. The capability to trace event-level logic not only facilitates efficient result review by clinicians but also provides interpretable guidance for home users. By delivering detailed, interpretable respiratory analysis, AIX enables more precise subject stratification and outcome assessment in practice. This alignment between AI and clinical reasoning accelerates the integration of AI tools in both clinical and home settings.

Furthermore, AIX achieves flexible sensor configuration adaptation while maintaining high performance, which has profound implications for healthcare delivery models. Traditional approaches either rely on complex PSG setups that limit accessibility or employ simplified signals at the expense of diagnostic accuracy. The flexible configuration of AIX provides a continuous spectrum of monitoring options, enabling healthcare providers to optimize the balance between diagnostic detail and accessibility based on individual subject needs. This facilitates alternative care delivery models, such as initiating with simplified home screening and progressively incorporating additional sensors only when necessary. Such adaptive monitoring simultaneously reduces healthcare costs while preserving care quality. This versatility makes AIX particularly suitable for expanding SA diagnosis beyond specialized sleep laboratories into accessible population screening tools. The AIX workflow comprises: (1) generating 30-s granularity abnormal respiratory event predictions aligned with sleep stages, accompanied by detailed interpretable analysis, and (2) predicting AHI and the risk indicator SARI based on whole-night probability estimation sequences. This workflow preserves AIX's capability to interpret specific events throughout the sleep process.

AIX demonstrates exceptional reliability in real-world applications. Our validation using the SHHS cohort, which comprises unattended home sleep recordings where signal quality and subject compliance may vary significantly, provides compelling evidence for AIX's adaptability to home monitoring scenarios. The validation across these community-based cohorts demonstrates AIX's effectiveness in populations with diverse characteristics. SHHS was designed to investigate cardiovascular outcomes in sleep-disordered breathing with participants having various cardiovascular comorbidities, including hypertension, coronary heart disease, and cerebrovascular disease. MESA focused on subclinical cardiovascular disease across ethnic groups, specifically enrolling middle-aged and older participants initially free of clinical cardiovascular disease to study disease progression. The CFS specifically examined familial aggregation of SA, demonstrating that sleep-disordered breathing clusters within families independently of obesity. MROS enrolled older men aged 65 years or older who were not selected on the basis of sleep problems or cognitive impairment to study osteoporotic fractures and cardiovascular outcomes, with participants having age-related comorbidities such as cognitive impairment and cardiovascular conditions. The robust performance maintained across these heterogeneous populations demonstrates AIX's capability in handling diverse clinical scenarios. From family-based studies investigating genetic factors to large community-based cohorts with varying comorbidity profiles, AIX consistently delivers reliable results, indicating its readiness for real-world clinical applications involving complex interactions of genetic and environmental factors. Unlike many AI methods that perform well only in laboratory settings, AIX successfully overcomes key challenges, including training data distribution bias and model overfitting[17]. Validation using five large retrospective test cohorts and two external real-world test cohort demonstrates AIX's generalization capability beyond single-center studies. The system exhibits remarkable stability across different gender, age, and BMI groups, establishing a foundation for practical clinical implementation (Fig. 4d and Supplementary Fig. 25). Particularly noteworthy is AIX's performance in terms of racial fairness (Supplementary Table 6). While the open-source NSRR database primarily comprises U.S. populations (including Black or African American, Hispanic, and White individuals), the incorporation of Asian population data from Fudan University Huashan Hospital expands the system's racial applicability, promoting more democratic AI healthcare management[42]. Our large-scale validation across different clinical information and racial subgroups reveals crucial insights into algorithmic fairness in healthcare. The consistent performance across demographic characteristics suggests universal features in SA respiratory patterns. However, subtle performance variations observed between subgroups indicate the need for further investigation into potential physiological differences in SA manifestation across populations, with important implications for improving AI system design and deepening understanding of disease pathophysiology.

The low operational threshold and flexible monitoring scenario adaptation of AIX significantly advance SA diagnosis popularization while reducing costs (Supplementary Fig. 34). Traditional Lab-PSG sleep studies, requiring high-cost equipment (>$50,000) and several hours of technical time in strictly controlled laboratory environments, prove difficult to extend to home monitoring practices. While HSAT provides a more economical option with moderate equipment costs (approximately $10,000–20,000), it still demands specialized technical expertise and considerable analysis time. In contrast, AIX's flexible workflow accommodates multiple device configurations, including portable PSG systems and simple pulse oximeters, with substantially lower per-test costs and reduced analysis time (1–10 min). This adaptability enables AIX to generate valuable analysis reports even with missing channel data, making it particularly suitable for home monitoring scenarios. Although AASM guidelines recommend single-night sleep studies for uncomplicated subjects' SA diagnosis, evidence

regarding night-to-night variability suggests the necessity of repeated monitoring[43]. Through providing objective and consistent AI-assisted diagnosis, AIX establishes a practical paradigm for economically feasible long-term disease progression monitoring.

Several limitations warrant consideration in this study. While our validation demonstrates robust performance in large-scale cohorts, these data predominantly originate from developed healthcare systems. Future evaluation of system performance in resource-limited settings with suboptimal monitoring conditions remains necessary. Despite validation across multiple cohorts with varying study designs and population characteristics demonstrates robustness, systematic evaluation of AIX's performance in specific clinical subgroups, such as those with strong family history or particular comorbidities, remains an important direction for future research. Furthermore, while our results validate performance in home-based settings, further studies are needed to evaluate AIX across different home monitoring devices and in resource-limited settings. Additionally, while $SpO_2$-based monitoring demonstrates excellent capability in detecting clinically significant SA, there are inherent physiological limitations in characterizing specific respiratory event types when using single-channel monitoring alone. Particularly, $SpO_2$ signals cannot definitively differentiate between central and obstructive events, as both may present similar desaturation patterns without the corresponding respiratory effort and airflow information[34]. The synchronous assessment of respiratory effort presence and airflow cessation, which is essential for distinguishing between central and obstructive events, cannot be captured through $SpO_2$ monitoring alone. However, our research demonstrates that for the primary goal of identifying SA patients requiring further evaluation, $SpO_2$-based analysis maintains robust diagnostic effectiveness, particularly in large-scale screening applications where accessibility is paramount. Another challenge lies in achieving accurate sleep staging with limited channels. Although previous studies indicate that respiratory effort signals can achieve reliable sleep-wake classification[18], $SpO_2$ signals alone show limitations in this aspect[17]. Future research could explore integrating additional channels (such as PPG or respiratory signals) under minimal interference principles to achieve more precise sleep staging[44]. Furthermore, while AIX currently focuses on respiratory event detection, its modular architecture allows future integration of additional channels like electroencephalogram (EEG), electrooculogram (EOG), and electromyogram (EMG) for comprehensive sleep stage analysis when such detailed clinical assessment is necessary[45]. A crucial future research direction involves evaluating whether AIX's detailed respiratory analysis can reveal early markers of SA progression or treatment response. The system's ability to track subtle respiratory pattern changes may enable disease progression identification before clinical symptoms appear, facilitating more proactive and personalized treatment approaches[46,47].

Our comprehensive analysis of demographic feature integration reveals intriguing insights into the relationship between physiological signals and subject characteristics in SA assessment. Through extensive experiments with both feature concatenation and cross-attention mechanisms (Supplementary Fig. 36), we find that continuous physiological signals alone achieve optimal diagnostic performance across cohorts. The AIX system demonstrates robust performance with Gold channel (ACC = 0.7951–0.8057, Macro F1 = 0.7853–0.8083) and maintains strong diagnostic capability even when simplified to a single-channel $SpO_2$ configuration (ACC = 0.7743–0.7851, Macro F1 = 0.7672–0.7883). As demonstrated in Supplementary Figs. 37 and 38, both configurations maintain high performance metrics without demographic features, with $R^2$ values (0.9254–0.9579 for Gold, 0.9334–0.9551 for $SpO_2$) and ICC coefficients (0.9225–0.9570 for Gold, 0.9311–0.9540 for $SpO_2$) consistently outperforming the demographic-assisted approaches (single-channel $SpO_2$). Regarding the choice of monitoring channels,

our current focus on specific physiological signals is driven by practical considerations for widespread clinical adoption. While EEG provides valuable information for detailed sleep staging and comprehensive sleep analysis, its complex setup procedures and requirement for expert interpretation make it less suitable for large-scale screening programs, particularly in home settings. Similarly, video monitoring, while informative, raises significant privacy concerns and poses substantial data storage challenges that could impede long-term monitoring capabilities. These practical limitations, combined with our findings on the robust performance of simplified physiological monitoring, support our strategic approach of optimizing the balance between diagnostic accuracy and implementation feasibility for initial screening purposes.

Looking ahead, AIX demonstrates tremendous potential in transforming SA diagnosis from a specialized procedure into a widespread screening tool. Building on the system's transparency and flexibility features, we envision developing smartphone applications to track fine-grained nocturnal events, significantly expanding screening coverage through interpretable decision support. In conclusion, AIX not only advances SA diagnostic technology but, more significantly, demonstrates how transparent AI can transform clinical practice. By making AI decision processes interpretable and adaptable to various clinical environments, it provides a blueprint for developing AI systems that clinicians can trust and integrate into practice. The principles demonstrated here can guide the development of transparent AI solutions for other complex medical diagnostic challenges.

## Methods
### Ethical approval
This research complies with all relevant ethical regulations. For the FDU-HSH Sleep Study, the study protocol was approved by the Institutional Review Board (IRB) of Huashan Hospital, Fudan University (Approval No. KY2021-811), and all participants provided written informed consent. For the historical databases (SHHS, MESA, MROS, and CFS), data were used in accordance with the original ethical approvals and data use agreements from the NSRR. The authors affirm that human research participants provided written informed consent for publication of the images and videos in Supplementary Figs. 30, 34, and Supplementary Videos 1–3.

### Statistics & reproducibility
This study employed a cohort analysis design using five established retrospective sleep study cohorts (SHHS1 $n = 5793$, SHHS2 $n = 2651$, MESA $n = 2056$, MROS $n = 3930$, CFS $n = 730$) for model development and validation, with additional real-world testing performed on two FDU-HSH cohorts (retrospective $n = 350$ and prospective $n = 297$) to develop and validate the AI-based SA diagnostic system. No statistical method was used to predetermine sample size. Sample sizes were determined by the availability of high-quality PSG recordings from the selected databases. Data were excluded based on predefined quality control criteria: lack of required physiological channels (Flow, Chest, or $SpO_2$), sleep durations under 4 h, missing AHI annotations, and signal quality issues ($SpO_2$ artifacts and signals with SD < 0.0001). The experiments were not randomized. The investigators were not blinded to allocation during experiments and outcome assessment. Model performance was evaluated using accuracy, sensitivity, specificity, F1 score, area under the ROC curve for classification tasks, and $R^2$, intraclass correlation coefficient, and mean absolute error for regression tasks. Statistical comparisons were performed using the Kruskal-Wallis test for multi-group comparisons, the Mann-Whitney U test for pairwise comparisons, and Spearman's rank correlation for correlation analyses. Source code and experimental configurations have been made publicly available to ensure reproducibility.

## Large-scale and multi-ethnic cohort description

All overnight PSG data in this study consist of three components: multi-channel digital signals stored in European Data Format (EDF), overnight annotation files in XML format (including the onset and duration of respiratory and other abnormal events, along with 30-s sleep stage annotations), and subject-level clinical baseline information. For sleep staging performed using the Rechtschaffen & Kales (R&K)[48] standard, we standardize it to the AASM[34] criteria. In addition, we statistically assess the significant differences in all clinical information among different SA severity groups (Supplementary Table 1). The cohorts used in this study are described as follows:

(1) **SHHS cohort:** The SHHS cohort includes two longitudinal multicenter cohorts, referred to as Visit 1 and Visit 2. The aim is to investigate the interaction between sleep-disordered breathing and cardiovascular health, as well as other health factors, particularly the association with coronary heart disease, stroke, and hypertension risk. Visit 1 includes men and women aged 40 years or older, while Visit 2 involves a subset of participants who underwent a second PSG exam. The number of participants providing analyzable files in the two cohorts is 5793 and 2651, respectively. The timeframes for these visits range from November 1995 to January 1998 for Visit 1, and from January 2001 to June 2003 for Visit 2.

(2) **MESA cohort:** The MESA cohort is a longitudinal investigation conducted by six collaborating centers, involving American men and women aged 45 to 84 from Black, White, Hispanic, and Chinese ethnic backgrounds. As a multi-ethnic study, its primary goal is to determine whether there are associations between subclinical atherosclerosis and sleep, as well as sleep disorders, across different genders, races, or other demographic variations. The cohort provides analyzable files for 2056 participants.

(3) **MROS cohort:** The MROS cohort is conducted between December 2003 and March 2005 as part of the sleep research initiative within the Osteoporotic Fractures in Men study. The objective is to assess the extent to which sleep disorders are linked to adverse health outcomes, such as increased mortality, fractures, falls, and cardiovascular disease risk. Participants underwent a comprehensive PSG. The cohort includes recordings from two visits, with a total of 3930 analyzable participant files.

(4) **CFS cohort:** The CFS cohort is a large-scale family-based study designed to quantify the familial aggregation of SA. The cohort data is collected during PSG monitoring in a clinical laboratory setting, with 730 analyzable subject files available. This cohort includes a nearly equal proportion of White and Black participants.

(5) **FDU-HSH cohort:** The FDU-HSH cohort is a recent investigation focusing on sleep, with all participants being of Asian descent, aiming to address the lack of representation of this ethnicity in publicly available cohorts. This cohort comprises a retrospective cohort of 350 participants enrolled from January 2021 to December 2023 in clinical laboratory settings and a prospective cohort of 297 participants monitored from December 2023 to December 2024 in home environments. The retrospective cohort establishes baseline validation in clinical settings, while the prospective cohort evaluates system performance under genuine home sleeping conditions with maintained high-quality reference standards. The primary goal is to assess the potential of the AIX system in real-world data applications. For both cohorts, overnight PSG recordings are available, along with analyses of subject age, BMI, and AHI. The annotation standards for respiratory events are consistent with public cohorts (SHHS, MESA, MROS, and CFS), referencing the NSRR annotation variable *ahi_aOh3a*, while sleep stage annotations follow the AASM guidelines.

For historical databases (SHHS, MESA, MROS, and CFS), respiratory events are initially scored without preset criteria for desaturation or arousal. Instead, sleep scoring software (such as Compumedics Profusion) preliminarily identifies apnea and hypopnea events based on airflow reductions lasting more than 10 s[49]. The scoring system then links these events to SpO2 and EEG data, enabling researchers to retrospectively apply different standardization criteria to generate AHI variables. To ensure scoring consistency, we specifically select the AHI variable *ahi_aOh3a* that complies with AASM 2012 recommended rules from these cohorts[34]. This has been confirmed through direct communication with NSRR database administrators in previous cross-cohort research[17].

For our recently collected Asian cohort (FDU-HSH cohort collected after 2012), events are directly scored following AASM 2012 guidelines, with all scoring performed by certified sleep technicians. This standardized approach enables us to maintain consistent definitions of apnea and hypopnea events, ensuring valid comparisons of AHI measurements across all cohorts regardless of their original collection period.

## Different channel types for different scenarios

In clinical practice, scorers primarily rely on three types of information to annotate apnea events: Flow, Chest, and SpO2. Combining these three types of information is expected to aid in the physiological detection of corresponding respiratory events. Other information, such as snoring (Snore)[50] and pulse rate (PR)[51], can also provide further assistance. However, considering the significant number of subjects in public cohorts who lack these two channel types, they are not included. It is also worth exploring whether simplifying commonly used signal combinations in SA screening can meaningfully replace the traditional integration of the three classic types of information, pointing towards simplified and more comfortable monitoring of sleep-disordered breathing.

This study explores the topic using large-scale, multi-ethnic, multi-center sleep data, providing a foundation for the reliability of the conclusions. Our investigation includes three single-channel types (Flow/Chest/SpO2) and one multi-channel type, Gold (the integration of Flow, Chest, and SpO2). Recent large-scale studies have compellingly demonstrated that SpO2-based analysis can achieve robust SA diagnosis, with only 0.2% missed moderate-to-severe cases compared to 21% for traditional metrics across multiple validation cohorts[17]. The selection of SpO2 as a primary single channel is supported by both clinical and practical considerations. From a clinical perspective, SpO2 directly reflects the physiological impact of breathing cessation during SA, and the AASM guidelines specifically include oxygen desaturation (≥3%) as a key criterion for scoring respiratory events[34]. The practical advantages of SpO2 are substantial. It requires only a simple finger sensor for non-invasive monitoring, demonstrates superior signal stability with fewer artifacts compared to Flow signals requiring nasal cannulas or Chest signals requiring thoracic bands, and enables comfortable long-term monitoring in both clinical and home-based settings. These characteristics make SpO2 particularly valuable for expanding screening coverage while maintaining diagnostic reliability. Some studies suggest that SA screening and classification based on single-channel SpO2 information is feasible, but there is a lack of consistent cross-sectional comparisons across subject populations, especially between single-channel SpO2 information and the performance of the Gold channel type. Therefore, this study examines the significance of balancing accuracy and comfort in various channel combinations. We consider two scenarios applicable to SA management: the first is a more professional-grade analysis based on the Gold channel, as it includes all the necessary information for manual review, helping to correct ambiguous samples from AI models. The second is based solely on the SpO2 channel, which is easier to implement in more accessible home settings, where participants only need to monitor

SpO$_2$ throughout the night. Finally, since Flow and SpO$_2$ contain morphological features of SA episodes, AI interpretability analysis based on these two channels is valuable, as reflected in this study's consideration of transparency in AI model design logic.

## Analysis of AHI-related factors under different gender conditions

Considering the potential differences in SA phenotypes between genders, we analyze the associations between AHI, BMI, and age using all available measurement data from SHHS1, SHHS2, MROS, CFS, and FDU-HSH (male, $n = 7013$; female, $n = 4694$) to understand the relationships of these variables across different gender groups (MESA is excluded due to missing BMI data). This analysis helps reveal the extent to which gender influences the characteristics and pathophysiological features of SA patients (Supplementary Fig. 29). First, we divide the cohorts into male and female categories and derive three correlation analysis groups for each gender: AHI-BMI, AHI-age, and age-BMI. We then construct scatter plots and calculate Spearman's $\rho$ for the variable pairs, with statistical significance assessed using two-sided tests. To better reflect the information of each variable, we also provide probability density estimates for all data points. In the AHI-BMI and AHI-age analyses, different colors (blue, light yellow, red, and black) represent healthy (AHI < 5), mild ($5 \leq$ AHI < 15), moderate ($15 \leq$ AHI < 30), and severe (AHI $\geq$ 30) subjects, respectively, according to the AASM guidelines[34]. In the age-BMI group analysis, youth ($18 \leq$ age < 30, light yellow), middle-aged adults ($30 \leq$ age < 60, red), and older adults (age $\geq$ 60, black) are distinguished by different color scatter points. The results show that the correlation between BMI and AHI is similar in both males ($\rho = 0.2824$, $P < 0.0001$) and females ($\rho = 0.2882$, $P < 0.0001$). However, in the correlation analysis between age and AHI, females exhibit a significant positive correlation ($\rho = 0.3085$, $P < 0.0001$), while males show no significant correlation ($\rho = 0.0206$, $P = 0.0846$), suggesting that age may be a particularly important factor for women but not for men. Additionally, we observe a significant negative correlation between age and BMI in males ($\rho = -0.1848$, $P < 0.0001$), while the correlation is weaker in females ($\rho = -0.0401$, $P < 0.01$), indicating that these factors are relatively independent in women.

## Evaluation approach

The algorithm is evaluated using several metrics: accuracy (ACC, the ratio of the number of correctly classified samples among the total number of samples), sensitivity (SEN, the ratio of correctly classified events to all true events), specificity (SPE, the ratio between correctly classified non-events and all non-events), positive predictive value (PPV, the ratio of correctly classified events in all recognized events), F1 score (the harmonic mean of SEN and PPV), AUC (area under the ROC curve), and $R^2_{AHI}$ ($R^2$ of AHI)[17,18]:

$$ACC(\%) = \frac{TP + TN}{TP + TN + FP + FN} \tag{1}$$

$$SEN(\%) = \frac{TP}{TP + FN} \tag{2}$$

$$SPE(\%) = \frac{TN}{TN + FP} \tag{3}$$

$$PPV(\%) = \frac{TP}{TP + FP} \tag{4}$$

$$F1(\%) = \frac{2 \times SEN \times PPV}{SEN + PPV} \tag{5}$$

$$AUC = Area\,(under\,ROC\,curve) \tag{6}$$

$$R^2_{AHI} = 1 - \frac{\sum_{sub}(AHI_{true} - AHI_{pred})^2}{\sum_{sub}(AHI_{true} - AHI_{mean})^2} \tag{7}$$

where TP is true positive, TN is true negative, FP is false positive, and FN is false negative.

## Signal quality control and data preprocessing

Due to the complexity of sleep monitoring and the susceptibility to interference signals, there are factors that disrupt the accurate AHI assessment in subjects. First, SpO$_2$ artifacts may occur due to loose sensor attachment or excessive movement. We exclude these contaminated intervals to improve AHI evaluation accuracy.

Second, subjects may improperly wear airflow sensors or thoracic bands before and after formal sleep testing, resulting in flat-line recordings with negligible signal variation. We calculate the standard deviation (SD) for these signal segments and apply an exclusion threshold of 0.0001. Signal segments with SD below this threshold are excluded. To ensure reliable sleep evaluation, we remove recordings where the duration of valid signals during sleep is less than 4 h. This data cleaning process ensures a more accurate AHI assessment.

The sampling rates of physiological signals vary considerably across different sleep cohorts. Specifically, the airflow and thoracic signals are sampled at 32 Hz in MESA and CFS cohorts, 16 Hz or 32 Hz in MROS cohort, and predominantly 8 Hz or 10 Hz in SHHS cohort. The SpO$_2$ signals maintain a consistent sampling rate of 1 Hz across all cohorts. To address these variations, we implement a standardized resampling approach that converts all 270-s segments to a fixed length of 256 points. This resampling strategy is theoretically sound as normal human respiratory rates during sleep typically range from 12 to 20 breaths per minute (0.2–0.33 Hz)[52], and even during respiratory events, the fundamental frequency rarely exceeds 0.5 Hz[53]. Our resampling frequency ($256/270 \approx 0.95$ Hz) satisfies the Nyquist sampling theorem, ensuring adequate capture of respiratory event characteristics. For SpO$_2$ signals, this sampling rate is also sufficient given the relatively gradual nature of blood oxygen variations. The effectiveness of this approach is validated by our model's consistent performance across multiple external validation cohorts with different original sampling rates, demonstrating successful preservation of discriminative features necessary for accurate SA detection.

## Performance comparison across sampling rates

To assess the impact of sampling rates on model performance, Macro F1 scores are compared across signal types (Flow, Chest, SpO$_2$, and Gold) and four sampling rates: 0.95 Hz, 1 Hz, 4 Hz, and 10 Hz, selected based on clinical standards and prior literature[54–56]. The lowest rate (0.95 Hz) corresponds to performance calculated for 256-length signal segments (270-s windows), while 1 Hz reflects the relatively stable nature of SpO$_2$ signals compared to respiratory signals. To handle varying input resolutions, the TSD-Net's Stem layer is replaced with a stack of convolutional layers that reduce inputs to a fixed length of 64. High-resolution inputs (e.g., 10 Hz) require more layers, while low-resolution inputs (e.g., 0.95 Hz, 1 Hz) require fewer. These layers are configured with appropriate kernel sizes, strides, and paddings, ensuring effective feature extraction and dimensionality reduction.

Signals are downsampled for each configuration, and Macro F1 scores are evaluated across test cohorts (SHHS2, MESA, MROS, CFS, FDU-HSH). Error bars represent 95% CIs to capture performance variability. Results indicate stable performance across most sampling rates, with minor variations (Supplementary Figs. 9 and 10). For instance, in the Flow channel, SHHS1 achieves a Macro F1 score of

0.603 (95% CI: 0.591–0.615) at 0.95 Hz and 0.630 (95% CI: 0.624–0.636) at 10 Hz, showing slight improvements at higher rates. However, the Gold configuration, which integrates Flow, Chest, and $SpO_2$ signals, exhibits consistent performance across rates. For example, in SHHS2, Gold achieves 0.810 (95% CI: 0.807–0.814) at (Flow: 0.95 Hz, Chest: 0.95 Hz, $SpO_2$: 0.95 Hz) and 0.809 (95% CI: 0.806–0.812) at (Flow: 10 Hz, Chest: 10 Hz, $SpO_2$: 1 Hz), suggesting that $SpO_2$ compensates for information loss caused by downsampling respiratory signals. For $SpO_2$ signals, performance differences between 0.95 Hz and 1 Hz are negligible. In SHHS2, Macro F1 scores are 0.786 (95% CI: 0.783–0.789) at 0.95 Hz and 0.787 (95% CI: 0.781–0.793) at 1 Hz, reflecting the stability and low variability of $SpO_2$ signals. Notably, in cohorts like MESA, the Flow channel shows a performance decline at lower sampling rates (e.g., 0.429 at 10 Hz versus 0.399 at 0.95 Hz). However, the Gold configuration maintains robust performance, achieving 0.741 (95% CI: 0.731–0.751) at (Flow: 10 Hz, Chest: 10 Hz, $SpO_2$: 1 Hz) compared to 0.735 (95% CI: 0.728–0.742) at (Flow: 0.95 Hz, Chest: 0.95 Hz, $SpO_2$: 0.95 Hz).

These findings demonstrate that while higher sampling rates slightly improve performance for isolated respiratory signals (Flow, Chest), the Gold configuration effectively mitigates the impact of downsampling by integrating $SpO_2$ signals. This underscores the robustness of the proposed method and its potential for practical applications in resource-constrained environments, such as wearable devices and home-based monitoring systems, where lower sampling rates reduce computational demands without compromising performance.

## Granularity prediction model architecture

The study proposes the TSD-Net to comprehensively explore local and global dependencies of one-dimensional physiological signals[38,57], achieving cross-scale attention decision disclosure. TSD-Net consists of four stages, as shown in Supplementary Fig. 12.

Stages 1 and 2 use the multi-scale convolutional attention (MSCA) module to capture local dependencies and model local attention. The MSCA module includes three key components: a multi-head convolutional encoder (MHCE), a cross-scale encoding fusion (CSEF) module, and a squeeze-and-excitation network (SE-Net). MHCE aims to expand the diversity of receptive fields for a comprehensive learning of local features, while CSEF effectively integrates multi-scale local fine-grained information from different heads, maintaining the network's lightweight structure. SE-Net applies attention mechanisms to the channel dimensions of feature maps, further enhancing information fusion. Stage 3 combines the MSCA module with the multi-head self-attention (MHSA) module to achieve a smooth transition from local to global dependencies. The final Stage 4 uses only the MHSA module to efficiently capture long-range dependencies. Detailed parameters are presented in Supplementary Table 12.

(1) **MSFA block:** The MSCA block constitutes a primary component in both stage 1 and stage 2, comprising the MHCE module, the CSEF module, and SE-Net (Supplementary Fig. 12b).

I. **MHCE module:** As shown in Supplementary Fig. 12c, the MHCE module splits the input channels into $N$ distinct heads and applies separable convolutions with different depths to each. This approach effectively captures cross-scale features, enhancing local detail capture while also reducing parameter size and computational costs:

$$MHCE(X) = Concat(DW_{k_1 \times k_1}(x_1), \ldots, DW_{k_n \times k_n}(x_n)) \quad (8)$$

Here, $x_1$ to $x_n$ split the original features into multiple heads along the channel dimension, focusing on different granular features through various convolutional kernel sizes $k_1$ to $k_n$.

II. **CSEF module:** The CSEF module recombines and groups local features of various granularities produced by the MHCE module

(Supplementary Fig. 12d). It selects one channel from each head to form a group and performs contextual feature fusion within each group using an inverted bottleneck structure, enhancing multi-scale feature diversity. Finally, it aggregates information through pointwise convolution, facilitating the fusion of information from multiple heads within the MHCE module. The formula is as follows:

$$M = W_{agg}([G_1, G_2, \ldots, G_M]) \quad (9)$$

$$G_i = W_{group}([H_1^i, H_2^i, \ldots, H_N^i]) \quad (10)$$

$$H_j^i = DWConv_{k_j \times k_j}(x_j^i) \in \mathbb{R}^{L \times 1} \quad (11)$$

where $W_{group}$ and $W_{agg}$ represent the group and aggregation weights of pointwise convolutions, respectively. The $i$-th channel corresponds to the group $G_i$, which consists of $N$ different heads $H_j^i$.

III. **SE-Net:** The SE-Net structure (Supplementary Fig. 12e) enhances the network's feature modeling capability through an attention mechanism. It adaptively learns the correlations among channels and dynamically adjusts the importance of different channels in the feature maps. This allows the model to focus precisely on the most meaningful features for the task.

In conclusion, assuming there exists a feature input $X$, with weights $W_s$ and $W_v$ for two linear layers, the MSFA module modulates the contribution of $V$ through branch $M$ to obtain the final output $Z$. The formula is as follows:

$$Z = SENet(M \odot V) \quad (12)$$

$$V = W_v X \quad (13)$$

$$M = CSEF(MHCE(W_s X)) \quad (14)$$

(2) **MSCA-MHSA block:** Supplementary Fig. 12f presents the composition of this block, which includes a MSCA block and a MHSA block. By interleaving the stacking of MSCA and MHSA, the transition from local information to global information is accomplished, effectively reducing computational complexity. This stacking process facilitates the evolution of information from local to global, thereby optimizing computational efficiency.

(3) **MHSA block:** For stage 4, this study exclusively employs the MHSA block to efficiently capture long-range dependencies[21]. Supplementary Table 12 provides detailed model parameters for different parameter configurations. The model is composed of a stem layer and stage 1 through stage 4. The stem layer includes a 1D convolution layer with a stride of 2 and an output dimension of 64 (64-d), and a batch normalization (BN) layer. Stage 1-4 each consist of two MSCA blocks, one MSCA-MHSA block, and one MHSA block. *dim 64* denotes an embedding dimension of 64, and *sam.ep.r 2* indicates an expansion rate of 2 for the CSEF module.

For a one-dimensional physiological signal $P$, its dimension is defined as $[B,C,L]$, where $B$ represents batch size, $C$ represents the number of channels, and $L$ represents the length of the signal. In this study, with an input dimension of [512,1,256], after passing through the

Stem layer, the dimension becomes [512,64,64], with the number of channels reduced to 64 and the length down-sampled by a factor of 4 to 64, resulting in an output feature map size of 64 × 64. Subsequently, in the second stage, the feature map resolution is halved, and the number of channels is doubled, resulting in an output feature map size of 128 × 32. The structures of the third and fourth stages are similar to the second stage, with their output feature map sizes being 256 × 16 and 512 × 8, respectively. The final output is obtained by passing through AdaptiveAvgPool1d, Flatten, a 1-d layer, and finally, using Sigmoid to output probabilities.

Unlike traditional convolutional neural networks (CNNs) that focus only on local patterns or recurrent neural networks (RNNs) that process sequences step by step, TSD-Net combines the advantages of both convolutional and Transformer architectures. The MSCA modules effectively capture local respiratory patterns through multi-scale convolutions, while the MHSA modules, based on Transformer self-attention mechanisms, excel at modeling long-range physiological dependencies without the sequential constraints of RNNs. This hybrid architecture is particularly suitable for SA analysis, where both local respiratory events and their long-term temporal relationships are crucial for accurate diagnosis.

### Lightweight metric comparison and evaluation of TSD-Net

This study introduces the TSDNet series of models, including two variants: TSDNet-B and TSDNet-T, and compares them with multiple baseline models (including ResNet18, ResNet34, ResNet50, ResNet101, EfficientNet-B0, 1D ViT-S, 1D ViT-B, 1D ViT-L, and 1D ViT-H) across four key performance metrics (Supplementary Fig. 15a–d).

Model parameter analysis (Supplementary Fig. 15a) demonstrates TSDNet's excellent lightweight characteristics. TSDNet-T contains 3.7 M parameters, while TSDNet-B has 10.0 M parameters. Both variants are significantly smaller than most comparison models, such as 1D ViT-H (629.7 M) and even lighter than EfficientNet-B0 (16.0 M).

Computational complexity evaluation (Supplementary Fig. 15b) shows TSDNet's exceptional FLOPS performance. TSDNet-T achieves 0.05 Gmac FLOPS, the lowest among all comparison models, while TSDNet-B operates at 0.15 Gmac FLOPS. For reference, other models like 1D ViT-B and ResNet50 require substantially higher FLOPS (1.45 Gmac and 0.20 Gmac, respectively).

Inference time testing (Supplementary Fig. 15c), conducted on a 12th Gen Intel(R) Core(TM) i5-12600KF, 3.70 GHz CPU with 10 repeated experiments using samples of shape [1,1,256], reveals that TSDNet-T achieves 9.8 ms average inference time, while TSDNet-B requires 28.9 ms. Both outperform larger models such as 1D ViT-H (158.2 ms) and 1D Swin-B (61.9 ms).

Classification performance evaluation (Supplementary Fig. 15d) across four independent cohorts (SHHS2, MESA, MROS, and CFS) demonstrates TSDNet's superior effectiveness. Despite its lightweight design, TSDNet achieves higher classification performance (AUC = 0.887, 95% CI: 0.882-0.892) compared to all baseline models, including larger architectures like ViT-B (AUC = 0.872, 95% CI: 0.868-0.876) and ResNet101 (AUC = 0.856, 95% CI: 0.851-0.861).

TSDNet successfully achieves an excellent balance between model complexity, computational efficiency, and classification performance. Its lightweight design and strong performance make it particularly suitable for deployment in resource-constrained environments while maintaining high classification accuracy. TSDNet-B provides solutions for tasks requiring stronger expressiveness through slightly increased complexity, while TSDNet-T offers an ideal choice for lightweight application scenarios. This flexibility makes the TSDNet series particularly suitable for various application scenarios, from resource-constrained mobile devices to complex systems requiring higher precision, proving its potential as a highly promising, scalable, and transparent DL architecture.

### Details and advantages of transparent scale diffusion technology

The AIX system incorporates a transparent scale diffusion mechanism, which is a technology driven by a next-generation of transparent AI. To comprehensively evaluate our interpretability framework, we conduct systematic comparisons between TSD-Net and the widely-used Grad-CAM[58] method implemented on a 1D ResNet50 backbone across four external validation cohorts (SHHS2, MESA, MROS, and CFS). From each cohort, we randomly select 50 overnight recordings (total $n = 200$), with particular emphasis on analyzing 30-s windows containing positive respiratory events accompanied by expert second-by-second annotations. Both TSD-Net and ResNet50 are trained with identical configurations, including early stopping criteria, to ensure fair comparison. The attention matrices from all four stages of TSD-Net and Grad-CAM attention are normalized and thresholded at 0.5 to generate high-confidence regions. Quantitative assessment using Intersection over Union (IoU) metrics (Supplementary Fig. 24 and Supplementary Table 13) demonstrates TSD-Net's superior performance, with Stage 3 achieving mean IoU scores of 0.253-0.356 for Flow and 0.228–0.323 for SpO$_2$ across cohorts, consistently outperforming Grad-CAM (mean IoU of Flow: 0.145–0.273, mean IoU of SpO$_2$: 0.140–0.258), providing a solid decision-making basis for human interactive review.

The TSD-Net consists of four stages, each meticulously designed to capture features at different scales. Comparative analyses (Supplementary Figs. 22 and 23) demonstrate that TSD-Net's four-stage hierarchical feature importance analysis offers more comprehensive interpretability than the single-layer visualization provided by Grad-CAM on ResNet50. Stages 1 and 2 employ MSCA blocks, which include MHCE module, CSEF module, and SE-Net. The MHCE module effectively captures cross-scale features by segmenting input channels and applying separable convolutions of different depths. The CSEF module enhances the diversity of multi-scale features by reorganizing and grouping multi-granularity features from MHCE. SE-Net introduces an attention mechanism, further enhancing the network's feature modeling capability. Stage 3 combines MSCA blocks with MHSA blocks, achieving a smooth transition from local to global dependencies. Finally, Stage 4 exclusively uses MHSA blocks to efficiently capture long-range dependencies. This progressive feature extraction process starts with an input of [$B$,1,256] and gradually extracts and fuses features through four stages, ultimately resulting in a highly abstract feature map of [$B$,512,8]. The feature map resolution halves at each stage while the number of channels doubles, allowing the model to capture information at different scales, thus achieving a comprehensive understanding of complex physiological signals at different resolutions.

The core advantage of transparent scale diffusion technology lies in its visualization capability based on multi-scale attention matrices. As demonstrated in Supplementary Figs. 22–24, and Supplementary Table 13, TSD-Net exhibits robust attention distribution across all stages, with Flow channel maintaining stable high IoU scores (SHHS2: 0.315–0.330, MESA: 0.227–0.253, MROS: 0.346–0.353, CFS: 0.328–0.360) through different stages and SpO$_2$ channel showing progressive improvement from Stage 1 to Stage 4 (SHHS2: 0.218–0.328, MESA: 0.175–0.244, MROS: 0.207–0.340, CFS: 0.257–0.398). Statistical analysis shows that each stage of TSD-Net significantly outperforms Grad-CAM across all cohorts ($P < 0.001$ except for one comparison, Supplementary Table 13), demonstrating its superior feature representation capability. By average pooling the attention matrices output at each scale and projecting them onto the signal length to form heatmaps, the system provides multi-level interpretability perspective for the model's decision-making process, preserving richer and more detailed transparent explanatory power for physiological signal analysis.

## LightGBM model for channel feature integration

We use the traditional ML model LightGBM[35] to perform integrated prediction on intermediate features from the Flow, Chest, and $SpO_2$ channels. Based on the granularity prediction model, 8 features are output for each channel within a 30-s sliding window. By concatenating the features, we generate 24 features for the three channels, and then LightGBM is used to predict respiratory and normal events. We set the maximum number of leaves per tree to 31, with a learning rate of 0.05, and use 90% of the features in each iteration to avoid overfitting. The training is conducted for 100 rounds.

To justify our choice of LightGBM and ensure its superiority for our specific task, we conduct a comprehensive comparison with other classical ML models. We consider six classical ML classifiers: Support Vector Machine (SVM), artificial neural network (ANN), K-Nearest Neighbors (KNN), Random Forest (RF), Extreme Gradient Boosting (XGB), and LightGBM. We use data from SHHS1 for training, employing 11-fold cross-validation, and apply the models to test a mixed dataset from SHHS2, MESA, MROS, and CFS. The trained models are compared using ROC curves (95% CI), and the training and testing times for a single sample are measured. On both validation and test sets, RF (AUC 0.988–0.992, 0.819–0.823) and LightGBM (AUC 0.989–0.992, 0.818–0.822) demonstrate superior performance (Supplementary Fig. 16). For a single sample, the average training and testing times (conducted on a 12th Gen Intel(R) Core(TM) i5-12600KF, 3.70 GHz CPU) are 14.34 and 0.79 microseconds for RF, and 3.80 and 0.54 microseconds for LightGBM, respectively. To enhance the operational fluidity of AIX, we adopt the LightGBM model.

To further understand the contribution of different channel features in the integration process, we conduct SHAP (SHapley Additive exPlanations) analysis on the LightGBM model. As shown in Supplementary Fig. 17, the feature-level SHAP distribution reveals the relative importance and impact of different respiratory features extracted by TSD-Net. Each point represents a sample, with color indicating the feature value (red for high, blue for low) and horizontal position showing the SHAP value (positive values indicate increased likelihood of respiratory events). The features are ordered by their mean absolute SHAP values, providing insight into which features most strongly influence the model's predictions across different channels.

Furthermore, we perform a quantitative comparison of SHAP value magnitudes across respiratory channels (Flow, Chest, and $SpO_2$) as presented in Supplementary Fig. 18. This channel-wise analysis demonstrates that $SpO_2$-derived features generally exhibit higher SHAP values (mean ± SD: $3.39 ± 0.02$) compared to Flow ($0.54 ± 0.02$) and Chest ($0.73 ± 0.01$) features, aligning with clinical observations about the importance of oxygen desaturation in SA diagnosis. This analysis not only validates our feature integration strategy but also provides interpretable insights into how different channels contribute to the final prediction.

## AHI regression model architecture

For different channel types, the input shape of a single-channel signal is $[T_{sec} × F_s, 1]$, where $T_{sec}$ represents the number of seconds and $F_s$ represents the sampling rate. We train a granularity prediction model for each channel type. For single-channel scenarios, the granularity prediction model directly outputs the full-night probability sequence (normal probability and respiratory event probability). In multi-channel scenarios, the granularity prediction model acts as a feature extractor, outputting intermediate features for each channel type, which are concatenated and fed into LightGBM to generate the full-night probability sequence (Supplementary Fig. 11). For each subject, the length of the output full-night probability sequence is resampled to 1024 using the resample function from the Python SciPy library.

Considering the generalization and prediction speed of the regression model, we define a simple AHI regression architecture, consisting of three 1D convolutional layers and two fully connected

layers. The input to the regression model has a shape of $[B, 2, 1024]$, where $B$ represents the number of subjects, and 2 represents the normal probability and respiratory event probability (Supplementary Fig. 13).

## Model training

(1) **Model 1-granularity prediction model (TSD-Net):** Training was conducted for a maximum of 100 epochs, with an early stopping patience of 15 epochs and a batch size of 512. The optimization during training employed the AdamW optimizer with a weight decay set to 0.05[59]. The learning rate was reduced from 0.0001 to 0.00001 using the cosine annealing schedule.

The focal loss function was utilized in Model 1, which aims to reduce the weight of classes with a large number of samples[60]:

$$FL(p_t) = - \alpha_t(1 - p_t)^\gamma \log(p_t) \tag{15}$$

where $p_t$ denotes the predicted probability of belonging to the true class. $\gamma$ represents the focusing parameter, which smoothly adjusts the weight ratio of easy-to-classify samples. In this work, we set $\gamma$ to 2.

The key architectural hyperparameters of the model were optimized through Bayesian optimization over 100 trials. The training set was split into 80% for training and 20% for validation during the optimization process. Six critical parameters were explored: MLP expansion ratio (range: 2–8), channel expansion ratio (range: 1–4), dropout rate (range: 0.0-0.5), attention dropout rate (range: 0.0–0.5), path dropout rate (range: 0.0–0.3), and head convolution kernel size (options: 3, 5, 7, 9). Other architectural parameters, such as the number of blocks in each stage (2, 2, 8, 1), convolutional attention heads (4, 4, 4, 1), and Transformer attention heads (-1, -1, 8, 16), were adopted from validated configurations in previous studies[57,61]. The parameter configuration achieving the lowest validation loss was selected for the subsequent experiments (Supplementary Fig. 33).

(2) **Model 2-AHI regression model:** Training was conducted for up to 100 epochs, with an early stopping patience of 15 epochs and a batch size of 512. During training, the optimization used the AdamW optimizer with a learning rate of 0.001, and the loss function is MSE loss:

$$MSE = \frac{1}{N}\sum_{B=1}^{N}(AHI_{true} - AHI_{pred}) \tag{16}$$

(3) **Experimental environment:** As the computing environment for network training, the PyTorch (Ver. 1.13) DL framework was employed using the NVIDIA A100 GPU (NVIDIA Corporation, Santa Clara, CA, USA) with 80 GB VRAM in Python 3.9.

## Hyperparameter optimization

The hyperparameter optimization of TSD-Net (Model 1) was conducted through Bayesian optimization over 100 trials, revealing the relative importance of different architectural components in 30-s granular respiratory event detection (Supplementary Fig. 33). The *mlp_ratio*, controlling the hidden feature expansion in feed-forward networks, showed the highest importance and converged to 2. This moderate expansion suggests that the temporal patterns in physiological signals can be effectively captured without requiring extensive feature transformation capacity.

Dropout-related parameters (*drop_rate*, *attention_drop_rate*, and *path_drop_rate*) ranked as the second most important group but approached zero, indicating our four-stage progressive attention

design provides sufficient inherent regularization. The *expand_ratio*, which determines channel expansion in convolutional attention modules, stabilized at 2, achieving effective feature representation while maintaining computational efficiency.

The head convolution kernel size demonstrated relatively low importance and converged to 3. This small convolution window is sufficient for capturing local respiratory features in the initial embedding stage, preparing informative local representations for subsequent large-scale attention processing in deeper layers.

For the AHI regression model (Model 2), we adopted a proven architecture design with three convolutional layers using decreasing kernel sizes ($5 \rightarrow 3 \rightarrow 1$) followed by two fully connected layers[62]. This simple yet effective architecture helps prevent overfitting while enabling efficient multi-scale temporal feature extraction. The light-weight network structure with progressively reduced kernel sizes ensures both computational efficiency and robust prediction accuracy for clinical deployment.

## Data augmentation

To enhance the diversity of training data and improve the generalization performance of the model, we randomly add noise perturbations to the original inputs to create additional samples. This process applies to all channel signals after z-score normalization:

$$X_{\text{synthesis}} = X_{\text{raw}} + N \tag{17}$$

$$N \sim \mathcal{N}(0, \sigma_{\text{noise}}), \sigma_{\text{noise}} \in [0.005, 0.01] \tag{18}$$

where $X_{\text{synthesis}}$ represents the synthesized signal, $X_{\text{raw}}$ represents the original signal, and $N$ represents Gaussian-distributed noise. We randomly initialize SD $\sigma_{\text{noise}}$ to range from 0.005 to 0.01 to enhance the diversity of synthesis. This data augmentation process expands the sample variance of the input to the model, thereby mitigating overfitting.

## Transparency in overnight sleep monitoring

The AIX system consists of two key components: a 30-s granularity event prediction model (Model 1) and a full-night, 30-s granularity probability AHI regression model (Model 2). The transparency scale diffusion mechanism based on Model 1 helps reveal the model's attention to scale differences at the granularity of events, providing auxiliary interpretations for different channel features. From Model 1, we obtain a full-night, 30-s probability sequence for an individual subject, which is resampled to a length of 1024 (approximately 8.53 h) to standardize the input sequence length for Model 2. Based on this, we average the respiratory event probabilities in the sequence to generate the overnight SA risk index (SARI), which serves as an interpretable risk marker. This metric incorporates the varying lengths of respiratory events between 30-s windows, whereas the probability values help reflect these differences, providing a more nuanced consideration of event durations than the AHI value (Supplementary Fig. 11).

Additionally, we define an interaction threshold, which is the absolute difference between the model's 30-s probability outputs (normal and respiratory event). This metric aims to identify ambiguous predicted events and, based on the transparency scale diffusion mechanism, offers reference points for manual review of the model's logic. As illustrated in Fig. 6a–c, the selective review of a small number of ambiguous samples can further enhance the model's prediction performance.

## The concept and cost analysis of the AIX system

The AIX system consists of a user-friendly interactive interface that facilitates operator analysis and an AI model backend designed for overnight multi-channel data analysis (Supplementary Fig. 31). The data acquisition mode supports both overnight offline export and Bluetooth transmission, providing a comprehensive analysis of the subject's overnight sleep-breathing status. AIX accommodates various channel configurations, making it suitable for both professional and home monitoring scenarios. For the collected overnight raw data, the backend preprocessing algorithm removes poor-quality signals and applies appropriate filtering. The AIX system ultimately provides 30-s granularity visualizations (including surrounding signals), along with corresponding model prediction probabilities and event types. The system also displays SARI and AHI for the entire night. Additionally, attention heatmaps generated through a transparent scale diffusion mechanism are overlaid on each 30-s granular signal, offering decision logic for any optional segment. Moreover, the AIX system features an interactive sample indexing and search function, allowing rapid filtering of ambiguous samples for manual review by setting interaction thresholds. Finally, the one-click export function efficiently generates intelligent SA monitoring reports in batches (Supplementary Fig. 32). These reports contain detailed analyses of respiratory events and relevant physiological parameters during sleep, including AHI estimation and SA risk stratification assessment, to prompt subjects for timely further diagnosis and intervention.

To validate the cost-effectiveness of AIX, we conduct a comprehensive cost analysis comparing it with existing clinical solutions (Supplementary Fig. 34). All PSG devices included in the comparison are currently in clinical use, with their prices and average testing durations provided under the guidance of sleep medicine specialists. The prices of portable/simplified devices are obtained directly from equipment manufacturers, and the AIX system processing time is experimentally verified. All prices are converted to US dollars using current exchange rates. As shown in Supplementary Fig. 34, the AIX system demonstrates significant advantages in terms of equipment cost, per-test expenses, and technical time requirements. Traditional Lab-PSG systems require substantial initial investment (approximately $68,000) and involve high per-test costs ($1000–1500) with lengthy technical operations (3–4 h). While HSAT offers a more economical alternative with lower equipment costs ($14,000) and reduced per-test expenses ($200–400), it still requires considerable technical time (approximately 1 h). In contrast, the AIX system demonstrates remarkable cost efficiency and flexibility, supporting various monitoring devices from wearable sensors to portable PSG configurations. With equipment costs as low as $15 when using a single-channel finger pulse oximeter, and since AIX can be deployed on personal laptops without additional hardware requirements, the per-test costs remain under $10, which includes disposable supplies (such as nasal cannulas), brief training requirements, equipment depreciation, and basic technical support. The technical operation time is streamlined to 1–10 min through its transparent AI-assisted interaction logic.

## Demographic auxiliary value assessment in overnight AHI regression

To investigate the potential auxiliary value of demographic features in SA assessment, we explore their integration at the sequence level, hypothesizing that demographic risk factors might modulate the overall overnight respiratory patterns. We design two distinct approaches (Supplementary Fig. 36), operating on 1024-length sequences interpolated from 30-s window probabilities. The first approach employs a feature concatenation method, where demographic features (including gender, age, BMI, and race) are first processed through a demographic encoder (transforming from 4-dimensional input to 128-dimensional features through two fully connected layers), while the probability sequences are processed through three convolutional layers (with filter sizes of 5, 3, and 1, and channel dimensions of 16, 32, and 64) followed by max-pooling operations, as detailed in Supplementary Fig. 13. The flattened

sequence features (8 × 1024) are then concatenated with demographic features (128) before final AHI regression through two fully connected layers. The second approach utilizes a cross-attention mechanism, where the same convolutional architecture processes probability sequences into features of shape [$B$,64,128], while demographic features are mapped to 64 channels through a 1 × 1 convolution. A multi-head attention module (4 heads) then uses sequence features as queries and demographic features as keys and values for dynamic feature enhancement.

Taking the single-channel $SpO_2$ configuration as an example (Supplementary Fig. 38), experiments on SHHS2, MROS, and CFS cohorts show that both demographic feature integration approaches fail to improve performance compared to the baseline system. The concatenation approach achieves accuracies of 0.7633/0.7752/0.7465 and Macro F1 scores of 0.7714/0.7564/0.7449, while the cross-attention method yields accuracies of 0.7769/0.7687/0.7604 and Macro F1 scores of 0.7821/0.7259/0.7525, both underperforming the baseline AIX system (accuracies: 0.7831/0.7851/0.7743, Macro F1: 0.7883/0.7672/0.7750). The exceptional reliability of sequence probability features, validated by high $R^2$ values (0.9526/0.9551/0.9334) and ICC coefficients (0.9514/0.9540/0.9311), suggests that direct integration of static demographic features with dynamic sequence probabilities might not be optimal, leading us to explore more fundamental representation learning strategies.

### CLIP-inspired framework for fine-grained feature alignment

To address the challenge of effectively incorporating demographic information, we develop a CLIP[63]-inspired alignment framework operating at the 30-s segment level. This shift from sequence-level integration to representation learning was motivated by the observation that demographic features, while not directly beneficial for sequence-level prediction, might better serve as conditioning factors for learning more discriminative respiratory pattern representations.

We design the framework at the 30-s segment level (Supplementary Fig. 39), trained on 3,779,943 multi-channel segments from SHHS1 (2,631,479 normal events and 1,148,464 respiratory events). To ensure fair comparison, the framework utilizes identical TSD-Net architectures as encoders (without classification heads) for Flow, Chest, and $SpO_2$ signals, each producing 8-dimensional features projected to 64-dimensional embeddings through two fully connected layers with layer normalization and ReLU activation. The demographic encoder processes 4-dimensional features (gender, age, BMI, and race) through a deeper network of four fully connected layers ($4 \rightarrow 32 \rightarrow 64 \rightarrow 128 \rightarrow 64$) with layer normalization, ReLU activation, and dropout.

We implement three CLIP-based learning strategies: pairwise signal alignment between each pair of physiological signals, leave-one-out alignment comparing each signal against the mean representation of the other two[64], and signal-demographic alignment between each signal and demographic features. All features are normalized to unit length before computing scaled cosine similarities with a learnable temperature parameter. The model is trained using the AdamW optimizer (learning rate 1e-4) with composite loss functions weighted differently for various alignment tasks and cosine learning rate scheduling with warmup. Following CLIP's paradigm, the encoders are pre-trained through contrastive learning with only the classification head fine-tuned for downstream tasks.

The t-SNE visualization (Supplementary Fig. 40) reveals effective cross-channel alignment between physiological signals while maintaining a natural separation between physiological and demographic feature clusters. This separation reflects the inherent complementarity between dynamic respiratory patterns and static population-level risk factors. Through contrastive learning objectives, our framework facilitates meaningful interaction between these distinct feature modalities during representation learning, while preserving their respective characteristics essential for SA detection. Further performance evaluation (Supplementary Fig. 40) demonstrates the effectiveness of this design. While our fully-supervised TSD-Net framework demonstrates superior performance with sufficient labeled data (Flow: AUC 0.7853, Chest: 0.8147, $SpO_2$: 0.7982, Gold standard: 0.8202), the CLIP-inspired framework shows stable performance across varying amounts of labeled data (Flow: 0.6046–0.7535, Chest: 0.7065–0.7654, $SpO_2$: 0.6647–0.7645, Gold standard: 0.7731–0.7751), maintaining reasonable performance even with limited fine-tuning (1%–10%). These results establish the CLIP-inspired framework as a valuable complementary approach, particularly in scenarios with limited labeled data availability.

### Reporting summary
Further information on research design is available in the Nature Portfolio Reporting Summary linked to this article.

### Data availability
The SHHS, MESA, MROS, and CFS cohorts are publicly available from the National Sleep Research Resource (NSRR) with direct access links: SHHS, MESA, MROS, and CFS. These datasets can be accessed after appropriate deidentification and obtaining necessary permissions via their online portal. The FDU-HSH sleep study data cannot be made publicly available due to ethical approval requirements and participant privacy protections under institutional policies. Access restrictions exist because the data contain sensitive participant information subject to institutional review board approval. Researchers may request access to the FDU-HSH data by contacting the corresponding authors, with requests evaluated based on legitimate research purposes. Access requires completion of formal data use agreements with Huashan Hospital, Fudan University, and is limited to noncommercial academic research purposes. For the publicly available NSRR datasets (SHHS, MESA, MROS, and CFS), data access is typically granted within 2 weeks after completing the online registration and data use agreement process. For FDU-HSH data access requests, researchers can expect a response within 4-6 weeks from the date of initial contact, as requests require IRB evaluation and formal approval processes. Source data are provided with this paper.

### Code availability
The source code can be accessed under the MIT license (https://github.com/fdu-harry/Apnea-Interact-Xplainer) and is archived on Zenodo (https://doi.org/10.5281/zenodo.15788888).

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

## Acknowledgments

This work was supported in part by National Natural Science Foundation under Grant 62371138 (C.Y.), in part by Shanghai Municipal Science and Technology Major Project under Grant 2023SHZDZX02C01 (H.Y.), and in part by STI2030-Major Projects under Grant 2022ZD0214000 (C.F.).

## Author contributions

S.H. conceived and designed the study, developed the methodology, performed experiments and data analysis, implemented the system, created visualizations, and drafted the manuscript. J.L. contributed to methodological development, system design and architecture, and revised the manuscript. Y.W. assisted with system evaluation, methodology validation, and provided substantial revisions to the manuscript. C.F. contributed to clinical data collection, validation, and clinical guidance and provided funding support. J.Z. contributed to clinical data collection and validation and provided clinical guidance. H.Y. provided data access, clinical guidance and validation, supervised part of the research and provided funding support. C.Y. supervised the entire project, provided critical manuscript revision, overall guidance and funding support. All authors reviewed and approved the final version.

## Competing interests

The authors declare no competing interests.
