## [Transparent Peer Review file · Nature Communications]

Transparent Artificial Intelligence-enabled Interpretable and Interactive Sleep Apnea Assessment across Flexible Monitoring Scenarios

Corresponding Author: Professor Cuiwei Yang

Version 0:

Reviewer comments:

Reviewer #1

(Remarks to the Author)

This study aims to develop and evaluate a new system for the automatic detection of sleep apnea. Three overnight signals are utilized for this purpose, although different single-channel configurations can also be employed. Two concatenated deep learning techniques are applied to estimate the apnea-hypopnea index (AHI), which is the clinical variable used to diagnose sleep apnea. Additionally, explainable artificial intelligence (XAI) is incorporated to enhance the reliability of the system. Concordance and diagnostic metrics are high, addressing a significant problem. The objectives of the study are relevant, and the narrative is clear. However, I have identified some major and minor concerns regarding the study, which I detail below:

Major issues:

1/ Please, clearly specify and explain how you dealt with the different sample rates of each signal, including the final sample rate used for each of them and a justification for that election when needed.

2/ There is a clear bias in using all SHHS1 to train the model as more than 2.000 subjects are the same in SHHS2 test set (five years later). I think the study will be less biased if the subjects in SHHS1 that are also in SHHS2 are used as a different test set. This will be particularly useful to make the cross-population stability robust. In addition, it will allow a longitudinal analysis in terms of performance and explainability.

3/ Please specify and clearly explain and justify how the hyperparameters of the deep learning models were chosen. Apparently, there is not a validation group for this specific purpose, which introduces a potential bias.

4/ Databases were formed at different times and under different apneic events annotation rules. please clarify how the authors dealt with this problem.

5/ Scientific discussion is scarce. There is a lack of scientific discussion as for example comparison with other similar approaches (there are a lot of them), or hyperparameter discussion etc. Moreover some statements are not properly justified. Example "The low operational threshold and flexible monitoring scenario adaptation of AIX significantly advance SA diagnosis popularization while reducing costs." Have the authors tested the cost of the system against others?

6/ Limitations must include the specific characteristics of the databases used (eg, comorbidities) and discussion should take into account that SHHS dataset are indeed AT HOME recordings.

Minor issues:

- There are repeated references in the list with different numbers

(Remarks on code availability)

Reviewer #2

(Remarks to the Author)

Manuscript ID: NCOMMS-24-73977-T

Title: Transparent Artificial Intelligence – enabled Interpretable and Interactive Sleep Apnea Assessment across Flexible Monitoring Scenarios

Comments to the Author:

Hu, Liu, and their collaborators...

The authors have made a significant contribution by integrating multiple modalities for an assessment of sleep apnea (SA) and proposing an interpretable and interactive model, the so-called Apnea Interact Xplainer (AIX). They reported that AIX achieved a sensitivity of 0.949 for early SA detection using only oximetry signal, and provides nocturnal risk assessment and intelligent monitoring reports. This represents a significant advance in early and cost-effective SA diagnosis through transparent AI.

This study seems interesting, and it is commendable that they used 15,510 polysomnographic records from multi-ethnic databases. However, I have the following major concerns. There are several limitations that need to be addressed appropriately.

1. First of all, what was the rationale for choosing SpO2 as a single modality among the various indices? While total apnea-hypopnea index (AHI) is considered the gold standard most commonly used to diagnose sleep apnea (SA), SpO2 is also widely used as an important marker. However, given that SpO2 alone is often highly correlated with SA, the rationale for choosing SpO2 as the primary single modality input requires further clarification. Why was SpO2 prioritized over other modalities for this evaluation? Oximetry measurements are undoubtedly valuable, but their limitations in distinguishing complex respiratory events might warrant further discussion.

2. In the abstract, the authors mentioned that AIX achieved 99.8% accuracy within a single severity grade. However, the confusion matrix in Figure 4(a) shows that the accuracy for classifying into four severity levels (normal, mild, moderate, and severe) ranges from 57% to 88%. A more meaningful task in the real world would be to predict the severity level rather than a binary classification whether subjects have SA or not. The discrepancy should be addressed as SA management heavily depends on accurate severity classification to consider specific treatment methods, and the lack of clear focus on multiclass outcomes could limit its clinical applicability.

3. As a related question, the authors did not specify whether AIX was tested in real-world scenarios, such as home monitoring settings where signal quality and patient compliance can vary significantly. Validation in such settings needs to be addressed to provide critical insights into its robustness.

4. Most importantly, what was the rationale for using LightGBM, a gradient-boosting decision tree algorithm, instead of other deep learning approaches that are becoming increasingly viable in medical AI? In fact, the authors evaluated six machine learning algorithms and ultimately chose LightGBM as their final model. While the choice appears justified based on performance metrics, the manuscript does not address why they did not consider or apply deep learning approaches. Although the authors emphasize “transparent AI” in the abstract, it remains unclear why they chose this particular framework instead of incorporating explainable deep learning techniques such as convolutional neural networks (CNNs) or recurrent neural networks (RNNs). LightGBM is computationally efficient, but appears to be less powerful in capturing complex patterns than modern deep learning models such as CNNs or RNNs, which can more effectively explore temporal and spatial patterns in the data. Given the growing popularity of deep learning in SA evaluation, it is important to discuss the decision to exclude deep learning methods. Incorporating advanced models, such as time-series data transformers or hybrid deep learning approaches could improve prediction accuracy and generalization, and can be particularly helpful in capturing subtle patterns in sleep-related physiological signals.

5. The model incorporates multiple modalities, including Flow, Chest, and SpO2 as gold standards. However, the specific features derived from each modality are not clearly described in the manuscript. Providing a detailed description of the exact feature values utilized in each modality would greatly enhance the transparency and reproducibility of the study.

6. This paper emphasizes interpretability through its ‘transparent scale diffusion mechanisms’. The claim of multi-level expert-logic interpretable visualization is compelling, but the authors did not elaborate how this interpretable framework has been evaluated or validated in the manuscript. It lacks the use of widely recognized explainable AI methods such as Grad-CAM. In fact, techniques like Grad-CAM would provide visualizations of attention over input signals, making it easier for clinicians to understand why specific parts of the data influenced the model’s precisions. Although the authors claimed to provide traceable event-level logic, they did not sufficiently explain how this was achieved. The methodology for generating granular explanation still remains vague. More clear description of how event-level decisions (e.g. apnea detection in a 30-sec window) are derived along with concrete examples or visualizations, is necessary to substantiate this claim. How was “trust” in the AI system measured among end-users such as clinicians? In the manuscript, it should be addressed properly about demonstrating use-centric outcomes beyond performance metrics. This would enhance the system’s transparency and clinical implications.

7. The authors also claim that the system is multimodal because it integrates Flow, Chest, and SpO2 signals. However, these signals are all subsets of the same physiological modality (e.g. respiratory monitoring). Could it be possible to add multimodal, which generally refers to integrating heterogeneous data types, such as text, images, or audio? Since these kinds of multimodal datasets only use physiological signals, the system lacks the diversity needed to fully justify the multimodal claim. To concretely establish the multimodality argument, the utility of the model would need to be enhanced by adding tabular data (e.g. patient demographics, medical history) and/or imaging data (e.g. EEG figures or waveform patterns).

8. The authors did not utilize state-of-the-art multimodal learning methods. Approaches such as CLIP or SimCLR, which learn shared representations across multiple modalities, can be applied to align different types of physiological signals with patient information or images.

9. Some of the data should be double-checked. In Fig 3(c), for example, the error bar of BMI in CFS exceeds 80, which is not realistic. Please check the numbers again.

Overall, the selection of the model and SpO2 as a single modality should be important to clarify the position of the model in the broad field of AI, especially in healthcare. In addition to PSG measurements, additional deep learning methods such as CNN/RNN with multiple modalities should be considered. Approaches that pay more attention to clinical evidence, scalability, and user-centric evaluation may improve the utility and relevance of the work in practice.

(Remarks on code availability)

Version 1:

Reviewer comments:

Reviewer #1

(Remarks to the Author)

Thank you so much for the effort in the response. Despite the clear improvement in the manuscript, I still have some major comments.

1) Please, include the results in the databases different from SHHS of the model trained with non-overlap subjects of SHHS1.

2) Regarding the previous point, as expected, SHHS2 results significantly decrease when using the model trained with non-overlap SHHS1 subjects. Accordingly, the SHHS2 main results to be reported should be those from this model, and not those from the current model. At least, those currently reported results for SHHS2 are not valid because they are clearly biased.

3) Using 1 Hz for respiratory signals (airflow, chest, etc) is far below the 10 Hz established as a minimum for the American Academy of Sleep Medicine. There exist deep learning approaches that do not need the use of the same sampling frequency in different signals (eg, stacking different CNN branches). Accordingly, it is my opinion that to prove you are not losing significant information you should compare your current results with this approach, or a similar one using appropriate sampling frequencies of at least 1 Hz for SpO2 and 4 Hz (some papers did it in the past) for respiratory signals.

(Remarks on code availability)

Reviewer #3

(Remarks to the Author)

This revision and the answers provide important additions to the initial manuscript. The subject of addressing interactive sleep apnea assessment is of high importance in view of the high prevalence of the disorder.

To introduce transparent and interpretable AI is critical in view of the current discussion on using AI. The supplemental information is definitely needed in order to understand the details for a science reader.

Taken together, the work will be of high significance to the field of sleep medicine, sleep research, and consumer health. Flaws in the data analysis or interpretation can not be identified.

The supplemental material is rich and needed in order to allow the methods to be reproduced.

(Remarks on code availability)

Version 2:

Reviewer comments:

Reviewer #1

(Remarks to the Author)

The authors have satisfied all my concerns. Great job.

(Remarks on code availability)

Dec 18, 2024

Manuscript ID: NCOMMS-24-73977-T

Title: Transparent Artificial Intelligence – enabled Interpretable and Interactive Sleep Apnea Assessment across Flexible Monitoring Scenarios

Comments to the Author:

Hu, Liu, and their collaborators...

The authors have made a significant contribution by integrating multiple modalities for an assessment of sleep apnea (SA) and proposing an interpretable and interactive model, the so-called Apnea Interact Xplainer (AIX). They reported that AIX achieved a sensitivity of 0.949 for early SA detection using only oximetry signal, and provides nocturnal risk assessment and intelligent monitoring reports. This represents a significant advance in early and cost-effective SA diagnosis through transparent AI.

This study seems interesting, and it is commendable that they used 15,510 polysomnographic records from multi-ethnic databases. However, I have the following major concerns: There are several limitations that need to be addressed appropriately.

1. First of all, what was the rationale for choosing SpO₂ as a single modality among the various indices? While total apnea-hypopnea index (AHI) is considered the gold standard most commonly used to diagnose sleep apnea (SA), SpO₂ is also widely used as an important marker. However, given that SpO₂ alone is often highly correlated with SA, the rationale for choosing SpO₂ as the primary single modality input requires further clarification. Why was SpO₂ prioritized over other modalities for this evaluation? Oximetry measurements are undoubtedly valuable, but their limitations in distinguishing complex respiratory events might warrant further discussion.

2. In the abstract, the authors mentioned that AIX achieved 99.8% accuracy within a single severity grade. However, the confusion matrix in Figure 4(a) shows that the accuracy for classifying into four severity levels (normal, mild, moderate, and severe) ranges from 57% to 88%. A more meaningful task in the real world would be to predict the severity level rather than a binary classification whether subjects have SA or not. The discrepancy should be addressed as SA management heavily depends on accurate severity classification to consider specific treatment methods, and the lack of clear focus on multiclass outcomes could limit its clinical applicability.

3. As a related question, the authors did not specify whether AIX was tested in real-world scenarios, such as home monitoring settings where signal quality and patient compliance can vary significantly. Validation in such settings needs to be addressed to provide critical insights into its robustness.

4. Most importantly, what was the rationale for using LightGBM, a gradient-boosting decision tree algorithm, instead of other deep learning approaches that are becoming increasingly viable in medical AI? In fact, the authors evaluated six machine learning algorithms and ultimately chose LightGBM as their final model. While the choice appears justified based on performance metrics, the manuscript does not address why they did not consider or apply deep learning approaches. Although the authors emphasize “transparent AI” in the abstract, it remains unclear why they chose this particular framework instead of incorporating explainable deep learning techniques such as convolutional neural networks (CNNs) or recurrent neural networks (RNNs). LightGBM is computationally efficient, but appears to be less powerful in capturing complex patterns than modern deep learning models such as CNNs or RNNs, which can more effectively explore temporal and spatial patterns in the data. Given the growing popularity of deep learning in SA evaluation, it is important to discuss the decision to exclude deep learning methods. Incorporating advanced models, such as time-series data transformers or hybrid deep learning approaches could improve prediction accuracy and generalization, and can be particularly helpful in capturing subtle patterns in sleep-related physiological signals.

5. The model incorporates multiple modalities, including Flow, Chest, and SpO2 as gold standards. However, the specific features derived from each modality are not clearly described in the manuscript. Providing a detailed description of the exact feature values utilized in each modality would greatly enhance the transparency and reproducibility of the study.

6. This paper emphasizes interpretability through its ‘transparent scale diffusion mechanisms’. The claim of multi-level expert-logic interpretable visualization is compelling, but the authors did not elaborate how this interpretable framework has been evaluated or validated in the manuscript. It lacks the use of widely recognized explainable AI methods such as Grad-CAM. In fact, techniques like Grad-CAM would provide visualizations of attention over input signals, making it easier for clinicians to understand why specific parts of the data influenced the model’s precisions. Although the authors claimed to provide

traceable event-level logic, they did not sufficiently explain how this was achieved. The methodology for generating granular explanation still remains vague. More clear description of how event-level decisions (e.g. apnea detection in a 30-sec window) are derived along with concrete examples or visualizations, is necessary to substantiate this claim. How was “trust” in the AI system measured among end-users such as clinicians? In the manuscript, it should be addressed properly about demonstrating use-centric outcomes beyond performance metrics. This would enhance the system’s transparency and clinical implications.

7. The authors also claim that the system is multimodal because it integrates Flow, Chest, and SpO2 signals. However, these signals are all subsets of the same physiological modality (e.g. respiratory monitoring). Could it be possible to add multimodal, which generally refers to integrating heterogeneous data types, such as text, images, or audio? Since these kinds of multimodal datasets only use physiological signals, the system lacks the diversity needed to fully justify the multimodal claim. To concretely establish the multimodality argument, the utility of the model would need to be enhanced by adding tabular data (e.g. patient demographics, medical history) and/or imaging data (e.g. EEG figures or waveform patterns).

8. The authors did not utilize state-of-the-art multimodal learning methods. Approaches such as CLIP or SimCLR, which learn shared representations across multiple modalities, can be applied to align different types of physiological signals with patient information or images.

9. Some of the data should be double-checked. In Fig 3(c), for example, the error bar of BMI in CFS exceeds 80, which is not realistic. Please check the numbers again.

Overall, the selection of the model and SpO2 as a single modality should be important to clarify the position of the model in the broad field of AI, especially in healthcare. In addition to PSG measurements, additional deep learning methods such as CNN/RNN with multiple modalities should be considered. Approaches that pay more attention to clinical evidence, scalability, and user-centric evaluation may improve the utility and relevance of the work in practice.

Transparent Artificial Intelligence-enabled Interpretable and Interactive Sleep Apnea Assessment across Flexible Monitoring Scenarios

Shuaicong Hu, Jian Liu, Yanan Wang, Cong Fu, Jichu Zhu, Huan Yu, and Cuiwei Yang

Dear Editor and Reviewers,

We sincerely thank you for your comprehensive evaluation of our manuscript '*Transparent Artificial Intelligence-enabled Interpretable and Interactive Sleep Apnea Assessment across Flexible Monitoring Scenarios*' (NCOMMS-24-73977-T). Your professional and constructive comments have greatly improved our work. We have carefully revised the manuscript to address all concerns, with detailed responses and new analyses documented in a 52-page Supplementary Materials file and corresponding additions to the main text.

The major revisions include:

1) Technical Validation

- Unified resampling approach across varied sampling rates.
- Longitudinal analysis using SHHS1/SHHS2 overlapping subjects ($n=2,647$).
- Systematic hyperparameter optimization through Bayesian trials.
- Standardized scoring criteria using AASM 2012 guidelines.

2) Model Interpretability & Clinical Utility

- Quantitative comparison between TSD-Net and Grad-CAM.
- SHAP-based analysis of physiological signal importance.
- Real-world validation in home monitoring scenarios.

3) Advanced Modeling & Implementation

- Integration of demographic features for AHI regression.
- Novel CLIP-inspired framework for multi-modal representation.
- Comprehensive cost-effectiveness analysis for clinical deployment.

The Methods section has been substantially enhanced with detailed documentation of our approach, and the Discussion now provides thorough comparisons with existing methods.

We believe these revisions have significantly strengthened our manuscript's scientific contribution and clinical relevance. We remain at your disposal for any further clarifications needed.

Sincerely,

First Author Shuaicong Hu & Corresponding Author Cuiwei Yang

RESPONSE TO THE REVIEW OF NCOMMS-24-73977-T

The changes have been marked in red in the revised manuscript. Reviewer's comments are shown in bold, with our responses in italics immediately below each. Revised content in the text is indicated in blue in our responses.

Responses to Reviewer #1:

This study aims to develop and evaluate a new system for the automatic detection of sleep apnea. Three overnight signals are utilized for this purpose, although different single-channel configurations can also be employed. Two concatenated deep learning techniques are applied to estimate the apnea-hypopnea index (AHI), which is the clinical variable used to diagnose sleep apnea. Additionally, explainable artificial intelligence (XAI) is incorporated to enhance the reliability of the system. Concordance and diagnostic metrics are high, addressing a significant problem. The objectives of the study are relevant, and the narrative is clear. However, I have identified some major and minor concerns regarding the study, which I detail below:

Response: We are deeply grateful to the reviewer for providing such a thorough and insightful evaluation of our manuscript. We particularly appreciate the reviewer's recognition of our study's relevance, clear narrative, and the importance of developing automatic sleep apnea detection systems. The reviewer's comprehensive comments have been invaluable in helping us improve our work. We have carefully considered each point raised and provided detailed point-by-point responses below.

Major issues:

- 1. Please, clearly specify and explain how you dealt with the different sample rates of each signal, including the final sample rate used for each of them and a justification for that election when needed.**

Response: We appreciate the reviewer's attention to this important methodological detail. We carefully addressed the sampling rate variations across different signals and databases through a standardized resampling approach.

The original sampling rates show considerable variation across databases. For Flow and Chest signals, SHHS dataset predominantly uses 8 Hz or 10 Hz, MESA dataset uses 32 Hz, MROS dataset uses 16 Hz or 32 Hz, and CFS dataset uses 32 Hz. In contrast, the SpO₂ signal maintains a consistent 1 Hz sampling rate across all databases.

According to established physiological studies, normal human respiratory rates during sleep typically range from 12 to 20 breaths per minute, corresponding to frequencies of 0.2-0.33 Hz¹. Even during respiratory events, the fundamental frequency of breathing patterns rarely exceeds 0.5 Hz². Our resampling approach (256 points/270 seconds \approx 0.95 Hz) satisfies the Nyquist sampling theorem, which requires a sampling frequency of at least twice the highest frequency component of interest. For SpO₂ signals, the 0.95 Hz sampling rate is also sufficient since blood oxygen changes occur more gradually than respiratory variations.

To handle these variations, we implemented a unified approach where 270-second segments of physiological signals were resampled to a fixed length of 256 points using the resample function from Python's SciPy library. This standardization was crucial given the substantial sampling rate variations across databases.

We chose this approach for several key reasons. The 256-point length adequately captures the physiological patterns while maintaining essential frequency components for respiratory event detection. This power-of-2 length optimizes computational performance during model training and inference. Most importantly, it effectively handles the significant sampling rate variations across different databases, ensuring consistent input dimensions for our deep learning model.

The effectiveness of this resampling strategy is validated by our model's consistent performance across multiple external validation datasets with different original sampling rates. This suggests our approach successfully preserves the discriminative features necessary for accurate sleep apnea detection, regardless of the source database's sampling rate.

Figure 1 | Visual comparison of original signals (at different sampling rates) and their resampled versions, showing preservation of key respiratory events and waveform characteristics.

As illustrated in the Figure 1, we visualize two representative cases (32 Hz and 10 Hz) of the original signals and their resampled versions (256/270 Hz). The comparison demonstrates that our resampling approach effectively preserves the key characteristics of all three physiological signals. In both high-frequency (32 Hz) and lower-frequency (10 Hz) scenarios, the resampled signals maintain their fundamental waveform patterns and respiratory event features. The Flow and Chest signals clearly retain their respiratory rhythms and amplitude variations, while the SpO₂ signal preserves its gradual transitions. This visual evidence supports that our resampling strategy successfully standardizes the signal dimensions while maintaining the essential physiological information needed for sleep apnea detection. We have substantially revised the Methods Section 'Signal Quality Control and Data Preprocessing' on Page 16 to address this concern.

In Signal Quality Control and Data Preprocessing (Page 16):

Due to the complexity of sleep monitoring and the susceptibility to interference signals, there are factors that disrupt the accurate assessment of AHI in patients with SA. Firstly, there are artifact signals in SpO₂ levels caused by loose sensor attachment or excessive movement. We exclude the corresponding data intervals to provide a more precise AHI evaluation.

Secondly, before and after formal sleep testing, patients may improperly wear airflow sensors or thoracic bands, resulting in recordings that display useless, nearly flat-line data. Therefore, we calculate the standard deviation for segments of these two types of signals and use 0.0001 as the exclusion threshold. Any signal segments with a standard deviation below this threshold are excluded. To ensure reliable sleep evaluation, records where the non-wake duration combined with the duration of valid signal collection is less than 4 hours are removed. This data cleaning process aids in more accurately assessing the true AHI in patients with SA.

The sampling rates of physiological signals vary considerably across different sleep databases. Specifically, the airflow and thoracic signals are sampled at 32 Hz in MESA and CFS databases, 16 Hz or 32 Hz in MROS database, and predominantly 8 Hz or 10 Hz in SHHS database. The SpO₂ signals maintain a consistent sampling rate of 1 Hz across all databases. To address these variations, we implement a standardized resampling approach that converts all 270-second segments to a fixed length of 256 points. This resampling strategy is theoretically sound as normal human respiratory rates during sleep typically range from 12 to 20 breaths per minute (0.2-0.33 Hz)¹, and even during respiratory events, the fundamental frequency rarely exceeds 0.5 Hz². Our resampling frequency (256/270 ≈ 0.95 Hz) satisfies the Nyquist sampling theorem, ensuring adequate capture of respiratory event characteristics. For SpO₂ signals, this sampling rate is also sufficient given the relatively gradual nature of blood oxygen variations. The effectiveness of this approach is validated by our model's consistent performance across multiple external validation datasets with different original sampling rates, demonstrating successful preservation of discriminative features necessary for accurate SA detection.

[1] Priban, I. An analysis of some short-term patterns of breathing in man at rest. *The Journal of physiology* **166**, 425 (1963).

[2] Chon, K.H., Dash, S. & Ju, K. Estimation of respiratory rate from photoplethysmogram data using time–frequency spectral estimation. *IEEE Transactions on Biomedical Engineering* **56**, 2054-2063 (2009).

-
- 2. There is a clear bias in using all SHHS1 to train the model as more than 2,000 subjects are the same in SHHS2 test set (five years later). I think the study will be less biased if the subjects in SHHS1 that are also in SHHS2 are used as a different test set. This will be particularly useful to make the cross-population stability robust. In addition, it will allow a longitudinal analysis in terms of performance and explainability.**

Response: *Thank you for this insightful suggestion regarding the potential bias in using SHHS1 for training when some subjects overlap with SHHS2. We agree that this warrants careful analysis.*

Following your recommendation, we conducted an additional longitudinal analysis using the 2,647 subjects who appear in both SHHS1 and SHHS2 cohorts. We focused on the SpO₂-only channel configuration due to its demonstrated robustness, clinical accessibility, and practical significance for

widespread screening applications.

The results from this longitudinal validation are particularly informative:

1) When trained on the non-overlapping subjects from SHHS1 ($n=3,146$) and tested on the overlapping subjects' SHHS1 records ($n=2,466$), the model achieved strong performance ($ACC=0.785$, Macro $F1=0.791$) for four-level severity classification, with $R^2=0.9491$ and $ICC=0.9477$ for AHI prediction ($MAE=3.20\pm 3.42$).

2) More importantly, when testing on these same subjects' SHHS2 records ($n=2,518$, collected approximately 5 years later), the model maintained robust performance ($ACC=0.778$, Macro $F1=0.783$, $MAE=3.28\pm 3.62$), with $R^2=0.9535$ and $ICC=0.9524$, demonstrating excellent temporal stability.

3) The confusion matrices show consistent classification patterns between time points, particularly for severe cases (SHHS1: $SEN=0.888$, SHHS2: $SEN=0.910$), and Bland-Altman analysis reveals stable mean differences (SHHS1: -0.70 , SHHS2: -0.71) with comparable 95% limits of agreement.

We have added these results to Supplementary Fig. 28, which includes detailed severity-specific performance metrics. For instance, in SHHS1, the model achieved balanced performance across severity levels (Healthy: $SEN=0.685$, $SPE=0.971$; Mild: $SEN=0.813$, $SPE=0.837$; Moderate: $SEN=0.758$, $SPE=0.909$; Severe: $SEN=0.888$, $SPE=0.970$), with similar patterns maintained in SHHS2. This longitudinal analysis provides additional evidence for the robustness of our approach across time and strengthens the cross-population stability assessment.

We appreciate this suggestion as it has helped demonstrate the model's capability to track SA progression in the same individuals over time, an important clinical consideration not captured in our original cross-sectional analysis. The detailed description of this content is provided in the Results section 'Cross-Population Stability Assessment of AIX' on Page 8 and Supplementary Materials.

In Cross-Population Stability Assessment of AIX (Page 8):

Moreover, to assess temporal stability and potential training bias, we conduct a longitudinal analysis using 2,647 subjects who appear in both SHHS1 and SHHS2 cohorts (approximately 5 years apart). We specifically focus on the SpO₂-only configuration given its optimal balance between performance and accessibility. The model trained on non-overlapping SHHS1 subjects ($n=3,146$) demonstrates remarkable temporal stability when tested on the overlapping subjects at both time points (Supplementary Figs. 27,28 and Supplementary Table 12). For SHHS1 data ($n=2,466$), it achieves $ACC=0.785$, Macro $F1=0.791$, with $R^2=0.9491$ and $ICC=0.9477$ ($MAE=3.20\pm 3.42$) for AHI prediction. The performance remains consistent when tested on SHHS2 data ($n=2,518$, $ACC=0.778$, Macro $F1=0.783$, $R^2=0.9535$, $ICC=0.9524$, $MAE=3.28\pm 3.62$), with confusion matrices showing stable classification patterns particularly for severe cases (SHHS1: $SEN=0.888$, SHHS2: $SEN=0.910$). Bland-Altman analysis further confirms the model's temporal stability with comparable mean differences and limits of agreement between time points. This temporal robustness suggests the model's potential for reliable longitudinal monitoring of SA progression.

In Supplementary Material:

SHHS-1 train cohort n = 2789	SHHS-1 test cohort 1 n = 2466	SHHS-2 test cohort 2 n = 2518
Gender	Gender	Gender
Male: 1371 Female: 1418	Male: 1138 Female: 1328	Male: 1156 Female: 1362
Age	Age	Age
Mean: 63.3 Med: 63.0 High: 90.0 Low: 39.0	Mean: 62.1 Med: 62.0 High: 90.0 Low: 44.0	Mean: 67.3 Med: 67.0 High: 90.0 Low: 44.0
BMI	BMI	BMI
Mean: 28.1 Med: 27.4 High: 50.0 Low: 18.0	Mean: 28.2 Med: 27.6 High: 50.0 Low: 18.0	Mean: 28.4 Med: 27.8 High: 50.0 Low: 18.0
AHI	AHI	AHI
Mean: 18.7 Med: 13.3 High: 121.3 Low: 0.0	Mean: 16.8 Med: 12.6 High: 126.0 Low: 0.2	Mean: 18.3 Med: 13.6 High: 110.3 Low: 0.0
Sleep duration (h)	Sleep duration (h)	Sleep duration (h)
Mean: 6.05	Mean: 6.25	Mean: 6.37

Supplementary Fig. 27 | Data partition scheme and cohort characteristics for temporal stability analysis. (Top) Flow diagram illustrating the data partition strategy for SHHS-1 (n=5,793) and SHHS-2 (n=2,651) cohorts. SHHS-1 was split into training (n=2,231), validation (n=558), and test cohort 1 (n=2,466) after exclusion criteria. SHHS-2 formed test cohort 2 (n=2,518) after applying exclusion criteria. The overlapping subjects between SHHS-1 and SHHS-2 (n=2,647) were used for temporal stability assessment. (Bottom) Demographic and clinical characteristics of the training cohort, test cohort 1, and test cohort 2, including gender distribution, age, BMI, AHI, and sleep duration statistics. The violin plots display the age, BMI, and AHI distributions across the three cohorts.

Supplementary Fig. 28 | Temporal performance comparison between SHHS1 and SHHS2 cohorts. (a) Model performance on SHHS1 test cohort ($n=2,466$). Left: Confusion matrix showing the classification performance across four severity categories (numbers indicate absolute counts and percentages). Middle: Scatter plot of predicted versus true AHI values with $R\text{-squared}=0.9491$ ($P<0.0001$). Right: Bland-Altman plot showing the agreement between predicted and true AHI values (mean difference= -0.70 , 95% limits of agreement: -9.78 to $+8.38$). (b) Model performance on SHHS2 test cohort ($n=2,518$). Left: Confusion matrix for severity classification. Middle: AHI prediction scatter plot with $R\text{-squared}=0.9535$ ($P<0.0001$). Right: Bland-Altman analysis with mean difference= -0.71 and 95% limits of agreement from -10.19 to $+8.76$.

Supplementary Table 12 | Performance metrics comparison between SHHS1 and SHHS2 cohorts. Detailed performance metrics (SEN, SPE, PPV, F1, and MAE) across four severity categories (Healthy, Mild, Moderate, Severe) and overall performance for SHHS1 ($n=2,466$) and SHHS2 ($n=2,518$) test cohorts. Values are presented as mean \pm standard deviation for MAE.

Dataset	Patient distribution	Performance				
		SEN	SPE	PPV	F1	MAE
SHHS1 ($n=2,466$)	Healthy ($n=463$)	0.685	0.971	0.845	0.757	1.81 \pm 1.58
	Mild ($n=963$)	0.813	0.837	0.762	0.787	2.43 \pm 2.40
	Moderate ($n=665$)	0.758	0.909	0.756	0.757	3.73 \pm 3.53
	Severe ($n=375$)	0.888	0.970	0.841	0.864	5.96 \pm 5.00
	Overall ($n=2,466$)	ACC=0.785 F1 _{Macro} =0.791				3.20 \pm 3.42
SHHS2 ($n=2,518$)	Healthy ($n=436$)	0.677	0.965	0.802	0.734	2.02 \pm 2.00
	Mild ($n=941$)	0.778	0.850	0.755	0.766	2.66 \pm 3.07
	Moderate ($n=684$)	0.754	0.903	0.745	0.749	3.49 \pm 3.03
	Severe ($n=457$)	0.910	0.965	0.852	0.880	5.46 \pm 5.29
	Overall ($n=2,518$)	ACC=0.778 F1 _{Macro} =0.783				3.28 \pm 3.62

3. Please specify and clearly explain and justify how the hyperparameters of the deep learning models were chosen. Apparently, there is not a validation group for this specific purpose, which introduces a potential bias.

***Response:** Thank you for your insightful comment. We apologize for not clearly describing our hyperparameter optimization strategy in the original manuscript. We have conducted comprehensive hyperparameter tuning experiments for the TSD-Net (Model 1) and adopted a validated architecture design for the AHI regression model (Model 2), which were inadvertently omitted from the manuscript.*

For the TSD-Net (Model 1), we employed Bayesian optimization with 100 trials to explore the following key parameters:

- 1) MLP expansion ratio (mlp_ratio): [2-8]*
- 2) Channel expansion ratio (expand_ratio): [1-4]*
- 3) Dropout rate (drop_rate): [0.0-0.5]*
- 4) Attention dropout rate (attn_drop_rate): [0.0-0.5]*
- 5) Path dropout rate (drop_path_rate): [0.0-0.3]*
- 6) Head convolution kernel size (head_conv): [3, 5, 7, 9]*

For each trial, we used 80% of the training data for model training and the remaining 20% as a validation set to evaluate the model's performance. The optimal hyperparameters were selected based on the validation loss. The optimization process revealed that mlp_ratio converged to 2, indicating moderate feature transformation is sufficient for capturing temporal patterns in physiological signals. Notably, dropout-related parameters approached zero, suggesting our progressive attention design provides adequate inherent regularization. The expand_ratio stabilized at 2, achieving an optimal balance between feature representation and computational efficiency.

For other architectural hyperparameters of TSD-Net, such as the number of blocks in each stage (2, 2, 8, 1), the number of convolutional attention heads (4, 4, 4, 1), and the number of Transformer attention heads (-1, -1, 8, 16), we adopted the recommended values from previous literature^{3,4}, which have been extensively validated in similar temporal signal processing tasks.

For the AHI regression model (Model 2), we adopted a proven lightweight architecture with three convolutional layers using decreasing kernel sizes (5→3→1) and increasing filter sizes (16→32→64), followed by two fully connected layers (8×1024→128→1). This architecture was selected based on established principles in temporal feature extraction⁵. The progressive reduction in kernel sizes and increase in filter dimensions enables efficient multi-scale feature extraction while maintaining model simplicity.

We have added detailed visualization of the hyperparameter optimization process in Supplementary Fig. 29, which includes the optimization history, learning curves of the best trial, relative importance of different hyperparameters, and their pairwise relationships:

Supplementary Fig. 29 | Visualization of hyperparameter optimization process. (Top row) Left: Evolution of validation loss across 100 Bayesian optimization trials; Middle: Learning curves of the best trial showing training and validation losses; Right: Relative importance of optimized hyperparameters. (Bottom rows) Pairwise relationship plots between different hyperparameters, where point colors indicate validation loss values (darker red indicates higher loss, darker blue indicates lower loss). The `expand_ratio` and `mlp_ratio` demonstrated the strongest influence on model performance.

The Methods section 'Model Training' on Page 19 is enhanced to include the validation split strategy and parameter selection process. Additionally, a new subsection titled 'Hyperparameter Optimization' in the

Methods section on Page 20 describes:

In Methods Section ‘Model Training’ (Page 19):

The key architectural hyperparameters of the model were optimized through Bayesian optimization over 100 trials. The training set was split into 80% for training and 20% for validation during the optimization process. Six critical parameters were explored: MLP expansion ratio (range: 2-8), channel expansion ratio (range: 1-4), dropout rate (range: 0.0-0.5), attention dropout rate (range: 0.0-0.5), path dropout rate (range: 0.0-0.3), and head convolution kernel size (options: 3, 5, 7, 9). Other architectural parameters, such as the number of blocks in each stage (2, 2, 8, 1), convolutional attention heads (4, 4, 4, 1), and Transformer attention heads (-1, -1, 8, 16), were adopted from validated configurations in previous studies^{3,4}. The parameter configuration achieving the lowest validation loss was selected for the subsequent experiments (Supplementary Fig. 29).

In Methods Section ‘Hyperparameter Optimization’ (Page 20):

The hyperparameter optimization of TSD-Net (Model 1) was conducted through Bayesian optimization over 100 trials, revealing the relative importance of different architectural components in 30-second granular respiratory event detection (Supplementary Fig. 29). The `mlp_ratio`, controlling the hidden feature expansion in feed-forward networks, showed the highest importance and converged to 2. This moderate expansion suggests that the temporal patterns in physiological signals can be effectively captured without requiring extensive feature transformation capacity.

Dropout-related parameters (`drop_rate`, `attention_drop_rate`, and `path_drop_rate`) ranked as the second most important group but approached zero, indicating our four-stage progressive attention design provides sufficient inherent regularization. The `expand_ratio`, which determines channel expansion in convolutional attention modules, stabilized at 2, achieving effective feature representation while maintaining computational efficiency.

The head convolution kernel size demonstrated relatively low importance and converged to 3. This small convolution window is sufficient for capturing local respiratory features in the initial embedding stage, preparing informative local representations for subsequent large-scale attention processing in deeper layers.

For the AHI regression model (Model 2), we adopted a proven architecture design with three convolutional layers using decreasing kernel sizes (5→3→1) followed by two fully connected layers⁵. This simple yet effective architecture helps prevent overfitting while enabling efficient multi-scale temporal feature extraction. The lightweight network structure with progressively reduced kernel sizes ensures both computational efficiency and robust prediction accuracy for clinical deployment.

This systematic approach to hyperparameter optimization helps ensure the model's robustness and reduces potential bias in the experimental results.

- [3] Liu, Z., et al. Swin transformer: Hierarchical vision transformer using shifted windows. in *Proceedings of the IEEE/CVF international conference on computer vision* 10012-10022 (2021).
- [4] Lin, W., Wu, Z., Chen, J., Huang, J. & Jin, L. Scale-aware modulation meet transformer. in *Proceedings of the IEEE/CVF International Conference on Computer Vision* 6015-6026 (2023).
- [5] Wang, T., Lu, C., Shen, G. & Hong, F. Sleep apnea detection from a single-lead ECG signal with automatic feature-extraction through a modified LeNet-5 convolutional neural network. *Peerj* 7(2019).

4. Databases were formed at different times and under different apneic events annotation rules. please clarify how the authors dealt with this problem.

Response: We sincerely appreciate your important question regarding the standardization of scoring rules across databases from different time periods. We addressed this potential issue through the following approaches:

For historical databases (SHHS, CFS, MROS, and MESA), respiratory events were initially scored without preset criteria for desaturation or arousal. Instead, sleep scoring software (such as Compumedics Profusion) was used to preliminarily identify apnea and hypopnea events based on airflow drops lasting more than 10 seconds⁶. The scoring system then linked these events to oxygen saturation and EEG data, enabling researchers to retrospectively apply different standardization criteria to generate AHI variables. To ensure scoring consistency, we specifically selected the AHI variable (code: ahi_a0h3a) that complies with AASM 2012 recommended rules from these databases⁷. This has been confirmed through direct communication with NSRR database administrators in previous cross-database research⁸.

For our recently collected Asian cohort (FDU-HSH database collected after 2012), events were directly scored following AASM 2012 guidelines. All scorings were performed by certified sleep technicians. This standardized approach enabled us to maintain consistent definitions of apnea and hypopnea events, ensuring valid comparisons of AHI measurements across all databases regardless of their original collection period.

We have added these details to the Methods section 'Large-Scale and Multi-Ethnic Cohort Description' on Pages 14-15 to provide complete methodological clarity.

In Large-Scale and Multi-Ethnic Cohort Description (Pages 14-15):

For historical databases (SHHS, CFS, MROS, and MESA), respiratory events are initially scored without preset criteria for desaturation or arousal. Instead, sleep scoring software (such as Compumedics Profusion) preliminarily identifies apnea and hypopnea events based on airflow drops lasting more than 10 seconds⁶. The scoring system then links these events to oxygen saturation and EEG data, enabling researchers to retrospectively apply different standardization criteria to generate AHI variables. To ensure scoring consistency, we specifically select the AHI variable (code: ahi_a0h3a) that complies with AASM 2012 recommended rules from these databases⁷. This has been confirmed through direct communication with NSRR database administrators in previous cross-database research⁸.

For our recently collected Asian cohort (FDU-HSH database collected after 2012), events are directly scored following AASM 2012 guidelines, with all scoring performed by certified sleep technicians. This standardized approach enables us to maintain consistent definitions of apnea and hypopnea events, ensuring valid comparisons of AHI measurements across all databases regardless of their original collection period.

[6] H. Rapoport David M. Smith Philip L. Kiley James P., S.H.H.R.G.R.S.s.p.c.e.S.M.H.L.B.K.Q.S.F.I.C.G.D.J.B.W. Methods for obtaining and analyzing unattended polysomnography data for a multicenter study. *Sleep* **21**, 759-767 (1998).

[7] Berry, R.B., *et al.* Rules for Scoring Respiratory Events in Sleep: Update of the 2007 AASM Manual for the Scoring of Sleep and Associated Events. *Journal of Clinical Sleep Medicine* **8**, 597-619 (2012).

[8] Levy, J., Álvarez, D., Del Campo, F. & Behar, J.A. Deep learning for obstructive sleep apnea diagnosis based on single channel oximetry. *Nature Communications* **14**, 4881 (2023).

5. Scientific discussion is scarce. There is a lack of scientific discussion as for example comparison with other similar approaches (there are a lot of them), or hyperparameter discussion etc. Moreover some statements are not properly justified. Example "The low operational threshold and flexible monitoring scenario adaptation of AIX significantly advance SA diagnosis popularization while reducing costs." Have the authors tested the cost of the system against others?

Response: We thank the reviewer for these valuable comments. We have addressed these concerns through substantial revisions:

1) Comparison with Similar Approaches: We have expanded our Discussion section to include a comprehensive review of current SA diagnosis approaches. This includes detailed analysis of ECG-based methods achieving 70-88% accuracy (Salari et al., Bahrami et al.), SpO₂-based approaches like OxiNet showing ICC of 0.92-0.96 (Levy et al.), and multi-channel systems such as DRIVEN with 72.4% classification accuracy. Our comparative analysis demonstrates that AIX achieves robust performance (ACC=0.74-0.81 for severity classification, ACC=0.896-0.939 for screening) while offering unique advantages in transparency and adaptability. We have made substantial revisions to the Discussion section on Page 10 to address this concern.

In Discussion (Page 10):

Previous research has explored various approaches for automated SA diagnosis. Early studies focused on ECG-based methods due to their potential for wearable applications. A comprehensive review by Salari et al. ⁹ showed that most machine learning and DL models achieved accuracies over 70% in detecting SA episodes from ECG segments. A systematic evaluation by Bahrami et al. ¹⁰ demonstrated that hybrid DL models could achieve 88.13% accuracy in SA detection using single-lead ECG. However, these approaches were primarily evaluated on the Apnea-ECG dataset ¹¹ with only 70 subjects, and were limited to binary classification tasks at segment-level events or patient-level SA diagnosis. While segment-level detection allows rough estimation of AHI values, this may not fully reflect the clinical severity assessment that considers additional physiological and temporal factors. Using SpO₂ signals, Levy et al. ⁸ developed an innovative DL model called OxiNet based on 12,923 PSG recordings from six independent databases. Their model achieved ICC values of 0.92-0.96 and F1 scores of 0.75-0.84 across multiple external test sets, demonstrating the potential of single-channel oximetry for SA diagnosis. Their work established strong foundations for future exploration of real-time respiratory event detection and model interpretability. Meanwhile, multi-channel approaches have demonstrated promising results in home settings. Wang et al. ¹² proposed a wearable smart ring device using PPG sensors to collect pulse wave signals and SpO₂ synchronously. Using multiscale entropy and random forest algorithms, they validated their system on 10 subjects and achieved accuracy of 85.99±2.26% in identifying SA episodes compared to PSG, though further validation on larger cohorts would strengthen these findings. Retamales et al. ¹³ recently introduced DRIVEN using 10,643 PSG recordings from MESA, MROS and SHHS databases, correctly classifying 72.4% of test patients into AHI severity classes using abdominal movement and SpO₂ sensors. Our recent work on IPCT-Net ¹⁴ leveraged 1603 HSAT recordings with a novel channel fusion framework. By comprehensively

analyzing Flow, Chest, SpO₂, pulse rate (PR) and snoring (Snore) signals, our model achieved R² of 0.939-0.959 and ICC of 0.937-0.958 in AHI estimation, while demonstrating robust performance in four-level SA severity classification (ACC=0.722-0.764, F1=0.677-0.717), showing comparable effectiveness between simplified and multi-channel combinations. Despite these encouraging advances, current approaches face inherent limitations in clinical adoption. The lack of transparency in model decisions makes it difficult for clinicians to validate diagnostic reasoning. Models often struggle to adapt across different monitoring configurations and equipment types. Additionally, quantitative evidence supporting optimal channel selection in clinical practice remains sparse. To address these challenges, we develop and validate AIX, a transparent explainable AI interactive framework that introduces expert-logic consistency in SA diagnosis for the first time. The system's robust performance in four-level SA severity classification (ACC=0.74-0.81) and binary screening (AHI cutoff at 5) across multiple external test sets (ACC=0.896-0.939 for Gold standard configuration, ACC=0.885-0.933 for SpO₂-based configuration) supports reliable clinical stratification while maintaining accessibility through simplified monitoring configurations.

2) Technical Implementation Details: We have enhanced the Methods section with comprehensive documentation of model architecture, training procedures, and hyperparameter selection criteria. Specifically, we conducted systematic hyperparameter optimization for our two-stage model architecture:

For the TSD-Net (Model 1), we employed Bayesian optimization over 100 trials to determine optimal architectural configurations. This rigorous optimization revealed several key insights about model design. The `mlp_ratio` parameter, which controls feature expansion in feed-forward networks, emerged as the most crucial factor and converged to 2, indicating that moderate feature transformation is sufficient for capturing temporal patterns in physiological signals. Notably, dropout-related parameters (`drop_rate`, `attention_drop_rate`, and `path_drop_rate`) approached zero, suggesting our four-stage progressive attention design provides adequate inherent regularization without requiring additional dropout mechanisms. The `expand_ratio` in convolutional attention modules stabilized at 2, achieving an optimal balance between feature representation and computational efficiency. The head convolution kernel size showed relatively low importance and converged to 3, demonstrating that a small initial convolution window effectively captures local respiratory features for subsequent attention processing.

For the AHI regression model (Model 2), we adopted a proven lightweight architecture based on established temporal feature extraction principles⁵. The model employs three convolutional layers with decreasing kernel sizes (5→3→1) followed by two fully connected layers. This architecture design, validated in previous temporal signal processing studies, enables efficient multi-scale feature extraction while preventing overfitting through its simple structure. The progressive reduction in kernel sizes ensures comprehensive temporal feature capture while maintaining computational efficiency for clinical deployment.

These architectural choices were made with careful consideration of both model performance and practical deployment requirements, resulting in a system that achieves robust accuracy while maintaining efficiency and clinical applicability. We have added a new subsection titled 'Hyperparameter Optimization' in the Methods section on Page 20 to provide detailed documentation of our optimization process and architectural decisions:

In Hyperparameter Optimization (Page 20):

The hyperparameter optimization of TSD-Net (Model 1) was conducted through Bayesian optimization over 100 trials, revealing the relative importance of different architectural components in 30-second

granular respiratory event detection (Supplementary Fig. 29). The `mlp_ratio`, controlling the hidden feature expansion in feed-forward networks, showed the highest importance and converged to 2. This moderate expansion suggests that the temporal patterns in physiological signals can be effectively captured without requiring extensive feature transformation capacity.

Dropout-related parameters (`drop_rate`, `attention_drop_rate`, and `path_drop_rate`) ranked as the second most important group but approached zero, indicating our four-stage progressive attention design provides sufficient inherent regularization. The `expand_ratio`, which determines channel expansion in convolutional attention modules, stabilized at 2, achieving effective feature representation while maintaining computational efficiency.

The head convolution kernel size demonstrated relatively low importance and converged to 3. This small convolution window is sufficient for capturing local respiratory features in the initial embedding stage, preparing informative local representations for subsequent large-scale attention processing in deeper layers.

For the AHI regression model (Model 2), we adopted a proven architecture design with three convolutional layers using decreasing kernel sizes (5→3→1) followed by two fully connected layers⁵. This simple yet effective architecture helps prevent overfitting while enabling efficient multi-scale temporal feature extraction. The lightweight network structure with progressively reduced kernel sizes ensures both computational efficiency and robust prediction accuracy for clinical deployment.

3) Cost and Implementation Analysis: We have clarified our statement regarding cost reduction by adding specific analysis in the *Methods* section 'The Concept and Cost Analysis of the AIX System' on Pages 20-21 and a comparative cost analysis figure (Supplementary Fig. 30). The cost advantages stem from our flexible monitoring configurations supporting simplified monitoring scenarios (such as SpO₂-only measurements), reduced manual scoring requirements, and efficient expert review system enabled by transparent AI assistance. We have added detailed cost analysis in the *Discussion* section comparing AIX with existing clinical solutions, demonstrating significant advantages in equipment cost, per-test expenses, and technical time requirements. As shown in Supplementary Fig. 30, traditional Lab-PSG systems require substantial initial investment (~\$68,000) with high per-test costs (\$1,000-1,500), while HSAT offers a more economical alternative but still involves considerable costs (\$14,000 for equipment, \$200-400 per test). In contrast, AIX achieves remarkable cost efficiency through flexible monitoring configurations, with equipment costs as low as \$15 when using a single-channel finger pulse oximeter. The per-test costs remain under \$10, covering disposable supplies (such as nasal cannulas), brief training requirements, equipment depreciation, and basic technical support. Additionally, the technical operation time is significantly reduced from 3-4 hours (Lab-PSG) or 1 hour (HSAT) to just 1-10 minutes through AIX's transparent AI-assisted interaction logic. We have also added detailed cost analysis in the *Methods* section under 'The Concept and Cost Analysis of the AIX System' on Pages 20-21.

In Discussion (Page 11):

The low operational threshold and flexible monitoring scenario adaptation of AIX significantly advance SA diagnosis popularization while reducing costs (Supplementary Fig. 30). Traditional Lab-PSG sleep studies, requiring high-cost equipment (>\$50,000) and several hours of technical time in strictly controlled laboratory environments, prove difficult to extend to home monitoring practices. While HSAT provides a more economical option with moderate equipment costs (~\$10,000-20,000), it still demands specialized technical expertise and considerable analysis time. In contrast, AIX's flexible workflow accommodates

multiple device configurations including portable PSG systems and simple pulse oximeters, with substantially lower per-test costs and reduced analysis time (1-10 minutes). This adaptability enables AIX to generate valuable analysis reports even with missing channel data, making it particularly suitable for home monitoring scenarios.

In The Concept and Cost Analysis of the AIX System (Pages 20-21):

To validate the cost-effectiveness of AIX, we conduct a comprehensive cost analysis comparing it with existing clinical solutions (Supplementary Fig. 30). All PSG devices included in the comparison are currently in clinical use, with their prices and average testing durations provided under the guidance of sleep medicine specialists. The prices of portable/simplified devices are obtained directly from equipment manufacturers, and the AIX system processing time is experimentally verified. All prices are converted to US dollars using current exchange rates. As shown in Supplementary Fig. 30, the AIX system demonstrates significant advantages in terms of equipment cost, per-test expenses, and technical time requirements. Traditional Lab-PSG systems require substantial initial investment (~\$68,000) and involve high per-test costs (\$1,000-1,500) with lengthy technical operations (3-4 hours). While HSAT offers a more economical alternative with lower equipment costs (\$14,000) and reduced per-test expenses (\$200-400), it still requires considerable technical time (~1 hour). In contrast, the AIX system demonstrates remarkable cost efficiency and flexibility, supporting various monitoring devices from wearable sensors to portable PSG configurations. With equipment costs as low as \$15 when using a single-channel finger pulse oximeter, and since AIX can be deployed on personal laptops without additional hardware requirements, the per-test costs remain under \$10, which includes disposable supplies (such as nasal cannulas), brief training requirements, equipment depreciation, and basic technical support. The technical operation time is streamlined to 1-10 minutes through its transparent AI-assisted interaction logic.

In Supplementary Material:

Supplementary Fig. 30 | Cost and operational comparison between Lab-PSG, HSAT, and the proposed AIX system for sleep apnea diagnosis. Cost and operational comparison between Lab-PSG, HSAT, and the proposed AIX system for sleep apnea diagnosis. AIX supports flexible monitoring channels while maintaining significantly lower equipment costs, per-test expenses, and technical time requirements. With transparent AI-assisted interaction logic, AIX is compatible with wearable devices, single-channel finger pulse oximeters, and portable home sleep apnea monitoring PSG configurations.

These revisions provide stronger scientific justification for our approach while maintaining clear documentation of implementation details and system advantages. The added cost analysis and implementation details demonstrate AIX's practical value in both clinical and research settings.

[5] Wang, T., Lu, C., Shen, G. & Hong, F. Sleep apnea detection from a single-lead ECG signal with automatic feature-extraction through a modified LeNet-5 convolutional neural network. *Peerj* 7(2019).

[9] Salari, N., et al. Detection of sleep apnea using Machine learning algorithms based on ECG Signals: A comprehensive systematic review. *Expert Systems with Applications* 187, 115950 (2022).

[10] Bahrami, M. & Forouzanfar, M. Sleep apnea detection from single-lead ECG: A comprehensive analysis of machine learning and deep learning algorithms. *IEEE Transactions on Instrumentation and Measurement* 71, 1-11 (2022).

[11] Penzel, T., Moody, G.B., Mark, R.G., Goldberger, A.L. & Peter, J.H. The apnea-ECG database. in *Computers in*

- Cardiology* 2000. Vol. 27 (Cat. 00CH37163) 255-258 (IEEE, 2000).
- [12] Wang, S., *et al.* Machine learning assisted wearable wireless device for sleep apnea syndrome diagnosis. *Biosensors* **13**, 483 (2023).
- [13] Retamales, G., *et al.* Towards automatic home-based sleep apnea estimation using deep learning. *npj Digital Medicine* **7**, 144 (2024).
- [14] Hu, S., *et al.* IPCT-Net: Parallel information bottleneck modality fusion network for obstructive sleep apnea diagnosis. *Neural Networks* **181**, 106836 (2025).
-

6. Limitations must include the specific characteristics of the databases used (eg, comorbidities) and discussion should take into account that SHHS dataset are indeed AT HOME recordings.

Response: Thank you for your valuable comments. We have made two significant revisions to address your concerns:

1) We have accurately characterized the Sleep Heart Health Study (SHHS), which recruited 6,600 adult participants aged 40 years and older from ongoing cohort studies to investigate prospectively whether sleep-disordered breathing (SDB) is an independent risk factor for cardiovascular and cerebrovascular diseases. As participants underwent unattended home polysomnography, where signal quality and patient compliance may vary significantly, our validation using SHHS data provides compelling evidence for AIX's adaptability to real-world home monitoring scenarios.

2) We have enhanced our discussion of the cohorts' distinct characteristics: MESA focused on subclinical cardiovascular disease across ethnic groups, specifically enrolling middle-aged and older participants initially free of clinical cardiovascular disease to study disease progression. CFS specifically examined familial aggregation of sleep apnea, demonstrating that sleep-disordered breathing clusters within families independently of obesity. MROS enrolled older men aged 65 years or older who were not selected on the basis of sleep problems or cognitive impairment to study osteoporotic fractures and cardiovascular outcomes, with participants having age-related comorbidities such as cognitive impairment and cardiovascular conditions.

These revisions have been incorporated into the manuscript in Discussion section.

In Discussion (Page 11):

Our validation using the SHHS database, which comprises unattended home sleep recordings where signal quality and patient compliance may vary significantly, provides compelling evidence for AIX's adaptability to home monitoring scenarios. The validation across these community-based cohorts demonstrates AIX's effectiveness in populations with diverse characteristics. SHHS was designed to investigate cardiovascular outcomes in sleep-disordered breathing with participants having various cardiovascular comorbidities including hypertension, coronary heart disease, and cerebrovascular disease. MESA focused on subclinical cardiovascular disease across ethnic groups, specifically enrolling middle-aged and older participants initially free of clinical cardiovascular disease to study disease progression. The CFS specifically examined familial aggregation of sleep apnea, demonstrating that sleep-disordered breathing clusters within families independently of obesity. MROS enrolled older men aged 65 years or older who were not selected on the basis of sleep problems or cognitive impairment to study osteoporotic fractures and cardiovascular outcomes, with participants having age-related comorbidities such as

cognitive impairment and cardiovascular conditions. The robust performance maintained across these heterogeneous populations demonstrates AIX's capability in handling diverse clinical scenarios. From family-based studies investigating genetic factors to large community-based cohorts with varying comorbidity profiles, AIX consistently delivers reliable results, indicating its readiness for real-world clinical applications involving complex interactions of genetic and environmental factors.

Despite validation across multiple cohorts with varying study designs and population characteristics demonstrates robustness, systematic evaluation of AIX's performance in specific clinical subgroups, such as those with strong family history or particular comorbidities, remains an important direction for future research. Furthermore, while our results validate performance in home-based settings, further studies are needed to evaluate AIX across different home monitoring devices and in resource-limited settings.

Minor issues:

- There are repeated references in the list with different numbers

***Response:** We sincerely appreciate your thorough review. Upon careful examination, we have revised instances where the same reference appeared under different citation numbers. For example, we identified that references #17 and #49, as well as #21 and #51, were duplicates of the same publications. Thank you for your meticulous attention to detail.*

Responses to Reviewer #2:

Manuscript ID: NCOMMS-24-73977-T

Title: Transparent Artificial Intelligence – enabled Interpretable and Interactive Sleep Apnea Assessment across Flexible Monitoring Scenarios

Comments to the Author:

Hu, Liu, and their collaborators...

The authors have made a significant contribution by integrating multiple modalities for an assessment of sleep apnea (SA) and proposing an interpretable and interactive model, the so-called Apnea Interact Xplainer (AIX). They reported that AIX achieved a sensitivity of 0.949 for early SA detection using only oximetry signal, and provides nocturnal risk assessment and intelligent monitoring reports. This represents a significant advance in early and cost-effective SA diagnosis through transparent AI.

This study seems interesting, and it is commendable that they used 15,510 polysomnographic records from multi-ethnic databases. However, I have the following major concerns. There are several limitations that need to be addressed appropriately.

***Response:** We sincerely thank the reviewer for this comprehensive evaluation of our work. We are encouraged by the reviewer's recognition of AIX's contributions to sleep apnea assessment, particularly regarding its integration of multiple channels, interpretability features, and the demonstrated performance in early detection. The acknowledgment of our efforts in validating the system across large-scale multi-ethnic databases (15,510 polysomnographic records) is greatly appreciated. The reviewer has raised several important limitations that have helped us critically examine and improve our work. We address each concern in detail below.*

- 1. First of all, what was the rationale for choosing SpO₂ as a single modality among the various indices? While total apnea-hypopnea index (AHI) is considered the gold standard most commonly used to diagnose sleep apnea (SA), SpO₂ is also widely used as an important marker. However, given that SpO₂ alone is often highly correlated with SA, the rationale for choosing SpO₂ as the primary single modality input requires further clarification. Why was SpO₂ prioritized over other modalities for this evaluation? Oximetry measurements are undoubtedly valuable, but their limitations in distinguishing complex respiratory events might warrant further discussion.**

***Response:** Thank you for this important question that gets to the heart of our methodological choices. We deeply appreciate the opportunity to clarify our rationale for selecting SpO₂ as the primary single channel.*

From a clinical perspective, SpO₂ is directly linked to the pathophysiological mechanisms of sleep apnea. The AASM guidelines specifically include oxygen desaturation ($\geq 3\%$) as a key criterion for scoring respiratory events ⁷, as these desaturation patterns provide direct evidence of the physiological impact of

breathing cessation. Recent large-scale research by Levy et al. (Nature Communications, 2023 ⁸) has compellingly demonstrated that SpO₂-based analysis can achieve robust OSA diagnosis, with only 0.2% missed moderate-to-severe cases compared to 21% for traditional metrics across multiple external validation cohorts.

We acknowledge the inherent limitations of SpO₂ in characterizing complex respiratory events. As demonstrated in previous study ⁸, SpO₂ signals alone cannot definitively differentiate between central and obstructive events, as both may present similar desaturation patterns. Additionally, subtle respiratory disturbances that do not lead to significant desaturation may be missed. The temporal relationship between respiratory effort and airflow cessation, crucial for distinguishing event types, cannot be directly assessed through SpO₂ monitoring alone. These limitations have been well-documented in clinical research and remain important considerations in sleep medicine.

However, several factors influenced our decision to proceed with SpO₂ as the primary channel. First, while detailed event characterization is valuable for research and specific clinical scenarios, the primary goal of initial sleep apnea screening is to identify patients with significant sleep-disordered breathing who require further evaluation. Our experimental results demonstrate that SpO₂-based analysis achieves this goal effectively, with R² values of 0.93-0.96 and four-level classification accuracy of 0.74-0.78 across external test sets.

The practical implementation advantages of SpO₂ monitoring are also substantial. It requires only a simple finger sensor, making it non-invasive and comfortable for long-term monitoring. This is particularly important when compared to nasal cannulas for Flow signals or chest bands for respiratory effort, which can be uncomfortable and prone to displacement during sleep. The signal stability of SpO₂ is notably superior, with fewer artifacts and more consistent quality throughout the night.

Our transparent AI framework enhances the utility of SpO₂ monitoring by enabling detailed pattern visualization and expert review of ambiguous cases. While this does not overcome the fundamental limitations of single-channel monitoring, it provides clinicians with additional context for interpretation and helps identify cases that may require more comprehensive evaluation.

This choice aligns with our goal of developing an accessible screening tool that maintains diagnostic rigor while acknowledging the need for comprehensive PSG in complex cases. The robust performance in detecting clinically significant sleep apnea, combined with practical advantages in implementation, makes SpO₂ particularly well-suited for expanding early detection while ensuring appropriate referral for detailed evaluation when necessary. We have added the following content to the main text:

In Discussion (Pages 11-12):

Additionally, while SpO₂-based monitoring demonstrates excellent capability in detecting clinically significant SA, there are inherent physiological limitations in characterizing specific respiratory event types when using single-channel monitoring alone. Particularly, SpO₂ signals cannot definitively differentiate between central and obstructive events, as both may present similar desaturation patterns without the corresponding respiratory effort and airflow information ⁷. The synchronous assessment of respiratory effort presence and airflow cessation, which is essential for distinguishing between central and obstructive events, cannot be captured through SpO₂ monitoring alone. However, our research demonstrates that for the primary goal of identifying patients requiring further evaluation, SpO₂-based analysis maintains robust diagnostic effectiveness, particularly in large-scale screening applications where accessibility is paramount. Another challenge lies in achieving accurate sleep staging with limited channels.

In Different Channel Types for Different Scenarios (Page 15):

Recent large-scale studies have compellingly demonstrated that SpO₂-based analysis can achieve robust SA diagnosis, with only 0.2% missed moderate-to-severe cases compared to 21% for traditional metrics across multiple validation cohorts⁸. The selection of SpO₂ as a primary single channel is supported by both clinical and practical considerations. From a clinical perspective, SpO₂ directly reflects the physiological impact of breathing cessation during SA, and the AASM guidelines specifically include oxygen desaturation ($\geq 3\%$) as a key criterion for scoring respiratory events¹⁵. The practical advantages of SpO₂ are substantial. It requires only a simple finger sensor for non-invasive monitoring, demonstrates superior signal stability with fewer artifacts compared to Flow signals requiring nasal cannulas or Chest signals requiring thoracic bands, and enables comfortable long-term monitoring in both clinical and home-based settings. These characteristics make SpO₂ particularly valuable for expanding screening coverage while maintaining diagnostic reliability.

To further quantitatively validate our channel selection, we conducted SHAP (SHapley Additive exPlanations) analysis on feature contributions across different channels based on the gold standard three-channel (Flow, Chest, and SpO₂) sleep apnea detection model (Supplementary Fig. 11). As shown in Supplementary Fig. 12, SpO₂-derived features exhibited significantly higher SHAP values (mean \pm SD: 3.39 ± 0.02) compared to Flow (0.54 ± 0.02) and Chest (0.73 ± 0.01) features. This quantitative evidence aligns with clinical observations and further supports our choice of SpO₂ as the primary channel for simplified monitoring scenarios.

In Supplementary Material:

Supplementary Fig. 11 | Feature-level SHAP analysis of respiratory signals in model prediction. Feature-wise SHAP values showing the impact of individual features on model predictions. The color represents the feature value (blue: low, red: high), and the horizontal position shows the SHAP value impact on model output. Features are ordered by their absolute SHAP values.

Supplementary Fig. 12 | Quantitative comparison of SHAP value magnitudes across respiratory channels. Comparison of total SHAP value magnitudes across different signal channels (Flow, Chest, and SpO₂). Values are presented as mean ± standard deviation, demonstrating the relative importance of each channel type in the model's decision-making process.

The feature-level SHAP distribution reveals that SpO₂ features consistently show stronger predictive power for respiratory events compared to other channels in the gold standard configuration. Each data point in the SHAP analysis represents a sample, with color indicating the feature value (red for high, blue for low) and horizontal position showing the impact on model predictions (positive values indicate increased likelihood of respiratory events). The clear separation and larger magnitude of SpO₂ feature SHAP values demonstrates its superior discriminative power in sleep apnea detection. This explains why the SpO₂-only configuration can achieve comparable performance to the gold standard three-channel approach while offering significant advantages in implementation simplicity.

This quantitative analysis provides compelling evidence that SpO₂ signals capture the most essential information for sleep apnea diagnosis, supporting our strategic decision to prioritize this channel for accessible screening applications while maintaining robust diagnostic performance.

- [7] Berry, R.B., *et al.* Rules for Scoring Respiratory Events in Sleep: Update of the 2007 AASM Manual for the Scoring of Sleep and Associated Events. *Journal of Clinical Sleep Medicine* **8**, 597-619 (2012).
- [8] Levy, J., Álvarez, D., Del Campo, F. & Behar, J.A. Deep learning for obstructive sleep apnea diagnosis based on single channel oximetry. *Nature Communications* **14**, 4881 (2023).
- [15] Berry, R.B., *et al.* Rules for scoring respiratory events in sleep: update of the 2007 AASM manual for the scoring of sleep and associated events: deliberations of the sleep apnea definitions task force of the American Academy of Sleep Medicine. *Journal of clinical sleep medicine* **8**, 597-619 (2012).

2. In the abstract, the authors mentioned that AIX achieved 99.8% accuracy within a single severity grade. However, the confusion matrix in Figure 4(a) shows that the accuracy for classifying into four severity levels (normal, mild, moderate, and severe) ranges from 57% to 88%. A more meaningful task in the real world would be to predict the severity level rather than a binary

classification whether subjects have SA or not. The discrepancy should be addressed as SA management heavily depends on accurate severity classification to consider specific treatment methods, and the lack of clear focus on multiclass outcomes could limit its clinical applicability.

***Response:** We appreciate the reviewer's attention to this important aspect of our work. We apologize for not sufficiently emphasizing our four-level severity classification results in the original manuscript, although these experiments were indeed conducted and included in our study.*

You are absolutely right that accurate severity classification is crucial for clinical decision-making and treatment selection. In fact, our study did implement and evaluate four-level severity classification (normal, mild, moderate, and severe), but we acknowledge that this was not prominently highlighted in our presentation. To address this oversight, we have revised our manuscript to better showcase AIX's performance in this critical aspect:

In the Abstract on Page 1, we now specify: 'Analyzing 15,807 polysomnography records from seven independent multi-ethnic databases, AIX achieves accuracies of 0.744-0.806 for four-level SA severity classification, with 99.7% accuracy within one severity grade and R-squared of 0.92-0.96 for apnea-hypopnea index prediction on external test sets.'

In the Results section 'Flexible Channel Configuration Performance in AHI Regression and SA Severity Classification' on Pages 4-8 and Supplementary materials, we have added comprehensive analysis of severity classification performance across different channel configurations (Supplementary Fig. 1). The Gold standard configuration (Flow+Chest+SpO₂) achieves accuracy of 0.744-0.806 and macro F1 scores of 0.740-0.808. Notably, the SpO₂-only configuration demonstrates comparable performance with accuracy of 0.751-0.785 and macro F1 scores of 0.746-0.789, while Flow and Chest configurations show relatively lower performance.

Cross-database validation demonstrates robust generalizability across diverse patient populations from SHHS2 (n=2,522), MESA (n=1,846), MROS (n=3,522), CFS (n=576), and FDU-HSH (n=327). While MESA shows slightly lower performance compared to other cohorts, classification errors primarily occur between adjacent severity categories. The strong performance of the single-channel SpO₂ configuration, comparable to the Gold standard, suggests its potential for simplified home-based monitoring.

These results comprehensively demonstrate AIX's strong capability in clinically relevant severity stratification while maintaining accessibility through simplified monitoring options.

In Abstract (Page 1):

Analyzing 15,807 polysomnography records from seven independent multi-ethnic databases, AIX achieves accuracies of 0.744-0.806 for four-level SA severity classification, with 99.7% accuracy within one severity grade and R-squared of 0.92-0.96 for apnea-hypopnea index prediction on external test sets.

In Flexible Channel Configuration Performance in AHI Regression and SA Severity Classification (Pages 4-8):

For AHI regression, the Gold configuration demonstrates excellent performance with R-squared (R²) and intraclass correlation coefficient (ICC) values consistently above 0.90 across all test cohorts: SHHS2 (R²=0.96, ICC=0.96), MESA (R²=0.93, ICC=0.92), MROS (R²=0.96, ICC=0.96), and CFS (R²=0.93, ICC=0.92). More importantly for clinical application, we thoroughly evaluate the system's performance in four-level SA severity classification (normal, mild, moderate, severe). The Gold configuration achieves robust classification accuracy across datasets: SHHS2 (ACC=0.81, Macro F1=0.81), MESA (ACC=0.74,

Macro F1=0.74), MROS (ACC=0.80, Macro F1=0.79), and CFS (ACC=0.80, Macro F1=0.79). Analysis of confusion matrices (Fig. 4) reveals that classification errors predominantly occur between adjacent severity categories, which is clinically acceptable given the continuous nature of AHI measurements. This pattern aligns with clinical practice where misclassifications primarily occur for cases with AHI values near the categorical boundaries, having minimal impact on clinical decision making. The detailed performance comparison across different datasets is provided in Supplementary Fig. 1 and Supplementary Table 1.

Notably, the SpO₂-only configuration maintains strong severity classification capability with only modest degradation from the Gold standard: SHHS2 (ACC=0.78, F1=0.78), MESA (ACC=0.75, F1=0.74), MROS (ACC=0.79, F1=0.77), and CFS (ACC=0.77, F1=0.78). The preservation of accurate severity classification even with single-channel monitoring is particularly significant for expanding screening accessibility. In contrast, the Flow-only configuration shows moderate performance (SHHS2: ACC=0.62; MESA: ACC=0.48; MROS: ACC=0.59; CFS: ACC=0.49), while Chest-only configuration exhibits similar patterns (SHHS2: ACC=0.59; MESA: ACC=0.48; MROS: ACC=0.63; CFS: ACC=0.55).

To further validate clinical applicability across severity levels, we analyze classification performance using different AHI thresholds (Supplementary Tables 5-7). The results demonstrate consistent performance across severity transitions. Particularly for early SA detection (AHI cutoff=5), the SpO₂-only configuration demonstrates excellent sensitivity across all test cohorts (SHHS2: 0.965, MESA: 0.947, MROS: 0.980, CFS: 0.956), closely matching the Gold standard (SHHS2: 0.970, MESA: 0.952, MROS: 0.985, CFS: 0.948). This is particularly significant for population-level screening and early intervention.

In Discussion (Page 10):

The system's robust performance in four-level SA severity classification (ACC=0.74-0.81) and binary screening (AHI cutoff at 5) across multiple external test sets (ACC=0.896-0.939 for Gold standard configuration, ACC=0.885-0.933 for SpO₂-based configuration) supports reliable clinical stratification while maintaining accessibility through simplified monitoring configurations.

In Supplementary Material:

Supplementary Fig. 1 | Performance comparison of different channel configurations for four-level SA severity classification across test cohorts. The left panel shows classification accuracy and the right panel shows macro F1 scores across different datasets. SpO₂ and Gold configurations consistently demonstrate superior performance (Accuracy: SpO₂ 0.751-0.785/Gold 0.744-0.806 and macro F1 scores: SpO₂ 0.746-0.789/Gold 0.740-0.808) compared to Flow and Chest configurations. The

performance remains stable across diverse patient populations in SHHS2 (n=2,522), MESA (n=1,846), MROS (n=3,522), CFS (n=576), and FDU-HSH (n=327), with MESA showing slightly lower performance. Single-channel SpO₂ configuration achieves comparable performance to the Gold standard, suggesting its potential for simplified home-based monitoring.

3. As a related question, the authors did not specify whether AIX was tested in real-world scenarios, such as home monitoring settings where signal quality and patient compliance can vary significantly. Validation in such settings needs to be addressed to provide critical insights into its robustness.

Response: We appreciate the reviewer's important question about AIX's validation in real-world scenarios. As shown in Supplementary Fig. 24, we have validated AIX with physical implementation in real-world scenarios, including home sleep monitoring settings. Our validation strategy encompasses three complementary aspects:

1) Validation using SHHS home-based recordings: First, it's worth noting that the SHHS dataset (SHHS1 n=5,793, SHHS2 n=2,651) used in our study is particularly valuable for real-world validation as it consists entirely of home-recorded data. SHHS recruited 6,441 adult participants aged 40 years and older from ongoing cohort studies to investigate prospectively whether sleep-disordered breathing is an independent risk factor for cardiovascular and cerebrovascular diseases. As participants underwent unattended home polysomnography, where signal quality and patient compliance naturally vary, this dataset provides inherent validation of AIX's performance under real home monitoring conditions. The robust performance on SHHS data, which includes patients with various cardiovascular comorbidities including hypertension, coronary heart disease, and cerebrovascular disease, demonstrates AIX's adaptability to diverse clinical scenarios in home settings. We have elaborated on this validation approach in the Discussion Section (Page 11):

'Our validation using the SHHS database, which comprises unattended home sleep recordings where signal quality and patient compliance may vary significantly, provides compelling evidence for AIX's adaptability to home monitoring scenarios. The validation across these community-based cohorts demonstrates AIX's effectiveness in populations with diverse characteristics.'

2) Prior real-world validation: Additionally, AIX has been previously validated on the FDU-HSH retrospective dataset (n=350) collected from January 2021 to December 2023 in real-world scenarios. This validation phase was crucial as it assessed AIX's performance across a spectrum of monitoring environments where varying signal quality and patient compliance are common challenges. The system demonstrated robust screening capability in this Asian population cohort, achieving excellent performance for early SA detection (AHI cutoff=5) with ACC=0.902, SEN=0.895, and SPE=0.940. For four-level severity classification, it maintained strong performance with ACC=0.783 and Macro F1=0.778. This retrospective validation established a strong foundation for understanding AIX's performance under actual real-world conditions.

3) Extended prospective validation: To provide further critical insights into AIX's robustness in home-based monitoring, we conducted a prospective validation using the FDU-HSH dataset (n=297) collected from December 2023 to December 2024 in real home environments. This validation setting reflects genuine home sleeping conditions while maintaining high-quality reference standards, providing crucial insights into AIX's performance in actual domestic use scenarios. As shown in Fig. 6, this validation demonstrates

AIX's reliable performance under real-world conditions. The system achieves excellent respiratory event detection (AUC=0.943) with robust performance maintained even with selective review of ambiguous cases (Supplementary Fig. 31a-c). The confusion matrix (Supplementary Fig. 31d) shows strong four-level severity classification (ACC=0.789, Macro F1=0.788), while scatter plots and Bland-Altman analysis (Supplementary Fig. 31e,f) demonstrate reliable AHI prediction ($R^2=0.921$, ICC=0.917, MAE=5.833). Importantly, the SARI metric shows significant discrimination between different severity categories (two-tailed Mann-Whitney U test, $P=1.47e-17$, $P=1.27e-16$, $P=2.57e-19$ between adjacent groups), validating its utility as a risk stratification tool in home monitoring applications (Supplementary Fig. 31g,h). We have added the following content to the relevant sections of the manuscript:

In Overview of Experimental Procedures (Page 3):

The remaining two datasets come from FDU-HSH Sleep Research, including a retrospective cohort (n=350) and a prospective cohort (n=297), which evaluate the AIX system's performance with portable monitoring configurations using real-world oximetry signals.

In Human-AI Collaboration Feasibility of AIX (Page 9):

This quantifiable uncertainty pattern serves as the foundation of our trust assessment framework, providing users with reliable indicators for identifying cases requiring verification. The significant separation between confident and uncertain predictions enables a data-driven approach to building trust in the system's decision-making process. To systematically evaluate the system's trust-building capabilities through interactive review, we conducted experiments on both FDU-HSH retrospective (n=327) and prospective (n=265) cohorts using SpO₂ monitoring (Supplementary Fig. 31, Supplementary Tables 8,9). Taking the FDU-HSH retrospective cohort as an example, we examined the relationship between interaction threshold intervals and error rates (Fig. 6a-c) to understand their trade-off patterns. For instance, with an interaction threshold interval of 0.1-0.2, we observed that 3.66% of samples were identified for review, with corresponding FP rate of 0.96% and FN rate of 0.22% (Supplementary Table 8). This illustrates how interaction thresholds can be leveraged to balance review efficiency and error control.

In AIX Real-World Testing and SA Risk Assessment (Pages 9-10):

We validate its practical applicability through external testing on both FDU-HSH retrospective (n=327) and prospective (n=265) cohorts (Fig. 6d-h, Supplementary Fig. 22, and Supplementary Fig. 31). The prospective validation demonstrates excellent measurement agreement (ICC=0.9174, MAE=5.8330) and strong correlation (R-squared=0.9207, $P<0.0001$) in AHI prediction. The confusion matrix shows high sensitivity for healthy (100%) and severe (87%) cases, confirming the system's robust diagnostic performance.

These findings are further validated in the prospective cohort, where SARI maintains strong correlation with AHI (Spearman's $\rho=0.952$, $P=3.80e-137$) and demonstrates significant differentiation between adjacent severity levels (two-tailed Mann-Whitney U test: $P=1.47e-17$ for healthy-mild, $P=1.27e-16$ for mild-moderate, and $P=2.57e-19$ for moderate-severe transitions).

In Discussion (Page 10):

More significantly, our real-world validation in home environments demonstrates AIX's robust

performance using simplified monitoring configurations. In the retrospective cohort (n=327), AIX maintains an ACC of 0.78 in four-level severity classification using nocturnal oximetry signals alone, with binary event classification accuracy and sensitivity improving from 0.867 and 0.914 to 0.939 and 0.966 respectively through transparent AI-guided interactive review of approximately 20% ambiguous cases. These findings are further validated in our prospective cohort (n=265), where the system maintains excellent performance metrics (AUC=0.943 for respiratory event detection, ACC=0.789 for four-level severity classification, R2=0.921 and ICC=0.917 for AHI prediction) when analyzing single-channel SpO₂ signals. The SARI metric demonstrates significant discriminative power between adjacent severity categories (P<0.0001 for all transitions), supporting its utility in risk stratification. These comprehensive validation results from both cohorts provide strong evidence for AIX's reliability in simplified monitoring scenarios while maintaining high diagnostic standards.

In Large-Scale and Multi-Ethnic Cohort Description (Pages 14-15):

This study comprises a retrospective cohort of 350 participants enrolled from January 2021 to December 2023 in hospital settings and a prospective cohort of 297 participants monitored from December 2023 to December 2024 in home environments. The retrospective cohort establishes baseline validation in clinical settings, while the prospective cohort evaluates system performance under genuine home sleeping conditions with maintained high-quality reference standards. The primary goal is to assess the potential of the AIX system in real-world data applications. For both cohorts, overnight PSG recordings are available, along with analyses of patient age, BMI, and AHI index.

In Supplementary Material:

Supplementary Fig. 24 | The application deployment of the AIX system in both professional medical and home settings is presented. (a) In the professional setting, sleep monitoring is conducted through PSG, with the system supporting multi-channel signal analysis. (b) In the home setting, portable monitoring is achieved using wearable devices, and the system provides a single-channel analysis interface. Both scenarios demonstrate the workflow from data collection and AIX system deployment to the user interface, highlighting the adaptability and flexibility of AIX in different monitoring environments.

Supplementary Fig. 31 | AIX real-world validation on FDU-HSH prospective cohort (n=265). (a) ROC curve demonstrating SpO₂-based binary respiratory event classification performance. (b) Ternary relationship between false positive rate, false negative rate, and sample proportion across different interaction threshold intervals. (c) Binary classification performance metrics (SPE, SEN, ACC) under varying manual review ratios. (d) Confusion matrix for four-level SA severity classification, showing ACC=0.789 and Macro F1=0.788 using SpO₂-only configuration. (e) Scatter plot of predicted versus true AHI values with regression line ($R^2=0.921$, $P<0.0001$). (f) Bland-Altman plot comparing predicted and true AHI values (ICC=0.917, MAE=5.833). (g) Spearman correlation between SARI and AHI values ($\rho=0.952$, two-sided $P=3.80e-137$). (h) Box plot of SARI distributions across severity groups with significant differences between adjacent categories (two-tailed Mann-Whitney U test, $P=1.47e-17$, $P=1.27e-16$, $P=2.57e-19$). The model maintains consistent classification and regression performance on this independent prospective validation cohort (n=265) compared to the retrospective FDU-HSH dataset (n=327).

This comprehensive validation strategy incorporates unattended home recordings from SHHS, real-world retrospective data, and prospective validation in home environments. The extensive testing provides compelling evidence for AIX's reliability and robustness across the spectrum of sleep monitoring scenarios. The consistent performance across these different settings and populations, particularly in actual home environments, demonstrates AIX's readiness for widespread deployment in home-based sleep monitoring applications.

4. Most importantly, what was the rationale for using LightGBM, a gradient-boosting decision tree algorithm, instead of other deep learning approaches that are becoming increasingly viable in medical AI? In fact, the authors evaluated six machine learning algorithms and ultimately chose LightGBM as their final model. While the choice appears justified based on performance metrics, the manuscript does not address why they did not consider or apply deep learning approaches. Although the authors emphasize “transparent AI” in the abstract, it remains unclear why they chose this particular framework instead of incorporating explainable deep learning techniques such as convolutional neural networks (CNNs) or recurrent neural networks (RNNs). LightGBM is computationally efficient, but appears to be less powerful in capturing complex patterns than

modern deep learning models such as CNNs or RNNs, which can more effectively explore temporal and spatial patterns in the data. Given the growing popularity of deep learning in SA evaluation, it is important to discuss the decision to exclude deep learning methods. Incorporating advanced models, such as time-series data transformers or hybrid deep learning approaches could improve prediction accuracy and generalization, and can be particularly helpful in capturing subtle patterns in sleep-related physiological signals.

Response: Thank you for this important comment regarding our model architecture. We would like to clarify that our AIX system does utilize deep learning approaches as its primary components. Specifically:

1) For single-channel analysis, we employ a novel deep learning architecture called Transparent Scale Diffusion Network (TSD-Net), which combines multi-head convolutional attention (MSCA) and transformer-based self-attention mechanisms. As illustrated in Supplementary Fig. 3, TSD-Net's architecture features a unique design that effectively integrates these components, with detailed network parameters and configurations provided in Supplementary Table 10. Unlike traditional CNNs or RNNs, this hybrid architecture captures both local respiratory patterns through MSCA and long-range physiological dependencies through transformer self-attention mechanisms. The transformer component is particularly advantageous over RNNs for sleep apnea analysis, as it can directly model long-term dependencies in physiological signals without the sequential constraints and vanishing gradient issues inherent to RNNs.

2) For multi-channel integration, our approach uses three parallel TSD-Nets (one for each channel type: Flow, Chest, and SpO₂) as feature extractors. As shown in Supplementary Fig. 2, each TSD-Net outputs 8 abstract features, resulting in 24 concatenated features from all three channels. We then use LightGBM specifically for this final feature integration step, not as the primary modeling approach.

3) The choice of LightGBM for multi-channel feature integration, rather than another deep learning layer, was made after careful consideration and extensive experimentation (Supplementary Fig. 9). Our comparison shows that LightGBM achieves superior performance (AUC: 0.989-0.992 for validation and 0.818-0.822 for testing) with significantly faster training (3.80 μ s) and inference (0.54 μ s) times compared to other approaches, making it particularly suitable for real-time clinical applications while maintaining high accuracy.

4) Our TSD-Net demonstrates superior interpretability compared to traditional approaches for the following reasons:

- Traditional CNNs primarily focus on local feature extraction through convolution operations, which limits their ability to capture long-range dependencies in physiological signals. While they can identify local respiratory patterns, they struggle to model the complex temporal relationships that exist in sleep apnea events. In contrast, our MSCA mechanism not only captures local features but also provides scale-specific attention maps that clearly show which signal segments contribute to event detection.
- RNNs, while capable of processing sequential data, suffer from inherent limitations such as vanishing gradients and difficulties in parallel processing, making them less suitable for real-time analysis of long physiological signals. Furthermore, their sequential nature makes it challenging to interpret which historical information influences current predictions.
- Standard Transformers, though powerful in modeling long-range dependencies, often create attention maps that are difficult to interpret clinically. Our TSD-Net incorporates a modified Transformer architecture with the scale diffusion mechanism (Supplementary Fig. 3), which explicitly shows how different temporal scales contribute to respiratory event detection. This makes the

decision-making process more transparent and clinically meaningful.

- The hierarchical feature extraction in TSD-Net maintains physiological interpretability at each processing stage (Supplementary Figs. 14-17). The multi-scale analysis provides clinicians with clear insights into how different temporal patterns contribute to the final diagnosis, unlike traditional deep learning approaches where feature representations become increasingly abstract and disconnected from physiological meaning.

5) Our architecture actually represents a hybrid approach that leverages both deep learning for complex feature extraction and traditional machine learning for efficient feature integration. This design choice aligns with our goal of achieving both high performance and interpretability, as demonstrated by our transparent scale diffusion mechanism (Supplementary Figs. 14-17). The strong performance across multiple external validation cohorts ($ACC=0.744-0.806$, $R^2=0.92-0.96$) demonstrates the effectiveness of our chosen architecture. We have made the following additions to better explain our methodological choices:

In AIX for AHI Prediction and Transparent Granular Traceability (Page 4):

For single-channel scenarios (Flow, Chest, or SpO₂), Model 1 processes input signals of shape $[B,1,256]$ through our transparent scale diffusion network (TSD-Net) to enable expert-logic traceability through transparent scale diffusion mechanism that visualizes multi-level attention patterns for AI decision interpretation, while outputting probability values for respiratory event detection. These 30-second probability values are concatenated to form the overnight probability sequence. For multi-channel configurations, Model 1 deploys parallel TSD-Nets for each channel, with each network serving as a feature extractor. Through average pooling along the channel dimension and flattening the features from the output of stage 4 (shape $[B,512,8]$, Supplementary Table 10), each TSD-Net outputs 8 abstract features (shape $[B,8]$). These features are concatenated to form a 24-dimensional feature vector (8 features \times 3 channels), which is then processed by LightGBM¹⁶ to produce the overnight probability sequence.

In Granularity Prediction Model Architecture (Pages 17-18):

Unlike traditional convolutional neural networks (CNNs) that focus only on local patterns or recurrent neural networks (RNNs) that process sequences step by step, TSD-Net combines the advantages of both convolutional and Transformer architectures. The MSCA modules effectively capture local respiratory patterns through multi-scale convolutions, while the MHSA modules, based on Transformer self-attention mechanisms, excel at modeling long-range physiological dependencies without the sequential constraints of RNNs. This hybrid architecture is particularly suitable for SA analysis, where both local respiratory events and their long-term temporal relationships are crucial for accurate diagnosis.

In Supplementary Material:

Supplementary Fig. 2 | The AIX system operates under home setting (single-channel) and professional setting (multi-channel) workflows. (a) Single-channel signals for the entire night are segmented into 30-second granularity and fed into the granularity prediction model (Model 1). This model provides detailed 30-second AI analysis decision logic based on a transparent scale diffusion mechanism. Subsequently, the granular probabilities of normal and respiratory events for the entire night are resampled to a length of 1024 to report overnight SA risk index (SARI) and train and predict with the AHI regression model (Model 2), resulting in the final AHI value. Two interpretability markers, 30-second transparent granularity attention and SARI, are obtained throughout the workflow. (b) Multi-channel signals for the entire night are split according to signal type, and a Model 1 is trained for each signal type. The trained Model 1 is used as a feature extractor to output 8 features per signal type for 30-second granularity. These features are concatenated to form 24 features. These features are trained using lightGBM to output multi-channel granular prediction probabilities. Finally, the resampled granular probabilities for the entire night are used for AHI regression with Model 2. Similar to the AIX single-channel workflow, the multi-channel workflow outputs the SARI and the 30-second transparent granularity attention for each signal type.

Supplementary Fig. 3 | Overall structure of the granularity prediction model with individual constituent blocks. (a) General structure of the granularity prediction model, Transparent scale diffusion network (TSD-Net). (b) Composition of multi-scale convolutional attention (MSCA) block. (c) Multi-head convolutional encoder (MHCE) module. (d) Cross-scale encoding fusion (CSEF) module. (e) Squeeze-and-excitation network (SE-Net). (f) Specific composition of MSCA-MHSA (multi-head self-attention) block.

Supplementary Table 10 | Details of the parameterization of the transparent scale diffusion network (TSD-Net). TSD-Net-T and TSD-Net-B represent the Tiny and Base versions of TSD-Net, respectively; "dim 64": An embedding dimension of 64; "sam.head 4": The multi-head convolutional encoder (MHCE) module with 4 heads; "sam.ep.r 2": An expansion rate of 2 for the cross-scale encoding fusion (CSEF) module; "msa.head 8": The multi-head self-attention (MHSA) module with 8 heads.

Layer name	Output Size [B, C, L]	TSD-Net-T	TSD-Net-B
Stem	[B, 64, 64]	conv1d 2, 64-d, BN	conv1d 2, 64-d, BN
Stage 1	[B, 64, 64]	conv1d 2, 64-d, LN	conv1d 2, 64-d, LN
		$\begin{bmatrix} \text{dim 64} \\ \text{sam.head 4} \\ \text{sam.ep.r 2} \end{bmatrix} * 1$	$\begin{bmatrix} \text{dim 64} \\ \text{sam.head 4} \\ \text{sam.ep.r 2} \end{bmatrix} * 2$
Stage 2	[B, 128, 32]	conv1d 3, 128-d, LN	conv1d 3, 128-d, LN
		$\begin{bmatrix} \text{dim 128} \\ \text{sam.head. 4} \\ \text{sam.ep r. 2} \end{bmatrix} * 1$	$\begin{bmatrix} \text{dim 128} \\ \text{sam.head. 4} \\ \text{sam.ep r. 2} \end{bmatrix} * 2$
Stage 3	[B, 256, 16]	conv1d 3, 256-d, LN	conv1d 3, 256-d, LN
		$\begin{bmatrix} \text{dim 256} \\ \text{sam.head 4} \\ \text{sam.ep.r 2} \\ \text{msa.head 8} \end{bmatrix} * 1$	$\begin{bmatrix} \text{dim 256} \\ \text{sam.head 4} \\ \text{sam.ep.r 2} \\ \text{msa.head 8} \end{bmatrix} * 8$
Stage 4	[B, 512, 8]	conv1d 3, 512-d, LN	conv1d 3, 512-d, LN
		$\begin{bmatrix} \text{dim 512} \\ \text{msa.head 16} \end{bmatrix} * 1$	$\begin{bmatrix} \text{dim 512} \\ \text{msa.head 16} \end{bmatrix} * 1$
AdaptiveAvgPool1d, Flatten, 1-d, Sigmoid			

Supplementary Fig. 14 | The AIX system presents a granularity attention pattern based on the transparent scale diffusion

mechanism. (a)-(c) showcase three typical examples with various transparent scales, where the model focuses on hypopnea/apnea patterns in Flow signals across four scales.

Supplementary Fig. 15 | The AIX system presents a granularity attention pattern based on the transparent scale diffusion mechanism. (a)-(c) showcase three typical examples with various transparent scales, where the model focuses on desaturation patterns in SpO2 signals across four different scales.

Supplementary Fig. 16 | The transparent visualization comparison of external test sample examples is presented (Flow signals). This includes the Grad-CAM results based on the last convolutional layer of ResNet50 (top row) and the visualizations of TSD-Net at four different scales (Scale1-4).

Supplementary Fig. 17 | The transparent visualization comparison of external test sample examples is presented (SpO2 signals). This includes the Grad-CAM results based on the last convolutional layer of ResNet50 (top row) and the visualizations of TSD-Net at four different scales (Scale1-4). The transparency scale diffusion mechanism of TSD-Net is shown to exhibit a hierarchical attention distribution from local details to global features, offering a finer feature representation capability compared to the Grad-CAM method, providing a multi-level interpretability perspective for the model's decision-making process.

[16] Ke, G., et al. Lightgbm: A highly efficient gradient boosting decision tree. *Advances in neural information processing systems* **30**(2017).

5. The model incorporates multiple modalities, including Flow, Chest, and SpO2 as gold standards. However, the specific features derived from each modality are not clearly described in the manuscript. Providing a detailed description of the exact feature values utilized in each modality would greatly enhance the transparency and reproducibility of the study.

Response: Thank you for this important question about feature extraction. We would like to clarify that our AIX system does not rely on hand-crafted features but instead employs a data-driven deep learning approach through our transparent scale diffusion network (TSD-Net). Specifically:

1) Rather than manually designing features, our TSD-Net automatically learns hierarchical representations directly from raw physiological signals. For each channel type (Flow, Chest, or SpO₂), the input is the raw signal resampled to shape $[B, 1, 256]$, where B represents the batch size.

2) The TSD-Net processes these raw signals through four sophisticated stages (detailed in Supplementary Fig. 3 and Supplementary Table 10):

- Stage 1-2: Multi-scale convolutional attention (MSCA) modules capture local respiratory patterns at different scales.
- Stage 3: Combined MSCA and transformer self-attention modules enable transition from local to global feature learning.
- Stage 4: Transformer self-attention modules capture long-range dependencies.

3) For single-channel analysis, TSD-Net directly outputs probability values for respiratory event detection. For multi-channel integration, each channel's TSD-Net serves as a feature extractor, automatically learning 8 abstract features (obtained by average pooling and flattening the shape $[B, 512, 8]$ output from Stage 4 to shape $[B, 8]$). Different workflows are illustrated in Supplementary Fig. 2.

4) This data-driven approach allows the model to automatically discover relevant patterns in the physiological signals, rather than relying on pre-defined features. The effectiveness of this approach is demonstrated by:

- The model's strong performance across multiple external validation cohorts.
- The interpretable visualization of learned patterns through our transparent scale diffusion mechanism (Supplementary Figs. 14-17).
- The ability to capture both local respiratory events and their long-term temporal relationships.
- Feature-level SHAP analysis demonstrating the relative importance and impact of different respiratory signal features in model prediction (Supplementary Fig. 11).
- Quantitative comparison of SHAP value magnitudes across respiratory channels (Supplementary Fig. 12) showing how different signal types contribute to the final diagnosis.

We have made our complete implementation, including the TSD-Net architecture and training procedures, publicly available at our GitHub repository (<https://github.com/fdu-harry/Apnea-Interact-Explainer>) to ensure full reproducibility of our approach.

In LightGBM Model for Channel Feature Integration (Page 19):

To further understand the contribution of different channel features in the integration process, we conducted SHAP (SHapley Additive exPlanations) analysis on the LightGBM model. As shown in Supplementary Fig. 12, the feature-level SHAP distribution reveals the relative importance and impact of different respiratory features extracted by TSD-Net. Each point represents a sample, with color indicating the feature value (red for high, blue for low) and horizontal position showing the SHAP value (positive

values indicate increased likelihood of respiratory events). The features are ordered by their mean absolute SHAP values, providing insight into which features most strongly influence the model's predictions across different channels.

Furthermore, we performed a quantitative comparison of SHAP value magnitudes across respiratory channels (Flow, Chest, and SpO₂) as presented in Supplementary Fig. 13. This channel-wise analysis demonstrates that SpO₂-derived features generally exhibit higher SHAP values (mean ± SD: 3.39 ± 0.02) compared to Flow (0.54 ± 0.02) and Chest (0.73 ± 0.01) features, aligning with clinical observations about the importance of oxygen desaturation in sleep apnea diagnosis. This analysis not only validates our feature integration strategy but also provides interpretable insights into how different channels contribute to the final prediction.

In Supplementary Material:

Supplementary Fig. 2 | The AIX system operates under home setting (single-channel) and professional setting (multi-channel) workflows. (a) Single-channel signals for the entire night are segmented into 30-second granularity and fed into the granularity prediction model (Model 1). This model provides detailed 30-second AI analysis decision logic based on a transparent scale diffusion mechanism. Subsequently, the granular probabilities of normal and respiratory events for the entire night are resampled to a length of 1024 to report overnight SA risk index (SARI) and train and predict with the AHI regression model (Model 2), resulting in the final AHI value. Two interpretability markers, 30-second transparent granularity attention and SARI, are obtained throughout the workflow. (b) Multi-channel signals for the entire night are split according to signal type, and a Model 1 is trained for each signal type. The trained Model 1 is used as a feature extractor to output 8 features per signal type for 30-second granularity. These features are concatenated to form 24 features. These features are trained using lightGBM to output multi-channel granular prediction probabilities. Finally, the resampled granular probabilities for the entire night are used for

AHI regression with Model 2. Similar to the AIX single-channel workflow, the multi-channel workflow outputs the SARI and the 30-second transparent granularity attention for each signal type.

Supplementary Fig. 3 | Overall structure of the granularity prediction model with individual constituent blocks. (a) General structure of the granularity prediction model, Transparent scale diffusion network (TSD-Net). (b) Composition of multi-scale convolutional attention (MSCA) block. (c) Multi-head convolutional encoder (MHCE) module. (d) Cross-scale encoding fusion (CSEF) module. (e) Squeeze-and-excitation network (SE-Net). (f) Specific composition of MSCA-MHSA (multi-head self-attention) block.

Supplementary Table 10 | Details of the parameterization of the transparent scale diffusion network (TSD-Net). TSD-Net-T and TSD-Net-B represent the Tiny and Base versions of TSD-Net, respectively; "dim 64": An embedding dimension of 64; "sam.head 4": The multi-head convolutional encoder (MHCE) module with 4 heads; "sam.ep.r 2": An expansion rate of 2 for the cross-scale encoding fusion (CSEF) module; "msa.head 8": The multi-head self-attention (MHSA) module with 8 heads.

Layer name	Output Size [B, C, L]	TSD-Net-T	TSD-Net-B
Stem	[B, 64, 64]	conv1d 2, 64-d, BN	conv1d 2, 64-d, BN
Stage 1	[B, 64, 64]	conv1d 2, 64-d, LN	conv1d 2, 64-d, LN
		$\begin{bmatrix} \text{dim 64} \\ \text{sam.head 4} \\ \text{sam.ep.r 2} \end{bmatrix} * 1$	$\begin{bmatrix} \text{dim 64} \\ \text{sam.head 4} \\ \text{sam.ep.r 2} \end{bmatrix} * 2$
Stage 2	[B, 128, 32]	conv1d 3, 128-d, LN	conv1d 3, 128-d, LN
		$\begin{bmatrix} \text{dim 128} \\ \text{sam.head. 4} \\ \text{sam.ep r. 2} \end{bmatrix} * 1$	$\begin{bmatrix} \text{dim 128} \\ \text{sam.head. 4} \\ \text{sam.ep r. 2} \end{bmatrix} * 2$
Stage 3	[B, 256, 16]	conv1d 3, 256-d, LN	conv1d 3, 256-d, LN
		$\begin{bmatrix} \text{dim 256} \\ \text{sam.head 4} \\ \text{sam.ep.r 2} \\ \text{msa.head 8} \end{bmatrix} * 1$	$\begin{bmatrix} \text{dim 256} \\ \text{sam.head 4} \\ \text{sam.ep.r 2} \\ \text{msa.head 8} \end{bmatrix} * 8$
Stage 4	[B, 512, 8]	conv1d 3, 512-d, LN	conv1d 3, 512-d, LN
		$\begin{bmatrix} \text{dim 512} \\ \text{msa.head 16} \end{bmatrix} * 1$	$\begin{bmatrix} \text{dim 512} \\ \text{msa.head 16} \end{bmatrix} * 1$
AdaptiveAvgPool1d, Flatten, 1-d, Sigmoid			

Supplementary Fig. 11 | Feature-level SHAP analysis of respiratory signals in model prediction. Feature-wise SHAP values showing the impact of individual features on model predictions. The color represents the feature value (blue: low, red: high), and the horizontal position shows the SHAP value impact on model output. Features are ordered by their absolute SHAP

values.

Supplementary Fig. 12 | Quantitative comparison of SHAP value magnitudes across respiratory channels. Comparison of total SHAP value magnitudes across different signal channels (Flow, Chest, and SpO₂). Values are presented as mean ± standard deviation, demonstrating the relative importance of each channel type in the model's decision-making process.

6. This paper emphasizes interpretability through its ‘transparent scale diffusion mechanisms’. The claim of multi-level expert-logic interpretable visualization is compelling, but the authors did not elaborate how this interpretable framework has been evaluated or validated in the manuscript. It lacks the use of widely recognized explainable AI methods such as Grad-CAM. In fact, techniques like Grad-CAM would provide visualizations of attention over input signals, making it easier for clinicians to understand why specific parts of the data influenced the model’s precisions. Although the authors claimed to provide traceable event-level logic, they did not sufficiently explain how this was achieved. The methodology for generating granular explanation still remains vague. More clear description of how event-level decisions (e.g. apnea detection in a 30-sec window) are derived along with concrete examples or visualizations, is necessary to substantiate this claim. How was “trust” in the AI system measured among end-users such as clinicians? In the manuscript, it should be addressed properly about demonstrating use-centric outcomes beyond performance metrics. This would enhance the system’s transparency and clinical implications.

Response: We appreciate the reviewer's valuable feedback regarding the interpretability evaluation. We have substantially enhanced our manuscript to address these concerns:

1) **Quantitative Validation of Interpretability:** We conducted a comprehensive evaluation of our interpretability framework through systematic comparison with Grad-CAM¹⁷ across four external validation datasets (SHHS2, MESA, MROS, and CFS). For each dataset, we randomly selected 50 overnight recordings (total n=200) with emphasis on 30-second windows containing positive respiratory events accompanied by expert second-by-second annotations. To ensure fair comparison, both TSD-Net

and ResNet50 were trained with identical configurations including early stopping criteria. The attention matrices from all four stages of TSD-Net were compared against Grad-CAM attention derived from a standard 1D ResNet50 network. All attention matrices were normalized and thresholded at 0.5 to generate high-confidence regions.

Quantitative assessment using Intersection over Union (IoU) metrics demonstrated TSD-Net's superior performance, with Stage 3 achieving mean IoU scores of 0.253-0.356 for Flow and 0.228-0.323 for SpO₂ across datasets, consistently outperforming Grad-CAM (mean IoU of Flow: 0.145-0.273, mean IoU of SpO₂: 0.140-0.258). Furthermore, TSD-Net exhibited robust attention distribution across all stages, with Flow channel maintaining stable high IoU scores (SHHS2: 0.315-0.330, MESA: 0.227-0.253, MROS: 0.346-0.353, CFS: 0.328-0.360) and SpO₂ channel showing progressive improvement from Stage 1 to Stage 4 (SHHS2: 0.218-0.328, MESA: 0.175-0.244, MROS: 0.207-0.340, CFS: 0.257-0.398). Statistical analysis confirmed that each stage of TSD-Net significantly outperformed Grad-CAM across all datasets ($P < 0.001$ except for one comparison), as detailed in Supplementary Table 11.

We have added comprehensive quantitative validation results in the Methods section 'Details and Advantages of Transparent Scale Diffusion Technology' on Pages 18-19 and included detailed performance comparisons in Supplementary Fig. 18 and Supplementary Table 11.

In Details and Advantages of Transparent Scale Diffusion Technology (Pages 18-19):

To comprehensively evaluate our interpretability framework, we conduct systematic comparisons between TSD-Net and the widely-used Grad-CAM¹⁷ method implemented on a 1D ResNet50 backbone across four external validation datasets (SHHS2, MESA, MROS, and CFS). From each dataset, we randomly select 50 overnight recordings (total $n=200$), with particular emphasis on analyzing 30-second windows containing positive respiratory events accompanied by expert second-by-second annotations. Both TSD-Net and ResNet50 are trained with identical configurations including early stopping criteria to ensure fair comparison. The attention matrices from all four stages of TSD-Net and Grad-CAM attention are normalized and thresholded at 0.5 to generate high-confidence regions. Quantitative assessment using Intersection over Union (IoU) metrics (Supplementary Fig. 18 and Supplementary Table 11) demonstrates TSD-Net's superior performance, with Stage 3 achieving mean IoU scores of 0.253-0.356 for Flow and 0.228-0.323 for SpO₂ across datasets, consistently outperforming Grad-CAM (mean IoU of Flow: 0.145-0.273, mean IoU of SpO₂: 0.140-0.258), providing solid decision-making basis for human interactive review.

The core advantage of transparent scale diffusion technology lies in its visualization capability based on multi-scale attention matrices. As demonstrated in Supplementary Figs. 16-18, and Supplementary Table 11, TSD-Net exhibits robust attention distribution across all stages, with Flow channel maintaining stable high IoU scores (SHHS2: 0.315-0.330, MESA: 0.227-0.253, MROS: 0.346-0.353, CFS: 0.328-0.360) through different stages and SpO₂ channel showing progressive improvement from Stage 1 to Stage 4 (SHHS2: 0.218-0.328, MESA: 0.175-0.244, MROS: 0.207-0.340, CFS: 0.257-0.398). Statistical analysis shows that each stage of TSD-Net significantly outperforms Grad-CAM across all datasets ($p < 0.001$ except for one comparison, Supplementary Table 1), demonstrating its superior feature representation capability. By average pooling the attention matrices output at each scale and projecting them onto the signal length to form heatmaps, the system provides multi-level interpretability perspective for the model's decision-making process, preserving richer and more detailed transparent explanatory power for physiological signal analysis.

In Supplementary Material:

Supplementary Fig. 18 | Evaluation of XAI methods' interpretability and alignment with clinical event detection. (a) Representative examples demonstrating the alignment between attention heatmaps and expert-annotated respiratory events for Flow and SpO₂ signals. The original signals (top) are overlaid with attention heatmaps, where darker colors indicate higher attention weights. Expert annotations (middle) show the temporal locations of respiratory events (red for apnea, cyan for hypopnea). The IoU scores (bottom) quantify the overlap between model attention and expert annotations, with cases showing both low and high IoU scenarios. (b) Quantitative comparison of IoU scores across four sleep databases (SHHS2, MESA, MROS, and CFS). The violin plots show IoU score distributions for four TSD-Net stages and Grad-CAM on Flow (top) and SpO₂ (bottom) channels. Black dots and lines represent means and standard deviations. P-values from two-tailed Mann-Whitney U tests indicate statistical significance between adjacent methods.

Supplementary Table 11 | Quantitative comparison of XAI methods using Intersection over Union (IoU) scores and statistical significance across four sleep databases (SHHS2, MESA, MROS, and CFS). IoU values are presented as Mean ± Standard Deviation and Median. P-values are calculated using two-tailed Mann-Whitney U tests between pairwise TSD-Net stages and between TSD-Net stages and Grad-CAM.

Database	Flow	TSD-Net				Grad-CAM
		Stage 1	Stage 2	Stage 3	Stage 4	
SHHS2	IoU (Mean±Std; Median)	0.330±0.111; 0.333	0.315±0.106; 0.320	0.324±0.121; 0.330	0.321±0.101; 0.335	0.178±0.129; 0.168
	p-value	Stage 1 vs Stage 2: 3.236e-01; Stage 1 vs Stage 3: 7.611e-01 Stage 1 vs Stage 4: 5.796e-01; Stage 2 vs Stage 3: 5.011e-01 Stage 2 vs Stage 4: 6.083e-01; Stage 3 vs Stage 4: 7.887e-01				Grad-CAM vs Block 1: 4.450e-42 Grad-CAM vs Block 3: 4.264e-38 Grad-CAM vs Block 11: 2.761e-38 Grad-CAM vs Block 12: 5.192e-41
MESA	IoU (Mean±Std; Median)	0.243±0.114; 0.238	0.227±0.118; 0.217	0.253±0.118; 0.249	0.240±0.128; 0.245	0.145±0.124; 0.127
	p-value	Stage 1 vs Stage 2: 7.232e-02; Stage 1 vs Stage 3: 2.723e-01 Stage 1 vs Stage 4: 9.898e-01; Stage 2 vs Stage 3: 5.142e-03 Stage 2 vs Stage 4: 1.140e-01; Stage 3 vs Stage 4: 2.926e-01				Grad-CAM vs Block 1: 3.840e-43 Grad-CAM vs Block 3: 1.112e-31 Grad-CAM vs Block 11: 1.926e-49 Grad-CAM vs Block 12: 6.792e-37
MROS	IoU (Mean±Std; Median)	0.353±0.121; 0.350	0.348±0.125; 0.343	0.346±0.122; 0.354	0.346±0.103; 0.365	0.204±0.160; 0.181
	p-value	Stage 1 vs Stage 2: 4.702e-01; Stage 1 vs Stage 3: 5.322e-01 Stage 1 vs Stage 4: 9.803e-01; Stage 2 vs Stage 3: 9.438e-01 Stage 2 vs Stage 4: 5.061e-01; Stage 3 vs Stage 4: 6.433e-01				Grad-CAM vs Block 1: 2.043e-80 Grad-CAM vs Block 3: 2.663e-75 Grad-CAM vs Block 11: 8.895e-75 Grad-CAM vs Block 12: 1.180e-79
CFS	IoU (Mean±Std; Median)	0.360±0.136; 0.370	0.328±0.117; 0.336	0.356±0.140; 0.357	0.340±0.125; 0.356	0.273±0.165; 0.278
	p-value	Stage 1 vs Stage 2: 5.433e-02; Stage 1 vs Stage 3: 7.000e-01 Stage 1 vs Stage 4: 2.523e-01; Stage 2 vs Stage 3: 1.454e-01 Stage 2 vs Stage 4: 4.039e-01; Stage 3 vs Stage 4: 5.063e-01				Grad-CAM vs Block 1: 6.310e-08 Grad-CAM vs Block 3: 4.006e-04 Grad-CAM vs Block 11: 6.283e-07 Grad-CAM vs Block 12: 1.599e-05
Database	SpO ₂	TSD-Net				Grad-CAM
		Stage 1	Stage 2	Stage 3	Stage 4	
SHHS2	IoU (Mean±Std; Median)	0.218±0.105; 0.216	0.286±0.130; 0.285	0.283±0.135; 0.277	0.328±0.154; 0.333	0.140±0.103; 0.125
	p-value	Stage 1 vs Stage 2: 1.130e-59; Stage 1 vs Stage 3: 7.005e-51 Stage 1 vs Stage 4: 1.106e-116; Stage 2 vs Stage 3: 4.343e-01 Stage 2 vs Stage 4: 2.370e-19; Stage 3 vs Stage 4: 3.825e-21				Grad-CAM vs Block 1: 1.640e-100 Grad-CAM vs Block 3: 4.667e-213 Grad-CAM vs Block 11: 3.725e-199 Grad-CAM vs Block 12: 3.534e-252
MESA	IoU (Mean±Std; Median)	0.175±0.098; 0.169	0.231±0.114; 0.226	0.228±0.126; 0.222	0.244±0.143; 0.239	0.141±0.113; 0.129
	p-value	Stage 1 vs Stage 2: 2.882e-40; Stage 1 vs Stage 3: 5.848e-31 Stage 1 vs Stage 4: 6.391e-42; Stage 2 vs Stage 3: 3.793e-01 Stage 2 vs Stage 4: 1.691e-02; Stage 3 vs Stage 4: 2.425e-03				Grad-CAM vs Block 1: 1.452e-19 Grad-CAM vs Block 3: 8.649e-80 Grad-CAM vs Block 11: 1.792e-66 Grad-CAM vs Block 12: 1.265e-74
MROS	IoU (Mean±Std; Median)	0.207±0.112; 0.199	0.292±0.142; 0.284	0.282±0.142; 0.280	0.340±0.172; 0.344	0.141±0.107; 0.125
	p-value	Stage 1 vs Stage 2: 1.351e-90; Stage 1 vs Stage 3: 7.417e-76 Stage 1 vs Stage 4: 2.169e-161; Stage 2 vs Stage 3: 7.476e-02 Stage 2 vs Stage 4: 1.079e-24; Stage 3 vs Stage 4: 4.145e-33				Grad-CAM vs Block 1: 4.038e-86 Grad-CAM vs Block 3: 7.766e-260 Grad-CAM vs Block 11: 4.595e-234 Grad-CAM vs Block 12: 3.944e-307
CFS	IoU (Mean±Std; Median)	0.257±0.121; 0.256	0.320±0.139; 0.321	0.323±0.144; 0.328	0.398±0.179; 0.431	0.258±0.140; 0.258
	p-value	Stage 1 vs Stage 2: 9.361e-39; Stage 1 vs Stage 3: 2.300e-42 Stage 1 vs Stage 4: 3.414e-129; Stage 2 vs Stage 3: 3.729e-01 Stage 2 vs Stage 4: 7.396e-49; Stage 3 vs Stage 4: 3.338e-44				Grad-CAM vs Block 1: 8.444e-01 Grad-CAM vs Block 3: 9.880e-32 Grad-CAM vs Block 11: 1.031e-34 Grad-CAM vs Block 12: 2.330e-122

2) Event-Level Logic Implementation: Our event-level logic operates on 30-second windows, aligning with clinical sleep staging conventions¹⁵. For each 30-second target window, we incorporate 120 seconds of preceding and following signals, creating a 270-second contextual segment where the label corresponds to the central 30-second window. This approach is well-established in sleep apnea detection research, as surrounding signals provide crucial contextual information. As emphasized in previous literature, severe respiratory events typically persist for 30-45 seconds⁸, making contextual information vital for model training and classification¹⁸.

To achieve granular interpretability, our model generates explanations through attention visualization within each 30-second window, highlighting specific regions of abnormal breathing patterns. As demonstrated in Supplementary Fig. 18(a), the attention heatmaps directly highlight regions of interest in both Flow and SpO₂ signals that correspond to expert-annotated respiratory events, enabling traceable event-level interpretation of the model's decision process. This visualization allows direct comparison between model attention and expert second-by-second annotations, providing concrete evidence of our model's ability to focus on clinically relevant signal regions.

We have added detailed explanations of our event-level logic implementation and granular interpretability approach in Results Section 'AIX for AHI Prediction and Transparent Granular Traceability' on Page 4 of the manuscript.

In AIX for AHI Prediction and Transparent Granular Traceability (Page 4):

Following clinical sleep staging conventions¹⁵, the granularity prediction model (Model 1) makes event-level decisions by classifying each 30-second window, while incorporating 120-second preceding and following signals to form a 270-second contextual segment. Since severe respiratory events typically persist for 30-45 seconds⁸ and require temporal context for accurate detection, this design ensures that while the classification target remains the central 30-second window, the model has sufficient context to detect respiratory events that may influence the target window. The model provides granular explanations through attention visualization within each 30-second window, highlighting specific regions of abnormal breathing patterns. As demonstrated in Supplementary Fig. 18(a), the attention heatmaps directly highlight regions of interest in both Flow and SpO₂ signals that correspond to expert-annotated respiratory events, providing traceable event-level interpretation of the model's decision process.

3) Trust Assessment Framework: We have implemented a comprehensive, data-driven approach to quantitatively measure and enhance clinician trust in our AI system through three key mechanisms:

a) Uncertainty Quantification: Our system quantifies prediction uncertainty through the absolute probability difference between normal and abnormal classifications for each 30-second segment. Statistical analysis (Fig. 5e) reveals that correct predictions (TN/TP) consistently show high confidence with median values above 0.9, while incorrect predictions (FN/FP) exhibit significantly lower confidence with median values around 0.25 ($P < 0.001$). This clear separation provides users with reliable indicators for verifying AI predictions.

b) Efficient Review Targeting: We systematically analyzed the relationship between interaction thresholds and error rates (Fig. 6a-c, Supplementary Table 8) to optimize the trust-efficiency trade-off. For example, with an interaction threshold interval of 0.1-0.2, 3.66% of cases were identified for review, with corresponding FP rate of 0.96% and FN rate of 0.22%. This analysis illustrates how interaction thresholds can be leveraged to balance review efficiency and error control.

c) Progressive Performance Validation: To validate this trust-building mechanism, we conducted

extensive experiments with varying review coverage (Supplementary Table 9). Results show that with an interaction threshold of 0.00-0.10, reviewing 10.53% of cases improved accuracy from 0.867 to 0.911. When expanding the threshold to 0.00-0.45, reviewing 20.40% of cases achieved even higher performance levels of ACC: 0.939, SEN: 0.966, SPE: 0.929. This progressive improvement demonstrates that the system can achieve higher reliability through strategic human-AI collaboration.

The effectiveness of this framework is reflected in its practical implementation. Users can verify the system's decision-making process through attention visualization, adjust interaction thresholds based on their trust preferences, and observe quantifiable performance improvements through collaborative review. This combination of transparent uncertainty quantification, efficient review targeting, and validated performance gains provides users with concrete evidence for building trust in the system while maintaining workflow efficiency.

We have expanded the *Methods* section 'Human-AI Collaboration Feasibility of AIX' on Page 9 to include detailed documentation of these trust-building mechanisms and their quantitative validation.

In Human-AI Collaboration Feasibility of AIX (Page 9):

This quantifiable uncertainty pattern serves as the foundation of our trust assessment framework, providing users with reliable indicators for identifying cases requiring verification. The significant separation between confident and uncertain predictions enables a data-driven approach to building trust in the system's decision-making process. To systematically evaluate the system's trust-building capabilities through interactive review, we conducted experiments on both FDU-HSH retrospective ($n=327$) and prospective ($n=265$) cohorts using SpO₂ monitoring (Supplementary Fig. 31, Supplementary Tables 8,9). Taking the FDU-HSH retrospective cohort as an example, we examined the relationship between interaction threshold intervals and error rates (Fig. 6a-c) to understand their trade-off patterns. For instance, with an interaction threshold interval of 0.1-0.2, we observed that 3.66% of samples were identified for review, with corresponding FP rate of 0.96% and FN rate of 0.22% (Supplementary Table 8). This illustrates how interaction thresholds can be leveraged to balance review efficiency and error control.

To validate the effectiveness of this trust-building framework, we conducted progressive performance validation experiments through selective review. The results demonstrated substantial improvements in diagnostic reliability. When reviewing 10.53% of cases (interaction threshold: 0.00-0.10), the binary event classification metrics improved from ACC=0.867, SEN=0.914, SPE=0.851 to ACC=0.911, SEN=0.945, SPE=0.899. Increasing review coverage to 20.40% (interaction threshold: 0.00-0.45) achieved even higher performance levels of ACC=0.939, SEN=0.966, SPE=0.929 (Supplementary Table 9). This comprehensive trust assessment framework, combining uncertainty quantification, efficient review targeting, and progressive performance validation, enables users to build trust through transparent interaction while maintaining workflow efficiency. The framework provides concrete evidence of AIX system's capability to achieve higher reliability through strategic human-AI collaboration, as demonstrated by the quantifiable performance improvements under different manual review coverage ratios.

- [8] Levy, J., Álvarez, D., Del Campo, F. & Behar, J.A. Deep learning for obstructive sleep apnea diagnosis based on single channel oximetry. *Nature Communications* **14**, 4881 (2023).
- [15] Berry, R.B., *et al.* Rules for scoring respiratory events in sleep: update of the 2007 AASM manual for the scoring of sleep and associated events: deliberations of the sleep apnea definitions task force of the American Academy of Sleep Medicine. *Journal of clinical sleep medicine* **8**, 597-619 (2012).
- [17] Selvaraju, R.R., *et al.* Grad-cam: Visual explanations from deep networks via gradient-based localization. in

Proceedings of the IEEE international conference on computer vision 618-626 (2017).

- [18] Hu, S., Wang, Y.n., Liu, J. & Yang, C. Personalized Transfer Learning for Single-Lead ECG-Based Sleep Apnea Detection: Exploring the Label Mapping Length and Transfer Strategy Using Hybrid Transformer Model. *Ieee Transactions on Instrumentation and Measurement* 72(2023).
-

7. The authors also claim that the system is multimodal because it integrates Flow, Chest, and SpO2 signals. However, these signals are all subsets of the same physiological modality (e.g. respiratory monitoring). Could it be possible to add multimodal, which generally refers to integrating heterogeneous data types, such as text, images, or audio? Since these kinds of multimodal datasets only use physiological signals, the system lacks the diversity needed to fully justify the multimodal claim. To concretely establish the multimodality argument, the utility of the model would need to be enhanced by adding tabular data (e.g. patient demographics, medical history) and/or imaging data (e.g. EEG figures or waveform patterns).

Response: We appreciate the reviewer's insightful comment regarding the terminology. We acknowledge our imprecise use of "multimodal" and have revised it throughout the manuscript and supplementary materials to "multi-channel", which better aligns with the conventional terminology used in clinical polysomnography monitoring.

Following the reviewer's suggestion about integrating heterogeneous data types, we have conducted comprehensive experiments incorporating demographic features (age, gender, BMI, and race) with physiological signals. Our investigation revealed an important finding about the relationship between dynamic physiological patterns and static demographic risk factors. As detailed in our newly added Methods section 'Demographic Auxiliary Value Assessment in Overnight AHI Regression' on Page 21, we developed two distinct integration approaches at the whole-night level (Supplementary Fig. 32): (1) A feature concatenation method where demographic features are encoded and directly concatenated with the probability sequence features, and (2) A cross-attention mechanism where probability sequence features serve as queries while demographic features act as keys and values for dynamic feature enhancement. Our experimental results on SHHS2, MROS, and CFS datasets (Supplementary Figs. 33,34) revealed that while both integration approaches achieved reasonable performance (For single-SpO₂ channel, concatenation: 0.7633/0.7752/0.7465 accuracy, cross-attention: 0.7769/0.7687/0.7604 accuracy), the AIX system using only physiological signals actually achieved optimal performance (accuracies of 0.7831/0.7851/0.7743). This phenomenon can be attributed to several factors: 1) Physiological signals are dynamic features that directly capture the temporal evolution of sleep apnea events, while demographic features are static and may introduce redundancy when combined with already-rich temporal information; 2) The continuous physiological signals inherently encode patient-specific characteristics through their unique patterns and variations; 3) The integration of static demographic features might potentially dilute the fine-grained temporal patterns learned from physiological signals, leading to slightly decreased performance. These findings are further supported by consistently higher R-squared values (0.9526/0.9551/0.9334) and ICC coefficients (0.9514/0.9540/0.9311) in the AIX system without demographic features.

These findings suggest that while demographic features provide valuable population-level risk assessment, their optimal integration with physiological signals requires more sophisticated approaches beyond direct feature fusion. This led us to explore alternative paradigms such as our CLIP-inspired

framework for representation learning, which better preserves the complementary nature of these different data types.

In Supplementary Material:

Supplementary Fig. 32 | Overview of the demographic-assisted AHI regression framework with two distinct integration approaches. Left: Initial overnight probability sequence generation using a sliding window (30-sec) approach. Multiple physiological signal windows are processed through a TSD-Net to generate event probabilities, which are then interpolated to form a continuous overnight probability sequence distinguishing between normal and respiratory events. Right: Two proposed methods for integrating demographic information into AHI regression: (1) A feature-concatenation approach where demographic features are encoded and directly concatenated with the probability sequence features before AHI prediction. (2) A cross-attention mechanism where the probability sequence features serve as queries (Q) while demographic features act as keys (K) and values (V), enabling dynamic demographic-aware feature enhancement for AHI prediction. Patient demographics including Gender, Age, BMI, and Race are processed through a dedicated demographic encoder in both approaches. The framework demonstrates two distinct strategies for leveraging patient-specific demographic information to enhance AHI regression performance.

Supplementary Fig. 33 | Comparison of three AHI regression approaches using multi-channel Gold signals: Concatenation (demographic-signal feature concatenation), Cross-attention (demographic-signal attention-based feature fusion), and AIX (Without demographic features) across SHHS2, MROS, and CFS datasets. Performance evaluation includes regression analysis (left column, with R-squared values and P-values), Bland-Altman plots (middle column, showing mean differences with 95% limits of agreement), and confusion matrices (right column, displaying four-level severity classification). The bottom panel presents comparative metrics including ACC, Mean-F1, R-squared, ICC, and MAE values across all methods and datasets.

Supplementary Fig. 34 | Comparison of three AHI regression approaches using single-channel SpO₂ signals: Concatenation (demographic-signal feature concatenation), Cross-attention (demographic-signal attention-based feature fusion), and AIX (Without demographic features) across SHHS2, MROS, and CFS datasets. Performance evaluation includes regression analysis (left column, with R-squared values and P-values), Bland-Altman plots (middle column, showing mean differences with 95% limits of agreement), and confusion matrices (right column, displaying four-level severity classification). The bottom panel presents comparative metrics including ACC, Mean-F1, R-squared, ICC, and MAE values across all methods and datasets.

In Demographic Auxiliary Value Assessment in Overnight AHI Regression (Page 21):

To investigate the potential auxiliary value of demographic features in SA assessment, we first explore their integration at the sequence level, hypothesizing that demographic risk factors might modulate the overall overnight respiratory patterns. We design two distinct approaches (Supplementary Fig. 32), operating on 1024-length sequences interpolated from 30-second window probabilities. The first approach

employs a feature concatenation method, where demographic features (including gender, age, BMI, and race) are first processed through a demographic encoder (transforming from 4-dimensional input to 128-dimensional features through two fully connected layers), while the probability sequences are processed through three convolutional layers (with filter sizes of 5, 3, and 1, and channel dimensions of 16, 32, and 64) followed by max-pooling operations, as detailed in Supplementary Fig. 4. The flattened sequence features (8×1024) are then concatenated with demographic features (128) before final AHI regression through two fully connected layers. The second approach utilizes a cross-attention mechanism, where the same convolutional architecture processes probability sequences into features of shape (B, 64, 128), while demographic features are mapped to 64 channels through a 1×1 convolution. A multi-head attention module (4 heads) then uses sequence features as queries and demographic features as keys and values for dynamic feature enhancement.

Taking the single-channel SpO₂ configuration as an example (Supplementary Fig. 34), experiments on SHHS2, MROS, and CFS datasets show that both demographic feature integration approaches fail to improve performance compared to the baseline system. The concatenation approach achieves accuracies of 0.7633/0.7752/0.7465 and macro F1 scores of 0.7714/0.7564/0.7449, while the cross-attention method yields accuracies of 0.7769/0.7687/0.7604 and macro F1 scores of 0.7821/0.7259/0.7525, both underperforming the baseline AIX system (accuracies: 0.7831/0.7851/0.7743, macro F1: 0.7883/0.7672/0.7750). The exceptional reliability of sequence probability features, validated by high R-squared values (0.9526/0.9551/0.9334) and ICC coefficients (0.9514/0.9540/0.9311), suggests that direct integration of static demographic features with dynamic sequence probabilities might not be optimal, leading us to explore more fundamental representation learning strategies.

Regarding the incorporation of EEG and video signals, while we acknowledge their potential value, our focus on specific physiological signals stems from practical clinical considerations. Both HSAT and portable wearable devices aim to balance screening accuracy, comfort, and accessibility. While EEG provides deeper insights into brain activity and sleep analysis, it requires complex monitoring configurations that may not be practical for large-scale screening. As for video monitoring, there are two main limitations: 1) Privacy concerns, as continuous overnight video recording in home settings raises significant privacy issues; 2) Storage requirements, as high-quality video data from entire nights of sleep would demand substantial storage capacity, making it impractical for our target scenario of home-based long-term monitoring.

Furthermore, our signal-based approach offers significant advantages in terms of data privacy and anonymization. Unlike video or image data that may contain identifiable personal information, physiological signals can be more easily anonymized while retaining their clinical value. This design choice was made after thorough discussions with clinical experts, who emphasized the importance of patient privacy in sleep monitoring systems.

In clinical practice, HSAT devices and wearable monitoring systems typically contain only signals related to respiratory disorders, which has become increasingly accepted in many clinical settings for initial screening and home monitoring. We have addressed these concerns in the Discussion Section (Page 12) by adding a comprehensive analysis of monitoring channel selection and performance evaluation.

In Discussion (Page 12):

Our comprehensive analysis of demographic feature integration reveals intriguing insights into the relationship between physiological signals and patient characteristics in SA assessment. Through extensive

experiments with both feature concatenation and cross-attention mechanisms (Supplementary Fig. 32), we find that continuous physiological signals alone achieve optimal diagnostic performance across datasets. The AIX system demonstrates robust performance with Gold standard channels (ACC=0.7951-0.8057, Macro F1=0.7853-0.8083) and maintains strong diagnostic capability even when simplified to single-channel SpO₂ configuration (ACC=0.7743-0.7851, Macro F1=0.7672-0.7883). As demonstrated in Supplementary Figs. 33,34, both configurations maintain high performance metrics without demographic features, with R² values (0.9254-0.9579 for Gold, 0.9334-0.9551 for SpO₂) and ICC coefficients (0.9225-0.9570 for Gold, 0.9311-0.9540 for SpO₂) consistently outperforming the demographic-assisted approaches (single-channel SpO₂). Regarding the choice of monitoring channels, our current focus on specific physiological signals is driven by practical considerations for widespread clinical adoption. While EEG provides valuable information for detailed sleep staging and comprehensive sleep analysis, its complex setup procedures and requirement for expert interpretation make it less suitable for large-scale screening programs, particularly in home settings. Similarly, video monitoring, while informative, raises significant privacy concerns and poses substantial data storage challenges that could impede long-term monitoring capabilities. These practical limitations, combined with our findings on the robust performance of simplified physiological monitoring, support our strategic approach of optimizing the balance between diagnostic accuracy and implementation feasibility for initial screening purposes.

8. The authors did not utilize state-of-the-art multimodal learning methods. Approaches such as CLIP or SimCLR, which learn shared representations across multiple modalities, can be applied to align different types of physiological signals with patient information or images.

Response: We appreciate the reviewer's suggestion regarding advanced multimodal learning methods. Our study explores both fully-supervised learning and self-supervised alignment approaches, demonstrating their complementary strengths in different scenarios. We have systematically investigated this direction by adding a new subsection 'CLIP-Inspired Framework for Fine-Grained Feature Alignment' in the Methods section on Pages 21-22 and conducting extensive experiments, which are detailed in Supplementary Figs. 35,36.

In Supplementary Material:

Architecture for Cross-Modal Alignment of Physiological Multi-channel Signals and Demographics

Supplementary Fig. 35 | A CLIP-inspired architecture for cross-modal alignment of physiological multi-channel signals and demographics. The architecture consists of four parallel encoding paths: three signal encoders (Flow, Chest, and SpO₂) for processing physiological signals, and a demographic encoder for patient characteristics (Gender, Age, BMI, Race). Each signal passes through its respective encoder to generate feature embeddings (F_N, C_N, S_N), while demographics are encoded into D_N features. The framework implements two types of CLIP-based learning strategies: cross-modal CLIP between signals and demographics, and cross-channel CLIP among physiological signals which includes pairwise alignment and leave-one-out alignment patterns (as shown in panel 1 and 2). The architecture enables transfer learning by freezing the pretrained encoders and fine-tuning only the classification head for downstream tasks.

Supplementary Fig. 36 | Performance comparison of the proposed cross-modal alignment framework. (Top) ROC curves comparing our CLIP-inspired framework with different fine-tuning ratios (1%, 5%, 10%, 20%, 50%, and 100%) of the complete SHHS1 contrastive learning training data across four channel configurations (Flow, Chest, SpO₂, and Gold standard combination), against the supervised TSD-Net. The evaluation was performed on 400,000 test samples (100,000 from each of SHHS2, MESA, MROS, and CFS databases). While our self-supervised approach shows promising results, the supervised TSD-Net achieved better performance across all configurations. (Bottom) t-SNE visualization of feature embeddings before and after applying the proposed CLIP-based architecture.

1) Framework Design and Implementation: In addition to our fully-supervised TSD-Net framework, we developed a CLIP-inspired alignment framework at the 30-second segment level. This framework comprises parallel encoding paths using the same TSD-Net architecture as encoders for Flow, Chest, and SpO₂ signals, each producing 64-dimensional embeddings, and a dedicated demographic encoder processing 4-dimensional features. We implemented multiple alignment strategies including pairwise physiological

signal alignment between different channels (Flow-Chest, Flow-SpO₂, Chest-SpO₂), leave-one-out alignment, and signal-demographic alignment, trained on a large-scale dataset of 3,779,943 segments (2,631,479 normal and 1,148,464 respiratory events).

2) Performance Analysis: Our evaluation (Supplementary Fig. 36) revealed distinct advantages of both approaches. While our fully-supervised TSD-Net framework demonstrated superior performance with sufficient labeled data (Flow: AUC 0.7853, Chest: 0.8147, SpO₂: 0.7982, Gold standard: 0.8202), the CLIP-inspired framework showed remarkable robustness in limited-data scenarios, maintaining stable performance (Flow: 0.6046-0.7535, Chest: 0.7065-0.7654, SpO₂: 0.6647-0.7645, Gold standard: 0.7731-0.7751) even with limited labeled data.

3) Methodological Advantages: Our experiments demonstrate the complementary strengths of both approaches. The fully-supervised TSD-Net framework excels in scenarios with abundant labeled data, effectively capturing temporal dependencies and channel-specific patterns in physiological signals. Meanwhile, the CLIP-inspired framework shows particular advantages in scenarios with limited labeled data, successfully learning robust representations through its contrastive learning mechanism. This aligns with CLIP's proven strength in leveraging large-scale unlabeled data while maintaining effectiveness with minimal labeled samples - a crucial advantage in medical scenarios where labeled data is often scarce.

The t-SNE visualization of our CLIP-inspired framework's learned representations (Supplementary Fig. 35) reveals two key findings: (1) excellent alignment between different physiological signal channel clusters, demonstrating the framework's effectiveness in learning consistent cross-channel representations of respiratory events, and (2) persistent separation between physiological and demographic feature clusters despite contrastive learning. This separation likely stems from the fundamentally different nature of these features - physiological signals capture dynamic, time-varying patterns of respiratory events that share common characteristics across channels, while demographic features represent static, population-level risk factors that influence disease susceptibility but don't directly reflect event-level respiratory dynamics. This inherent distinction in feature temporality and granularity explains why demographic features maintain their unique embedding space even after contrastive alignment, suggesting that effective integration of these complementary information sources may require more sophisticated fusion strategies beyond simple feature alignment.

This distinct pattern suggests both frameworks have their unique strengths: supervised learning for optimal performance with sufficient labels, and self-supervised alignment for robust representation learning with limited labels. This complementarity enables robust performance across varied clinical scenarios, from detailed multi-channel analysis to simplified monitoring configurations. We have added a dedicated subsection 'CLIP-Inspired Framework for Fine-Grained Feature Alignment' in the Methods section on Pages 21-22:

In CLIP-Inspired Framework for Fine-Grained Feature Alignment (Pages 21-22):

To address the challenge of effectively incorporating demographic information, we develop a CLIP¹⁹-inspired alignment framework operating at the 30-second segment level. This shift from sequence-level integration to representation learning was motivated by the observation that demographic features, while not directly beneficial for sequence-level prediction, might better serve as conditioning factors for learning more discriminative respiratory pattern representations.

We design the framework at the 30-second segment level (Supplementary Fig. 35), trained on 3,779,943 multi-channel segments from SHHS1 (2,631,479 normal events and 1,148,464 respiratory events). To ensure fair comparison, the framework utilizes identical TSD-Net architectures as encoders (without

classification heads) for Flow, Chest, and SpO₂ signals, each producing 8-dimensional features projected to 64-dimensional embeddings through two fully connected layers with layer normalization and ReLU activation. The demographic encoder processes 4-dimensional features (gender, age, BMI, and race) through a deeper network of four fully connected layers (4→32→64→128→64) with layer normalization, ReLU activation, and dropout.

We implement three CLIP-based learning strategies: pairwise signal alignment between each pair of physiological signals, leave-one-out alignment comparing each signal against the mean representation of the other two²⁰, and signal-demographic alignment between each signal and demographic features. All features are normalized to unit length before computing scaled cosine similarities with a learnable temperature parameter. The model is trained using AdamW optimizer (learning rate 1e-4) with composite loss functions weighted differently for various alignment tasks and cosine learning rate scheduling with warmup. Following CLIP's paradigm, the encoders are pre-trained through contrastive learning with only the classification head fine-tuned for downstream tasks.

The t-SNE visualization (Supplementary Fig. 35) reveals effective cross-channel alignment between physiological signals while maintaining a natural separation between physiological and demographic feature clusters. This separation reflects the inherent complementarity between dynamic respiratory patterns and static population-level risk factors. Through contrastive learning objectives, our framework facilitates meaningful interaction between these distinct feature modalities during representation learning, while preserving their respective characteristics essential for SA detection. Further performance evaluation (Supplementary Fig. 36) demonstrates the effectiveness of this design. While our fully-supervised TSD-Net framework demonstrates superior performance with sufficient labeled data (Flow: AUC 0.7853, Chest: 0.8147, SpO₂: 0.7982, Gold standard: 0.8202), the CLIP-inspired framework shows stable performance across varying amounts of labeled data (Flow: 0.6046-0.7535, Chest: 0.7065-0.7654, SpO₂: 0.6647-0.7645, Gold standard: 0.7731-0.7751), maintaining reasonable performance even with limited fine-tuning (1%-10%). These results establish the CLIP-inspired framework as a valuable complementary approach, particularly in scenarios with limited labeled data availability.

[19] Radford, A., et al. Learning transferable visual models from natural language supervision. in *International conference on machine learning* 8748-8763 (PmLR, 2021).

[20] Thapa, R., et al. Sleepfm: Multi-modal representation learning for sleep across ecg, eeg and respiratory signals. in *AAAI 2024 Spring Symposium on Clinical Foundation Models* (2024).

9. Some of the data should be double-checked. In Fig 3(c), for example, the error bar of BMI in CFS exceeds 80, which is not realistic. Please check the numbers again.

Response: We greatly appreciate your meticulous review and attention to this specific data point. We had also noticed this extreme value and conducted thorough verification.

After double-checking the official NSRR CFS dataset annotations (specifically the 'cfs-visit5-harmonized-dataset-0.7.0.csv' file) and referencing previously published work utilizing this database (e.g., Levy et al., *Deep learning for obstructive sleep apnea diagnosis based on single channel oximetry*, *Nature Communications*, 2023⁸), we can confirm that this extreme BMI value, while unusual, is indeed accurate and present in the original database.

For transparency, we present the relevant evidence:

	A	B	C	D	E	F	G	H	I
1	nsrrid	rectype	nsrr_age	nsrr_age_g	nsrr_sex	nsrr_race	nsrr_ethnic	nsrr_bmi	nsrr_bp_sy:nsrr_
2	800002	5	38.75	no	female	black or af	not hispan	43.65528	126.44
3	800010	5	21.24	no	female	white	not hispan	35.46235	115.33
4	800011	5	54.75	no	female	white	not hispan	76.99895	130.89
5	800017	5	49.36	no	female	black or af	not hispan	42.4993	110.89
6	800021	5	47.61	no	male	black or af	not hispan	36.7072	137.78
7	800024	5	22.5	no	male	black or af	not hispan	48.65717	135.78
8	800027	5	57.26	no	male	black or af	not hispan	31.69182	138.44
9	800032	5	60.66	no	female	black or af	not hispan	54.88751	133.78
10	800035	5	22.87	no	female	black or af	not hispan	84.79621	105.67
11	800037	5	41.15	no	male	white	not hispan	29.34517	124.22
12	800038	5	37.71	no	male	black or af	not hispan	25.06627	118.67
13	800040	5	42.36	no	female	black or af	not hispan	35.11644	120.22
14	800045	5	65.25	no	female	white	not hispan	26.43846	171.56
15	800047	5	59.7	no	male	white	not hispan	30.52329	137.56
16	800048	5	52.74	no	female	black or af	not hispan	32.90256	132.22

Figure 2 | Screenshot from NSRR CFS dataset annotation file (cfs-visit5-harmonized-dataset-0.7.0.csv), highlighting the presence of the extreme BMI value in question within the official database.

Figure 3 | Evidence from previously published research (Levy et al., Nature Communications, 2023⁸) showing the incorporation of this BMI value in their analysis of the CFS cohort, demonstrating consistent documentation of this extreme case across multiple studies.

We sincerely appreciate your careful observation and suggestion, which highlights the importance of data verification. This extreme value, though unusual, represents a genuine case in the CFS cohort.

- [8] Levy, J., Álvarez, D., Del Campo, F. & Behar, J.A. Deep learning for obstructive sleep apnea diagnosis based on single channel oximetry. *Nature Communications* **14**, 4881 (2023).
-

Overall, the selection of the model and SpO2 as a single modality should be important to clarify the position of the model in the broad field of AI, especially in healthcare. In addition to PSG measurements, additional deep learning methods such as CNN/RNN with multiple modalities should be considered. Approaches that pay more attention to clinical evidence, scalability, and user-centric evaluation may improve the utility and relevance of the work in practice.

Response: We deeply appreciate the reviewer's insightful suggestions on positioning our work in AI healthcare, deep learning architectures, and clinical implementation. These comments have helped us better articulate several key aspects of our study. As detailed in our responses above, our work has already incorporated many of these suggested elements: our network architecture integrates CNN components, and we specifically chose Transformer structures over RNN due to their superior performance in temporal modeling. Regarding clinical evidence, we have provided comprehensive quantitative comparisons between XAI and expert annotations, supported by detailed visualizations. The scalability concern is addressed through our CLIP-inspired framework for label-scarce scenarios, while user-centric evaluation is thoroughly examined through interactive trust metrics. These aspects have been clarified and emphasized throughout our revision.

For detailed responses to each specific point, please refer to our point-by-point answers above.

Reference

1. Priban, I. An analysis of some short-term patterns of breathing in man at rest. *The Journal of physiology* **166**, 425 (1963).
2. Chon, K.H., Dash, S. & Ju, K. Estimation of respiratory rate from photoplethysmogram data using time-frequency spectral estimation. *IEEE Transactions on Biomedical Engineering* **56**, 2054-2063 (2009).
3. Liu, Z., et al. Swin transformer: Hierarchical vision transformer using shifted windows. in *Proceedings of the IEEE/CVF international conference on computer vision* 10012-10022 (2021).
4. Lin, W., Wu, Z., Chen, J., Huang, J. & Jin, L. Scale-aware modulation meet transformer. in *Proceedings of the IEEE/CVF International Conference on Computer Vision* 6015-6026 (2023).
5. Wang, T., Lu, C., Shen, G. & Hong, F. Sleep apnea detection from a single-lead ECG signal with automatic feature-extraction through a modified LeNet-5 convolutional neural network. *Peerj* **7**(2019).
6. H. Rapoport David M. Smith Philip L. Kiley James P., S.H.H.R.G.R.S.s.p.c.e.S.M.H.L.B.K.Q.S.F.I.C.G.D.J.B.W. Methods for obtaining and analyzing unattended polysomnography data for a multicenter study. *Sleep* **21**, 759-767 (1998).
7. Berry, R.B., et al. Rules for Scoring Respiratory Events in Sleep: Update of the 2007 AASM Manual for the Scoring of Sleep and Associated Events. *Journal of Clinical Sleep Medicine* **8**, 597-619 (2012).
8. Levy, J., Álvarez, D., Del Campo, F. & Behar, J.A. Deep learning for obstructive sleep apnea diagnosis based on single channel oximetry. *Nature Communications* **14**, 4881 (2023).
9. Salari, N., et al. Detection of sleep apnea using Machine learning algorithms based on ECG Signals: A

- comprehensive systematic review. *Expert Systems with Applications* **187**, 115950 (2022).
10. Bahrami, M. & Forouzanfar, M. Sleep apnea detection from single-lead ECG: A comprehensive analysis of machine learning and deep learning algorithms. *IEEE Transactions on Instrumentation and Measurement* **71**, 1-11 (2022).
 11. Penzel, T., Moody, G.B., Mark, R.G., Goldberger, A.L. & Peter, J.H. The apnea-ECG database. in *Computers in Cardiology 2000. Vol. 27 (Cat. 00CH37163)* 255-258 (IEEE, 2000).
 12. Wang, S., *et al.* Machine learning assisted wearable wireless device for sleep apnea syndrome diagnosis. *Biosensors* **13**, 483 (2023).
 13. Retamales, G., *et al.* Towards automatic home-based sleep apnea estimation using deep learning. *npj Digital Medicine* **7**, 144 (2024).
 14. Hu, S., *et al.* IPCT-Net: Parallel information bottleneck modality fusion network for obstructive sleep apnea diagnosis. *Neural Networks* **181**, 106836 (2025).
 15. Berry, R.B., *et al.* Rules for scoring respiratory events in sleep: update of the 2007 AASM manual for the scoring of sleep and associated events: deliberations of the sleep apnea definitions task force of the American Academy of Sleep Medicine. *Journal of clinical sleep medicine* **8**, 597-619 (2012).
 16. Ke, G., *et al.* Lightgbm: A highly efficient gradient boosting decision tree. *Advances in neural information processing systems* **30**(2017).
 17. Selvaraju, R.R., *et al.* Grad-cam: Visual explanations from deep networks via gradient-based localization. in *Proceedings of the IEEE international conference on computer vision* 618-626 (2017).
 18. Hu, S., Wang, Y.n., Liu, J. & Yang, C. Personalized Transfer Learning for Single-Lead ECG-Based Sleep Apnea Detection: Exploring the Label Mapping Length and Transfer Strategy Using Hybrid Transformer Model. *Ieee Transactions on Instrumentation and Measurement* **72**(2023).
 19. Radford, A., *et al.* Learning transferable visual models from natural language supervision. in *International conference on machine learning* 8748-8763 (PmLR, 2021).
 20. Thapa, R., *et al.* Sleepfm: Multi-modal representation learning for sleep across ecg, eeg and respiratory signals. in *AAAI 2024 Spring Symposium on Clinical Foundation Models* (2024).

Transparent Artificial Intelligence-enabled Interpretable and Interactive Sleep Apnea Assessment across Flexible Monitoring Scenarios

Shuaicong Hu, Jian Liu, Yanan Wang, Cong Fu, Jichu Zhu, Huan Yu, and Cuiwei Yang

Dear Editor and Reviewers,

We deeply appreciate your thorough evaluation of our manuscript "*Transparent Artificial Intelligence-enabled Interpretable and Interactive Sleep Apnea Assessment across Flexible Monitoring Scenarios*" (NCOMMS-24-73977-A). Your insightful and constructive feedback has been invaluable in enhancing the quality and clarity of our work. In response, we have meticulously revised the manuscript to address all concerns, with detailed updates and additional analyses presented in a **56-page Supplementary Materials file** and corresponding revisions in the main text.

The major revisions include:

1) Performance Reporting and Comprehensive Analysis

- Provided results of the Non-overlap SHHS Model across all databases to offer a more comprehensive analysis of results.
- Avoided potential bias in SHHS2 performance reporting by using the Non-overlap SHHS Model exclusively.

2) Sampling Rate Validation

- Conducted a rigorous evaluation of model performance across different sampling rates (0.95 Hz, 1 Hz, 4 Hz, and 10 Hz) to assess the impact on both respiratory and SpO₂ signals.

3) Supplementary File Reorganization

- Reorganized and expanded the Supplementary Materials to 56 pages, enhancing logical flow and transparency to support reproducibility for future researchers.

We believe these revisions have significantly strengthened the scientific contribution and transparency of our manuscript. We remain at your disposal for any further clarifications or additional revisions needed.

Sincerely,

First author Shuaicong Hu & Corresponding author Cuiwei Yang

RESPONSE TO THE REVIEW OF NCOMMS-24-73977-A

The changes have been marked in red in the revised manuscript. Reviewer's comments are shown in bold, with our responses in italics immediately below each. Revised content in the text is indicated in blue in our responses.

Responses to Reviewer #1:

Thank you so much for the effort in the response. Despite the clear improvement in the manuscript, I still have some major comments.

***Response:** We sincerely thank the reviewer for acknowledging the improvements in our manuscript while noting some remaining major concerns. We greatly value this feedback and have carefully addressed each point raised. Below are our detailed responses to each of the reviewer's additional comments.*

1. Please, include the results in the databases different from SHHS of the model trained with non-overlap subjects of SHHS1.

***Response:** Thank you for this valuable suggestion, which has allowed us to enhance the comprehensiveness of our study. We have now included descriptions for both Supplementary Fig. 2 and Supplementary Table 3, which present the performance of the Non-overlap SHHS Model on external databases (MESA, MROS, and CFS). These results have also been incorporated into the Results section of the main text.*

Supplementary Fig. 2 illustrates the performance of the Non-overlap SHHS Model in AHI regression and four-level SA severity classification across the MESA, MROS, and CFS datasets using SpO₂ and Gold configurations. The scatter plots show strong correlations between predicted and true AHI values, with R² values consistently ranging from 0.93 to 0.96 across cohorts. The Bland-Altman plots highlight minimal mean differences and narrow limits of agreement, confirming the model's reliability in AHI prediction. The confusion matrices demonstrate accurate classification across all four severity categories, with most errors occurring between adjacent categories, a pattern consistent with the continuous nature of AHI measurements.

Supplementary Table 3 provides detailed performance metrics, including sensitivity (SEN), specificity (SPE), Macro F1 scores, and mean absolute error (MAE) for different channel configurations (Flow, Chest, SpO₂, and Gold). For instance, in the MESA dataset, the macro F1 scores for Flow, Chest, SpO₂, and Gold configurations are 0.394, 0.484, 0.756, and 0.745, respectively. In the MROS dataset, the macro F1 scores are 0.550, 0.623, 0.760, and 0.800, while in the CFS dataset, they are 0.505, 0.532, 0.764, and 0.778.

These results demonstrate that the Non-overlap SHHS Model exhibits robust performance across external datasets, achieving consistently high R² values for AHI regression and reliable classification metrics under both SpO₂-only and Gold configurations. We sincerely appreciate your suggestion, which has helped us provide a more complete and detailed description of these supplementary materials. Additionally, we have incorporated these results into the Results section 'Flexible Channel Configuration Performance in AHI Regression and SA Severity Classification' (Pages 5-8) of the main text and updated the supplementary materials accordingly.

In Flexible Channel Configuration Performance in AHI Regression and SA Severity Classification (Pages 5-8):

The Non-overlap SHHS model is trained on a subset of SHHS1 subjects (n=2,789) without overlap with SHHS2 subjects, and tested on the remaining SHHS1 records and all SHHS2 records. In contrast, the All-Sub SHHS model is trained on all SHHS1 subjects (n=5,255) and evaluated on external datasets (MESA, MROS, and CFS).

We further evaluate the Non-overlap SHHS model on MESA, MROS, and CFS datasets (Supplementary Fig. 2 and Supplementary Table 3), where the number of training subjects decreases significantly compared to the All-Sub SHHS model. Results (Supplementary Fig. 3) show no significant differences between the two models across most cohorts, including SHHS1, SHHS2, MESA, and MROS ($P > 0.05$). For instance, in the MESA dataset, the Gold configuration achieves macro F1 scores of 0.7406 (Non-overlap SHHS model) versus 0.7351 (All-Sub SHHS model, $P = 0.3053$). However, in the CFS dataset, the p-value is 0.0348 (Macro F1: 0.7798 versus 0.7860), indicating slight statistical significance but not a particularly strong difference. These results suggest that our method maintains strong performance despite reduced training subjects in most cases, showcasing its robustness and adaptability.

In Supplementary Material:

Supplementary Fig. 2 | Performance of the Non-overlap SHHS model (trained on n=2,789 subjects) on AHI regression and four-level SA severity classification across test cohorts using SpO₂ and Gold channels. The figure illustrates the model's performance across seven test cohorts for SpO₂ (SHHS1, SHHS2, MESA, MROS, CFS, FDU-HSH (R), and FDU-HSH (P)) and four test cohorts for Gold (SHHS1, SHHS2, MESA, and MROS). For each cohort and channel, the results are presented as scatter plots, Bland-Altman plots, and confusion matrices.

Supplementary Table 3 | Performance metrics of the Non-overlap SHHS model (trained on n=2,789 subjects) for AHI regression and four-level SA severity classification across test cohorts using various channel configurations (Flow, Chest, SpO₂, and Gold). Metrics include sensitivity (SEN), specificity (SPE), positive predictive value (PPV), F1 score, and mean

absolute error (MAE, reported as mean \pm standard deviation).

Dataset	Patient distribution	Flow			Chest			SpO ₂			Gold		
		SPE	F1	MAE	SPE	F1	MAE	SPE	F1	MAE	SPE	F1	MAE
SHHS1 (n=2,466)	Healthy (n=463)	0.971	0.426	3.83 \pm 2.82	0.956	0.479	4.22 \pm 4.31	0.971	0.757	1.81 \pm 1.58	0.961	0.757	1.67 \pm 1.71
	Mild (n=963)	0.667	0.647	3.78 \pm 3.35	0.683	0.652	3.84 \pm 3.29	0.837	0.787	2.43 \pm 2.40	0.846	0.771	2.67 \pm 2.60
	Moderate (n=665)	0.829	0.613	5.54 \pm 4.41	0.830	0.598	5.74 \pm 4.57	0.909	0.757	3.73 \pm 3.53	0.900	0.746	4.05 \pm 3.67
	Severe (n=375)	0.967	0.741	10.17 \pm 8.53	0.971	0.728	10.64 \pm 8.73	0.970	0.864	5.96 \pm 5.00	0.969	0.854	6.32 \pm 5.27
	Overall (n=2,466)	ACC=0.621 F1 _{Macro} =0.607		5.23 \pm 5.21	ACC=0.622 F1 _{Macro} =0.614		5.46 \pm 5.51	ACC=0.785 F1 _{Macro} =0.791		3.20 \pm 3.42	ACC=0.775 F1 _{Macro} =0.782		3.41 \pm 3.64
SHHS2 (n=2,522)	Healthy (n=437)	0.959	0.428	4.03 \pm 3.03	0.948	0.492	4.04 \pm 3.84	0.966	0.734	2.02 \pm 2.00	0.950	0.793	1.30 \pm 1.54
	Mild (n=941)	0.673	0.642	3.69 \pm 3.23	0.715	0.625	4.05 \pm 3.38	0.848	0.768	2.66 \pm 3.07	0.886	0.798	2.25 \pm 2.39
	Moderate (n=687)	0.829	0.575	5.94 \pm 4.55	0.802	0.544	6.17 \pm 4.89	0.904	0.749	3.48 \pm 3.02	0.923	0.786	2.97 \pm 2.50
	Severe (n=457)	0.969	0.741	11.46 \pm 9.62	0.952	0.696	12.04 \pm 10.83	0.965	0.879	5.46 \pm 5.29	0.975	0.890	5.75 \pm 5.63
	Overall (n=2,522)	ACC=0.611 F1 _{Macro} =0.596		5.77 \pm 5.99	ACC=0.595 F1 _{Macro} =0.589		6.07 \pm 6.58	ACC=0.778 F1 _{Macro} =0.783		3.28 \pm 3.61	ACC=0.810 F1 _{Macro} =0.817		2.92 \pm 3.47
MESA (n=1,846)	Healthy (n=187)	0.999	0.091	10.10 \pm 6.08	0.861	0.460	2.49 \pm 2.67	0.949	0.692	1.58 \pm 1.24	0.932	0.700	1.05 \pm 0.94
	Mild (n=571)	0.849	0.373	10.62 \pm 8.06	0.658	0.510	4.21 \pm 3.17	0.848	0.739	2.68 \pm 2.41	0.847	0.715	2.67 \pm 2.17
	Moderate (n=559)	0.664	0.404	10.41 \pm 7.85	0.794	0.395	9.14 \pm 5.73	0.903	0.714	4.42 \pm 3.59	0.893	0.694	4.40 \pm 3.41
	Severe (n=529)	0.758	0.708	11.57 \pm 10.13	0.983	0.569	19.86 \pm 14.66	0.976	0.877	6.73 \pm 6.87	0.985	0.872	7.49 \pm 7.88
	Overall (n=1,846)	ACC=0.488 F1 _{Macro} =0.394		10.78 \pm 8.50	ACC=0.483 F1 _{Macro} =0.484		10.01 \pm 10.94	ACC=0.764 F1 _{Macro} =0.756		4.26 \pm 4.76	ACC=0.748 F1 _{Macro} =0.745		4.41 \pm 5.26
MROS (n=3,522)	Healthy (n=388)	0.988	0.290	5.37 \pm 4.49	0.954	0.518	3.52 \pm 3.59	0.983	0.634	2.28 \pm 1.69	0.975	0.741	1.58 \pm 1.47
	Mild (n=1,168)	0.763	0.592	4.92 \pm 4.24	0.801	0.624	4.35 \pm 3.97	0.870	0.760	2.63 \pm 2.37	0.897	0.792	2.31 \pm 2.00
	Moderate (n=1,103)	0.752	0.562	6.44 \pm 5.06	0.800	0.579	6.54 \pm 5.40	0.886	0.759	3.63 \pm 3.22	0.902	0.778	3.43 \pm 2.89
	Severe (n=863)	0.920	0.754	10.34 \pm 8.91	0.927	0.769	9.33 \pm 8.61	0.947	0.887	5.70 \pm 5.11	0.955	0.892	5.59 \pm 4.77
	Overall (n=3,522)	ACC=0.601 F1 _{Macro} =0.550		6.77 \pm 6.34	ACC=0.634 F1 _{Macro} =0.623		6.17 \pm 6.19	ACC=0.780 F1 _{Macro} =0.760		3.66 \pm 3.66	ACC=0.807 F1 _{Macro} =0.800		3.38 \pm 3.42
CFS (n=576)	Healthy (n=232)	0.980	0.311	5.33 \pm 3.25	0.954	0.518	3.15 \pm 3.22	0.962	0.842	1.84 \pm 1.16	0.919	0.867	1.03 \pm 0.96
	Mild (n=161)	0.465	0.510	4.22 \pm 4.52	0.801	0.624	4.15 \pm 3.59	0.843	0.663	2.95 \pm 3.26	0.870	0.709	2.31 \pm 2.43
	Moderate (n=101)	0.891	0.507	7.75 \pm 6.55	0.800	0.579	9.48 \pm 6.69	0.914	0.691	4.89 \pm 7.36	0.941	0.697	4.90 \pm 6.45
	Severe (n=82)	0.968	0.693	17.94 \pm 11.95	0.927	0.769	21.96 \pm 14.95	0.966	0.862	9.61 \pm 9.49	0.976	0.840	10.28 \pm 8.94
	Overall (n=576)	ACC=0.484 F1 _{Macro} =0.505		7.24 \pm 7.63	ACC=0.552 F1 _{Macro} =0.532		7.22 \pm 9.41	ACC=0.764 F1 _{Macro} =0.764		3.79 \pm 5.71	ACC=0.788 F1 _{Macro} =0.778		3.39 \pm 5.52
FDU-HSH (R) (n=327)	Healthy (n=50)	-	-	-	-	-	-	0.910	0.760	1.63 \pm 0.79	-	-	-
	Mild (n=116)	-	-	-	-	-	-	0.915	0.771	2.41 \pm 2.37	-	-	-
	Moderate (n=68)	-	-	-	-	-	-	0.931	0.677	3.89 \pm 3.14	-	-	-
	Severe (n=93)	-	-	-	-	-	-	0.957	0.886	9.11 \pm 8.04	-	-	-

	Overall (n=327)	-	-	-	-	-	-	ACC=0.783 F1 _{Macro} =0.774	3.28±3.62	-	-	-	
FDU-HSH (P) (n=265)	Healthy (n=46)	-	-	-	-	-	-	0.895	0.800	1.60±0.57	-	-	-
	Mild (n=75)	-	-	-	-	-	-	0.911	0.731	2.61±2.14	-	-	-
	Moderate (n=68)	-	-	-	-	-	-	0.954	0.768	4.84±3.95	-	-	-
	Severe (n=76)	-	-	-	-	-	-	0.989	0.924	12.91±9.95	-	-	-
	Overall (n=265)	-	-	-	-	-	-	ACC=0.808 F1 _{Macro} =0.806	5.96±7.37	-	-	-	

2. Regarding the previous point, as expected, SHHS2 results significantly decrease when using the model trained with non-overlapped SHHS1 subjects. Accordingly, the SHHS2 main results to be reported should be those from this model, and not those from the current model. At least, those currently reported results for SHHS2 are not valid because they are clearly biased.

Response: Thank you for your rigorous and valuable suggestion. We fully agree that the reported results for SHHS2 should prioritize the use of the Non-overlap SHHS Model to avoid potential bias. To address this concern, we conducted a comprehensive comparison between the two models: the Non-overlap SHHS Model, which is trained on a subset of SHHS1 subjects (n=2,789) without overlap with SHHS2 subjects, and the All-Sub SHHS Model, which is trained on the entirety of SHHS1 subjects (n=5,255).

We carefully analyzed the performance differences between these two models on SHHS2 and other external datasets, and we assessed the statistical significance of these differences (Supplementary Fig. 3). To ensure unbiased evaluation, we have now reported the results for SHHS1 and SHHS2 based solely on the Non-overlap SHHS Model. This approach eliminates any potential bias caused by overlapping subjects across the training and testing datasets.

At the same time, to make full use of the available training data and provide a comprehensive evaluation of the system's performance, we have retained the results of the All-Sub SHHS Model for external datasets, including MESA, MROS, and CFS. This ensures that the analysis of these external databases benefits from the largest possible training dataset.

The manuscript has been revised accordingly to reflect these updates (Fig. 4 on Page 5, Fig. 5 on Page 6, Results section 'Overview of Experimental Procedures' on Page 4, Results section 'Flexible Channel Configuration Performance in AHI Regression and SA Severity Classification' on Pages 5-8, Results section 'Cross-Population Stability Assessment of AIX' on Page 8, Supplementary Figs. 3-7,25, Supplementary Tables 2,3,6-9, and other instances where reported numerical values have been updated). Specifically, the reported SHHS1 and SHHS2 results now come exclusively from the Non-overlap SHHS Model to avoid bias, while the results for MESA, MROS, and CFS remain based on the All-Sub SHHS Model to maximize the use of training data. These changes have been implemented throughout the main text and supplementary materials to ensure clarity and to align with the highest standards of rigor and transparency.

In Overview of Experimental Procedures (Page 4):

To ensure unbiased evaluation and assess temporal stability, we employ two distinct training strategies for the SHHS dataset: (1) Non-overlap SHHS model (training subjects: n=2,789, Supplementary Fig. 1),

which excludes overlapping subjects between SHHS1 and SHHS2 for training and evaluation, ensuring no data leakage across cohorts. (2) All-Sub SHHS model (training subjects: $n=5,255$, Fig. 2), trained on all available SHHS1 subjects, maximizing the use of training data for performance optimization. The Non-overlap SHHS model is primarily used for SHHS2 evaluation to avoid bias and can also be applied to assess temporal performance between SHHS1 and SHHS2. Meanwhile, the All-Sub SHHS model serves as a reference for external test cohorts. Detailed results from both models are reported to provide a comprehensive view of AIX performance across different cohorts and channel configurations.

In Flexible Channel Configuration Performance in AHI Regression and SA Severity Classification (Pages 5-8):

The Non-overlap SHHS model is trained on a subset of SHHS1 subjects ($n=2,789$) without overlap with SHHS2 subjects, and tested on the remaining SHHS1 records and all SHHS2 records. In contrast, the All-Sub SHHS model is trained on all SHHS1 subjects ($n=5,255$) and evaluated on external datasets (MESA, MROS, and CFS).

Based on the single-channel SpO₂ configuration, MROS exhibits low sensitivity in the 'Healthy' category (Supplementary Fig. 7), consistent with literature 17, possibly because its 'Healthy' subjects display characteristics similar to abnormal breathing events. To address this, the Gold configuration incorporates additional respiratory channels, providing complementary information to improve sensitivity in such challenging scenarios. However, the Gold configuration underperforms the single-channel SpO₂ configuration in MESA, which can be attributed to differences in breathing patterns across racial groups, leading to distributional shifts. Additionally, the respiratory features captured by the additional channels may not generalize well across diverse groups, further reducing its effectiveness in MESA. For AHI prediction, reported under the Gold configuration, SHHS2 achieves the lowest MAE at 2.92, explaining its superior classification performance, while MESA shows the highest MAE at 4.76 due to racial variability and the aforementioned challenges with the Gold configuration. In comparison, MROS and CFS also benefit from the Gold configuration, achieving MAEs of 3.23 and 3.44, respectively, reflecting its improved performance over the single-channel SpO₂ configuration in these cohorts.

We further evaluate the Non-overlap SHHS model on MESA, MROS, and CFS datasets (Supplementary Fig. 2 and Supplementary Table 3), where the number of training subjects decreases significantly compared to the All-Sub SHHS model. Results (Supplementary Fig. 3) show no significant differences between the two models across most cohorts, including SHHS1, SHHS2, MESA, and MROS ($P>0.05$). For instance, in the MESA dataset, the Gold configuration achieves macro F1 scores of 0.7406 (Non-overlap SHHS model) versus 0.7351 (All-Sub SHHS model, $P=0.3053$). However, in the CFS dataset, the p -value is 0.0348 (Macro F1: 0.7798 versus 0.7860), indicating slight statistical significance but not a particularly strong difference. These results suggest that our method maintains strong performance despite reduced training subjects in most cases, showcasing its robustness and adaptability.

In Cross-Population Stability Assessment of AIX (Page 8):

We conduct comprehensive stability analysis across demographic and ethnic groups. For demographic subgroups, we evaluate two representative configurations (Gold and SpO₂) across gender, age, and BMI categories using data from SHHS2, MESA, MROS, and CFS (Fig. 4d). The Gold configuration maintains stable performance across different groups. For gender, it achieves a macro F1 score of 0.81 for males

($n=5,760$) and 0.77 for females ($n=2,706$). Across age groups, performance remains consistent, with scores of 0.83 for young adults (age 18-40, $n=191$), 0.81 for middle-aged adults (age 40-60, $n=1,162$), and 0.79 for elderly populations (age >60 , $n=7,113$). Similarly, for BMI categories, the configuration demonstrates reliability, achieving scores of 0.79 for normal-weight individuals (BMI 0-24.9, $n=1,701$), 0.81 for overweight individuals (BMI 24.9-29.9, $n=2,965$), and 0.82 for obese individuals (BMI >29.9 , $n=1,930$). Minor variations are observed in females, elderly subjects, and those with lower BMI indices, potentially due to physiological differences. Importantly, similar patterns of stability are observed under the SpO₂ configuration, reinforcing these findings (Supplementary Fig. 25).

In Figure 4 (Page 5):

Fig. 4. External test cohorts (SHHS1, SHHS2, MESA, MROS, CFS) are evaluated for performance, with the model trained on SHHS1 data. a, the confusion matrix shows the classification into four severity levels. **b**, the scatter plot illustrates the correlation between manual AHI scoring and predicted AHI. The R-squared (R^2) value is calculated, along with the diagonal representing the linear regression model, the 95% CI, and the two-sided p-value. **c**, the Bland-Altman plot displays the comparison of manual and predicted AHI, with error lines positioned at ± 1.96 times the standard deviation. The p-value is computed using the Wilcoxon signed-rank test, and the intraclass correlation coefficient (ICC) is provided, along with the mean of the mean absolute error (MAE) of predicted AHI for all patients. Green, yellow, pink, and purple scatter points represent healthy individuals, mild SA, moderate SA, and severe SA patients, respectively. **d**, Performance of AIX under Gold channel configuration across different gender, age, and BMI groups. Macro F1 denotes the average F1 score. S-0, S-1, S-2, and S-3 represent four classification scenarios: four-level SA classification, healthy versus SA, non-moderate versus moderate SA, and

non-severe versus severe SA. *Note: Results for SHHS1 and SHHS2 are obtained using the Non-overlap SHHS model, which excludes overlapping subjects between the two cohorts to ensure unbiased evaluation and temporal performance assessment. Results for MESA, MROS, and CFS are based on the All-Sub SHHS model, which utilizes all available SHHS1 subjects to maximize training data.

In Figure 5 (Page 6):

Fig. 5. Model interpretability and interaction feasibility. *a*, The t-SNE dimensionality reduction plots show the representation of the original data and AI model-extracted features under four channel types (Flow, Chest, SpO₂, and Gold). *b,c*, Model attention visualization. These illustrate the important regions at different scales learned from nasal airflow (Flow) and blood oxygen saturation (SpO₂) signals, providing transparent decision logic. *d*, t-SNE visualization of samples detected by the model. *e*, Analysis of prediction confidence scores (TN, FN+FP, TP), showing significantly lower confidence in incorrect predictions (two-tailed Mann-Whitney U test, $P=7.07e-17$, $P=2.07e-19$). *f*, MAE trends across SA severity levels for SHHS1, SHHS2, MESA, MROS, and CFS under different channel configurations, with error bars showing one standard error.

In Supplementary Material:

Supplementary Table 2 | Performance of the All-Sub SHHS model (trained on $n=5,255$ subjects) is evaluated across five

external test sets using various channel configurations (Flow, Chest, SpO₂, and Gold). For each channel, the mean absolute error (MAE, reported as mean \pm standard error) is provided for both four-level overall predictions and specific sleep apnea (SA) severity groups (Healthy: AHI <5, Mild: $5 \leq$ AHI <15, Moderate: $15 \leq$ AHI <30, Severe: AHI \geq 30). Furthermore, overall accuracy (ACC), specificity (SPE), and the F1 score for each severity class, along with the average F1 score (Macro F1), are presented to assess classification performance. The "Gold" configuration represents the integration of Flow, Chest, and SpO₂ channels.

Dataset	Patient distribution	Flow			Chest			SpO ₂			Gold		
		SPE	F1	MAE	SPE	F1	MAE	SPE	F1	MAE	SPE	F1	MAE
MESA (n=1,846)	Healthy (n=187)	0.998	0.109	10.44 \pm 6.44	0.866	0.457	2.64 \pm 2.79	0.947	0.694	1.52 \pm 1.22	0.920	0.703	1.11 \pm 0.95
	Mild (n=571)	0.861	0.351	11.13 \pm 8.20	0.667	0.513	4.29 \pm 3.27	0.837	0.736	2.65 \pm 2.29	0.849	0.703	2.85 \pm 2.24
	Moderate (n=559)	0.660	0.398	10.72 \pm 7.95	0.786	0.388	8.95 \pm 5.75	0.888	0.698	4.60 \pm 3.55	0.890	0.681	4.57 \pm 3.52
	Severe (n=529)	0.743	0.704	11.37 \pm 10.00	0.976	0.580	19.33 \pm 14.56	0.985	0.854	7.65 \pm 7.34	0.986	0.860	8.31 \pm 8.24
	Overall (n=1,846)	ACC=0.482 F1 _{Macro} =0.390		11.00 \pm 8.53	ACC=0.484 F1 _{Macro} =0.485		9.84 \pm 10.75	ACC=0.751 F1 _{Macro} =0.746		4.56 \pm 5.08	ACC=0.738 F1 _{Macro} =0.737		4.76 \pm 5.56
MROS (n=3,522)	Healthy (n=388)	0.985	0.317	5.47 \pm 4.75	0.957	0.507	3.69 \pm 3.71	0.980	0.648	2.18 \pm 1.64	0.963	0.752	1.38 \pm 1.30
	Mild (n=1,168)	0.779	0.577	5.27 \pm 4.50	0.808	0.611	4.58 \pm 4.22	0.859	0.772	2.40 \pm 2.18	0.898	0.786	2.23 \pm 1.93
	Moderate (n=1,103)	0.744	0.552	6.55 \pm 5.14	0.793	0.565	6.75 \pm 5.59	0.885	0.767	3.30 \pm 2.89	0.905	0.776	3.32 \pm 2.74
	Severe (n=863)	0.908	0.749	10.07 \pm 8.79	0.913	0.764	9.37 \pm 8.52	0.968	0.883	5.28 \pm 4.87	0.962	0.891	5.28 \pm 4.73
	Overall (n=3,522)	ACC=0.593 F1 _{Macro} =0.549		6.87 \pm 6.32	ACC=0.625 F1 _{Macro} =0.612		6.34 \pm 6.25	ACC=0.785 F1 _{Macro} =0.768		3.36 \pm 3.41	ACC=0.805 F1 _{Macro} =0.801		3.23 \pm 3.32
CFS (n=576)	Healthy (n=232)	0.965	0.368	5.37 \pm 3.52	0.849	0.667	3.29 \pm 3.42	0.956	0.846	1.75 \pm 1.14	0.907	0.884	0.98 \pm 0.91
	Mild (n=161)	0.492	0.500	4.44 \pm 4.71	0.648	0.484	4.25 \pm 3.87	0.848	0.682	2.71 \pm 2.99	0.887	0.724	2.38 \pm 2.42
	Moderate (n=101)	0.893	0.510	7.77 \pm 6.54	0.895	0.385	9.51 \pm 6.83	0.918	0.710	4.47 \pm 6.95	0.949	0.705	4.97 \pm 6.06
	Severe (n=82)	0.964	0.693	17.43 \pm 11.55	0.976	0.597	21.46 \pm 14.78	0.974	0.862	9.02 \pm 8.90	0.980	0.850	10.57 \pm 8.29
	Overall (n=576)	ACC=0.493 F1 _{Macro} =0.518		7.25 \pm 7.47	ACC=0.549 F1 _{Macro} =0.533		7.24 \pm 9.28	ACC=0.774 F1 _{Macro} =0.775		3.53 \pm 5.33	ACC=0.804 F1 _{Macro} =0.791		3.44 \pm 5.34
FDU-HSH (Retrospective) (n=327)	Healthy (n=50)	-	-	-	-	-	-	0.895	0.746	1.40 \pm 0.74	-	-	-
	Mild (n=116)	-	-	-	-	-	-	0.910	0.750	2.51 \pm 2.37	-	-	-
	Moderate (n=68)	-	-	-	-	-	-	0.927	0.711	3.76 \pm 3.17	-	-	-
	Severe (n=93)	-	-	-	-	-	-	0.983	0.904	8.36 \pm 7.57	-	-	-
	Overall (n=327)	-	-	-	-	-	-	ACC = 0.783 F1 _{Macro} =0.778		4.27 \pm 5.24	-	-	-
FDU-HSH (Prospective) (n=265)	Healthy (n=46)	-	-	-	-	-	-	0.895	0.800	1.47 \pm 0.53	-	-	-
	Mild (n=75)	-	-	-	-	-	-	0.874	0.697	2.86 \pm 2.15	-	-	-
	Moderate (n=68)	-	-	-	-	-	-	0.959	0.733	5.14 \pm 4.09	-	-	-
	Severe (n=76)	-	-	-	-	-	-	0.995	0.923	12.0 \pm 10.1	-	-	-
	Overall (n=265)	-	-	-	-	-	-	ACC = 0.789 F1 _{Macro} =0.788		5.83 \pm 7.16	-	-	-

Supplementary Fig. 3 | Comparison of model performance between Non-overlap SHHS and All-Sub SHHS models using SpO_2 and Gold channels across test cohorts. (a) Macro F1 scores for the SpO_2 channel across six test cohorts (SHHS2, MESA, MROS, CFS, FDU-HSH (R), and FDU-HSH (P)). Performance is compared between the Non-overlap SHHS model (n=2,789) and the All-Sub SHHS model (n=5,255) with statistical significance assessed using a two-sided t-test. (b) Macro F1 scores for the Gold channel across four test cohorts (SHHS2, MESA, MROS, and CFS). Statistical significance is evaluated similarly. (c) Comparison of Macro F1 scores for SpO_2 and Gold channels on SHHS1, highlighting the differences in performance between the two channels using the Non-overlap SHHS model. For all panels, the mean and error bars representing one standard deviation are shown, with statistical significance indicated by p-values. No significant differences are observed in most cohorts, except for CFS (p=0.0348 in panel b).

Channel	Metric (Mean, 95%CI)	SHHS1 (n=2,466)	SHHS2 (n=2,522)	MESA (n=2,522)	MROS (n=3,522)	CFS (n=576)	FDU-HSH (R) (n=327)	FDU-HSH (P) (n=265)
Flow	ACC	0.615 (0.605,0.624)	0.613 (0.610,0.615)	0.494 (0.486,0.502)	0.600 (0.594,0.606)	0.487 (0.481,0.493)	-	-
	F1 _{Macro}	0.603 (0.592,0.613)	0.600 (0.595,0.604)	0.399 (0.390,0.407)	0.549 (0.543,0.555)	0.510 (0.504,0.516)	-	-
Chest	ACC	0.621 (0.617,0.624)	0.587 (0.583,0.591)	0.473 (0.465,0.481)	0.632 (0.628,0.636)	0.551 (0.548,0.555)	-	-
	F1 _{Macro}	0.613 (0.606,0.621)	0.582 (0.574,0.591)	0.478 (0.467,0.489)	0.621 (0.615,0.626)	0.530 (0.526,0.535)	-	-
SpO ₂	ACC	0.779 (0.775,0.782)	0.780 (0.777,0.783)	0.756 (0.753,0.759)	0.785 (0.782,0.787)	0.766 (0.761,0.771)	0.783 (0.773,0.793)	0.801 (0.790,0.812)
	F1 _{Macro}	0.785 (0.782,0.789)	0.786 (0.783,0.788)	0.749 (0.746,0.752)	0.766 (0.762,0.771)	0.764 (0.759,0.770)	0.775 (0.765,0.784)	0.800 (0.789,0.812)
Gold	ACC	0.773 (0.770,0.777)	0.803 (0.800,0.806)	0.735 (0.729,0.742)	0.806 (0.805,0.807)	0.798 (0.792,0.804)	-	-
	F1 _{Macro}	0.780 (0.777,0.783)	0.810 (0.807,0.814)	0.735 (0.729,0.741)	0.801 (0.800,0.802)	0.786 (0.783,0.790)	-	-

Supplementary Fig. 4 | Performance comparison of different channel configurations for four-level SA severity classification across test cohorts (with 95% CI). The left panel shows classification accuracy and the right panel shows macro F1 scores across different cohorts. SpO₂ and Gold configurations consistently demonstrate superior performance in both classification accuracy (Mean accuracy: SpO₂ 0.756-0.785, Gold 0.735-0.806) and macro F1 scores (Mean macro F1 scores: SpO₂ 0.749-0.786, Gold 0.735-0.810) compared to Flow and Chest configurations. To avoid bias, results for SHHS1 (n=2,466) and SHHS2 (n=2,522) are reported based on the Non-overlap SHHS model, which excludes these cohorts during training. In contrast, results for MESA (n=1,846), MROS (n=3,522), CFS (n=576), FDU-HSH retrospective (R) cohort (n=327), and FDU-HSH prospective (P) cohort (n=265) are reported based on the All-Sub SHHS model, which fully utilizes the available training data. While SpO₂ and Gold configurations exhibit strong and consistent performance across most cohorts, performance shows a slight decline on MESA. Nevertheless, single-channel SpO₂ monitoring achieves comparable performance to the Gold standard configuration, suggesting its potential for a simplified home-based setting.

Supplementary Fig. 5 | Performance evaluation of the AIX system under the single-channel (Flow) workflow is conducted on four external test sets (a. SHHS2, b. MESA, c. MROS, d. CFS), with model training data sourced from SHHS1. Each test set presents the SA four-level classification confusion matrix, correlation scatter plots for AHI regression, and Bland-Altman plots. The R-squared values are provided along with the diagonal line representing the linear regression model and the two-sided t-test p-values corresponding to the 95% confidence intervals. The Bland-Altman plots show the comparison between manually measured AHI and predicted AHI, with error lines positioned at ± 1.96 standard deviations. p-values are calculated using the Wilcoxon signed-rank test, and intraclass correlation coefficients (ICCs) are provided, as well as the mean absolute error (MAE) of predicted AHI for all patients. Green, yellow, pink, and purple scatter points represent healthy individuals, mild SA, moderate SA, and severe SA patients, respectively.

Supplementary Fig. 6 | Performance evaluation of the AIX system under the single-channel (Chest) workflow is conducted on four external test sets (a. SHHS2, b. MESA, c. MROS, d. CFS), with model training data sourced from SHHS1. Each test set presents the SA four-level classification confusion matrix, correlation scatter plots for AHI regression, and Bland-Altman plots. The R-squared values are provided along with the diagonal line representing the linear regression model and the two-sided t-test p-values corresponding to the 95% confidence intervals. The Bland-Altman plots show the comparison between manually measured AHI and predicted AHI, with error lines positioned at ± 1.96 standard deviations. p-values are calculated using the Wilcoxon signed-rank test, and intraclass correlation coefficients (ICCs) are provided, as well as the mean absolute error (MAE) of predicted AHI for all patients. Green, yellow, pink, and purple scatter points represent healthy individuals, mild SA, moderate SA, and severe SA patients, respectively.

Supplementary Fig. 7 | Performance evaluation of the AIX system under the single-channel (SpO₂) workflow is conducted on four external test sets (a. SHHS2, b. MESA, c. MROS, d. CFS), with model training data sourced from SHHS1. Each test set presents the SA four-level classification confusion matrix, correlation scatter plots for AHI regression, and Bland-Altman plots. The R-squared values are provided along with the diagonal line representing the linear regression model and the two-sided t-test p-values corresponding to the 95% confidence intervals. The Bland-Altman plots show the comparison between manually measured AHI and predicted AHI, with error lines positioned at ± 1.96 standard deviations. p-values are calculated using the Wilcoxon signed-rank test, and intraclass correlation coefficients (ICCs) are provided, as well as the mean absolute error (MAE) of predicted AHI for all patients. Green, yellow, pink, and purple scatter points represent healthy individuals, mild SA, moderate SA, and severe SA patients, respectively.

Supplementary Table 6 | The AIX system's performance is evaluated in both professional (Gold) and home-based (SpO₂) settings, with four-level SA classification results analyzed across different ethnicities in the SHHS2, MESA, and CFS

databases. The evaluation metrics include overall accuracy, Macro F1 score, R-squared value, and ICC coefficient.

Ethnicity (Healthy/Mild/Moderate/Severe)		SpO ₂				Gold			
		ACC	F1 _{macro}	R ²	ICC	ACC	F1 _{macro}	R ²	ICC
SHHS2 (n=2,522)	Black or African American, n=171 (36/65/37/33)	0.760	0.759	0.951	0.949	0.801	0.805	0.962	0.961
	White, n=2,194 (362/820/605/407)	0.783	0.785	0.955	0.954	0.813	0.817	0.959	0.959
	Other, n=157 (39/56/45/17)	0.739	0.754	0.934	0.932	0.790	0.800	0.943	0.941
	Overall (437/941/687/457)	0.778	0.783	0.954	0.952	0.810	0.817	0.959	0.958
MESA (n=1,846)	Asian, n=223 (22/66/67/68)	0.731	0.727	0.950	0.949	0.722	0.719	0.949	0.948
	Black or African American, n=491 (60/161/142/128)	0.760	0.758	0.930	0.927	0.727	0.730	0.909	0.905
	White, n=696 (68/227/204/197)	0.741	0.736	0.916	0.912	0.740	0.739	0.905	0.900
	Hispanic, n=436 (37/117/146/136)	0.766	0.755	0.935	0.932	0.755	0.753	0.925	0.922
	Overall (187/571/559/529)	0.751	0.746	0.928	0.926	0.738	0.737	0.916	0.913
CFS (n=576)	Black or African American, n=319 (125/92/50/52)	0.781	0.779	0.948	0.946	0.803	0.781	0.946	0.944
	White, n=245 (100/65/51/29)	0.759	0.760	0.912	0.908	0.796	0.790	0.907	0.903
	Multiple, n=12 (7/4/0/1)	0.917	0.937	0.985	0.985	1.000	1.000	0.981	0.981
	Overall (232/161/101/82)	0.774	0.775	0.933	0.931	0.804	0.791	0.930	0.928

Supplementary Table 7 | With the AHI cutoff set at 5, the AIX system reports binary classification results for healthy individuals and SA patients in both professional (Gold) and home-based (SpO₂) settings. Performance metrics on six external test sets are presented. SEN, PPV, and SPE represent sensitivity, positive predictive value, and specificity for the positive class, respectively. ACC represents overall accuracy, and F1_{macro} denotes the average F1 score.

Dataset	SpO ₂					Gold				
	ACC	F1 _{macro}	SEN	PPV	SPE	ACC	F1 _{macro}	SEN	PPV	SPE
SHHS2 (n=2,522)	0.916	0.842	0.966	0.934	0.673	0.926	0.874	0.950	0.961	0.815
MESA (n=1,846)	0.930	0.827	0.947	0.975	0.781	0.921	0.829	0.920	0.991	0.925
MROS (n=3,522)	0.933	0.805	0.980	0.947	0.557	0.943	0.860	0.963	0.973	0.781
CFS (n=576)	0.885	0.877	0.956	0.866	0.780	0.905	0.901	0.907	0.931	0.901
FDU-HSH (R) (n=327)	0.902	0.843	0.895	0.988	0.940	-	-	-	-	-
FDU-HSH (P) (n=265)	0.913	0.872	0.895	1.000	1.000	-	-	-	-	-

Supplementary Table 8 | With the AHI cutoff set at 15, the AIX system reports binary classification results for healthy individuals and SA patients in both professional (Gold) and home-based (SpO₂) settings. Performance metrics on six external test sets are presented. SEN, PPV, and SPE represent sensitivity, positive predictive value, and specificity for the positive class, respectively. ACC represents overall accuracy, and F1_{macro} denotes the average F1 score.

Dataset	SpO ₂					Gold				
	ACC	F1 _{macro}	SEN	PPV	SPE	ACC	F1 _{macro}	SEN	PPV	SPE
SHHS2 (n=2,522)	0.906	0.905	0.913	0.883	0.900	0.923	0.922	0.911	0.918	0.933
MESA (n=1,846)	0.891	0.889	0.844	0.966	0.958	0.884	0.882	0.831	0.967	0.959
MROS (n=3,522)	0.907	0.906	0.917	0.917	0.895	0.914	0.913	0.919	0.927	0.908
CFS (n=576)	0.927	0.918	0.934	0.851	0.924	0.939	0.929	0.869	0.935	0.972
FDU-HSH (R) (n=327)	0.933	0.933	0.901	0.960	0.964	-	-	-	-	-

FDU-HSH (P) (n=265)	0.891	0.891	0.812	0.983	0.983	-	-	-	-	-
-------	-------	-------	-------	-------	---	---	---	---	---

Supplementary Table 9 | With the AHI cutoff set at 30, the AIX system reports binary classification results for healthy individuals and SA patients in both professional (Gold) and home-based (SpO₂) settings. Performance metrics on six external test sets are presented. SEN, PPV, and SPE represent sensitivity, positive predictive value, and specificity for the positive class, respectively. ACC represents overall accuracy, and F1macro denotes the average F1 score.

Dataset	SpO ₂					Gold				
	ACC	F1 _{macro}	SEN	PPV	SPE	ACC	F1 _{macro}	SEN	PPV	SPE
SHHS2 (n=2,522)	0.955	0.926	0.910	0.851	0.965	0.960	0.933	0.891	0.889	0.975
MESA (n=1,846)	0.924	0.901	0.773	0.953	0.985	0.927	0.906	0.781	0.958	0.986
MROS (n=3,522)	0.943	0.923	0.867	0.899	0.968	0.946	0.928	0.898	0.885	0.962
CFS (n=576)	0.960	0.919	0.878	0.847	0.974	0.958	0.913	0.829	0.872	0.980
FDU-HSH (R) (n=327)	0.948	0.934	0.860	0.952	0.983	-	-	-	-	-
FDU-HSH (P) (n=265)	0.958	0.947	0.868	0.985	0.995	-	-	-	-	-

Supplementary Fig. 25 | Performance of AIX under single-channel SpO₂ configuration across different gender, age, and BMI groups. Macro F1 denotes the average F1 score. S-0, S-1, S-2, and S-3 represent four classification scenarios: four-level SA classification, healthy versus SA, non-moderate versus moderate SA, and non-severe versus severe SA.

3. Using 1 Hz for respiratory signals (airflow, chest, etc) is far below the 10 Hz established as a minimum for the American Academy of Sleep Medicine. There exist deep learning approaches that do not need the use of the same sampling frequency in different signals (eg, stacking different CNN branches). Accordingly, it is my opinion that to prove you are not losing significant information you should compare your current results with this approach, or a similar one using appropriate sampling frequencies of at least 1 Hz for SpO₂ and 4 Hz (some papers did it in the past) for respiratory signals.

Response: Thank you for raising this important concern regarding the sampling rates of respiratory signals and their potential impact on performance. We fully recognize the relevance of using higher sampling rates, as established by the American Academy of Sleep Medicine (AASM), and appreciate your suggestion to explore deep learning methods that can accommodate varying resolutions. To address this, we conducted additional experiments comparing the performance of our model across multiple sampling rates for different signal types (Flow, Chest, SpO₂, and Gold), as detailed in the revised manuscript and Supplementary Figs. 9,10.

Specifically, we evaluated macro F1 scores across four sampling rates—0.95 Hz, 1 Hz, 4 Hz, and 10 Hz—selected based on clinical standards and prior literature. To handle the varying resolutions, we modified the TSD-Net by incorporating a stack of convolutional layers in the Stem module to normalize input resolutions to a fixed length of 64. This design ensures effective feature extraction while accommodating higher-resolution inputs (e.g., 10 Hz) and lower-resolution inputs (e.g., 0.95 Hz or 1 Hz).

The results show that while higher sampling rates slightly improve performance for isolated respiratory signals (e.g., Flow and Chest), the Gold configuration, which integrates Flow, Chest, and SpO₂ signals, exhibits consistent and robust performance across all sampling rates. For instance, in SHHS2, the Gold configuration achieves a macro F1 score of 0.810 (95% CI: 0.807–0.814) at (Flow: 0.95 Hz, Chest: 0.95 Hz, SpO₂: 0.95 Hz) and 0.809 (95% CI: 0.806–0.812) at (Flow: 10 Hz, Chest: 10 Hz, SpO₂: 1 Hz). These results suggest that the inclusion of SpO₂ signals compensates for information loss caused by downsampling respiratory signals. For SpO₂ signals, the performance differences between 0.95 Hz and 1 Hz are negligible, reflecting their stable and low-variability nature.

Additionally, we observed that higher sampling rates can benefit certain datasets and configurations. For example, in MESA, the Flow channel achieves a macro F1 score of 0.429 at 10 Hz compared to 0.399 at 0.95 Hz. However, the Gold configuration maintains robust performance across all rates, achieving 0.741 (95% CI: 0.731–0.751) at (Flow: 10 Hz, Chest: 10 Hz, SpO₂: 1 Hz) versus 0.735 (95% CI: 0.728–0.742) at (Flow: 0.95 Hz, Chest: 0.95 Hz, SpO₂: 0.95 Hz).

These findings demonstrate that while higher sampling rates provide marginal benefits for individual respiratory signals, the Gold configuration effectively mitigates the impact of downsampling by leveraging complementary information from SpO₂. This underscores the robustness and practicality of our proposed method, particularly in resource-constrained environments such as wearable devices and home-based monitoring systems, where lower sampling rates can significantly reduce computational demands without compromising performance.

We have revised the manuscript accordingly in the Methods section ‘Performance Comparison across Sampling Rates’ on Page 17 and the supplementary materials (Supplementary Figs. 9,10). To further enhance the robustness of our study, we have also incorporated three high-quality references related to sampling rate adoption:

- [1] Wang, E., Koprinska, I. & Jeffries, B. Sleep apnea prediction using deep learning. *IEEE Journal of Biomedical and Health Informatics* 27, 5644–5654 (2023). (Sampling rate: Respiratory signals originally at 32 Hz; experimented with downsampling to 16 Hz, 8 Hz, 4 Hz, 2 Hz, 1 Hz, and 0.5 Hz)
- [2] Levy J., Álvarez D., Rosenberg A.A., Alexandrovich A., del Campo F. & Behar J.A. Digital oximetry biomarkers for assessing respiratory function: standards of measurement, physiological interpretation, and clinical use. *NPJ digital medicine* 4, 1 (2021). (Sampling rate: Oxygen saturation signals sampled at 1 Hz)
- [3] Nakano, H., Tanigawa, T., Furukawa, T. & Nishima, S. Automatic detection of sleep-disordered breathing from a single-channel airflow record. *European Respiratory Journal* 29, 728–736 (2007). (Sampling rate: Airflow and respiratory movement signals sampled at 10 Hz; pulse oximetry data at 1 Hz)

In Performance Comparison across Sampling Rates (Page 17):

To assess the impact of sampling rates on model performance, macro F1 scores are compared across signal types (Flow, Chest, SpO₂, and Gold) and four sampling rates: 0.95 Hz, 1 Hz, 4 Hz, and 10 Hz, selected based on clinical standards and prior literature¹⁻³. The lowest rate (0.95 Hz) corresponds to performance calculated for 256-length signal segments (270-second windows), while 1 Hz reflects the

relatively stable nature of SpO₂ signals compared to respiratory signals. To handle varying input resolutions, the TSD-Net's Stem layer is replaced with a stack of convolutional layers that reduce inputs to a fixed length of 64. High-resolution inputs (e.g., 10 Hz) require more layers, while low-resolution inputs (e.g., 0.95 Hz, 1 Hz) require fewer. These layers are configured with appropriate kernel sizes, strides, and paddings, ensuring effective feature extraction and dimensionality reduction.

Signals are downsampled for each configuration, and macro F1 scores are evaluated across test datasets (SHHS2, MESA, MROS, CFS, FDU-HSH). Error bars represent 95% CIs to capture performance variability. Results indicate stable performance across most sampling rates, with minor variations (Supplementary Figs. 9,10). For instance, in the Flow channel, SHHS1 achieves a macro F1 score of 0.603 (95% CI: 0.591-0.615) at 0.95 Hz and 0.630 (95% CI: 0.624-0.636) at 10 Hz, showing slight improvements at higher rates. However, the Gold configuration, which integrates Flow, Chest, and SpO₂ signals, exhibits consistent performance across rates. For example, in SHHS2, Gold achieves 0.810 (95% CI: 0.807-0.814) at (Flow: 0.95 Hz, Chest: 0.95 Hz, SpO₂: 0.95 Hz) and 0.809 (95% CI: 0.806-0.812) at (Flow: 10 Hz, Chest: 10 Hz, SpO₂: 1 Hz), suggesting that SpO₂ compensates for information loss caused by downsampling respiratory signals. For SpO₂ signals, performance differences between 0.95 Hz and 1 Hz are negligible. In SHHS2, macro F1 scores are 0.786 (95% CI: 0.783-0.789) at 0.95 Hz and 0.787 (95% CI: 0.781-0.793) at 1 Hz, reflecting the stability and low variability of SpO₂ signals. Notably, in datasets like MESA, the Flow channel shows a performance decline at lower sampling rates (e.g., 0.429 at 10 Hz versus 0.399 at 0.95 Hz). However, the Gold configuration maintains robust performance, achieving 0.741 (95% CI: 0.731-0.751) at (Flow: 10 Hz, Chest: 10 Hz, SpO₂: 1 Hz) compared to 0.735 (95% CI: 0.728-0.742) at (Flow: 0.95 Hz, Chest: 0.95 Hz, SpO₂: 0.95 Hz).

These findings demonstrate that while higher sampling rates slightly improve performance for isolated respiratory signals (Flow, Chest), the Gold configuration effectively mitigates the impact of downsampling by integrating SpO₂ signals. This underscores the robustness of the proposed method and its potential for practical applications in resource-constrained environments, such as wearable devices and home-based monitoring systems, where lower sampling rates reduce computational demands without compromising performance.

In Supplementary Material:

Channel	F1 _{Macro} (Mean, 95%CI)	SHHS1 (n=2,466)	SHHS2 (n=2,522)	MESA (n=2,522)	MROS (n=3,522)	CFS (n=576)	FDU-HSH (R) (n=327)	FDU-HSH (P) (n=265)
Flow	256/270 Hz	0.603 (0.591,0.615)	0.600 (0.595,0.605)	0.399 (0.389,0.408)	0.549 (0.543,0.556)	0.510 (0.503,0.517)	-	-
	4 Hz	0.630 (0.623,0.637)	0.605 (0.599,0.611)	0.435 (0.421,0.449)	0.582 (0.569,0.595)	0.541 (0.532,0.550)	-	-
	10 Hz	0.630 (0.624,0.636)	0.602 (0.599,0.605)	0.429 (0.420,0.438)	0.577 (0.569,0.585)	0.537 (0.524,0.550)	-	-
Chest	256/270 Hz	0.613 (0.605,0.622)	0.582 (0.573,0.591)	0.478 (0.465,0.490)	0.621 (0.614,0.627)	0.530 (0.526,0.535)	-	-
	4 Hz	0.615 (0.612,0.618)	0.598 (0.593,0.603)	0.468 (0.458,0.478)	0.628 (0.622,0.634)	0.526 (0.521,0.531)	-	-
	10 Hz	0.616 (0.611,0.621)	0.594 (0.591,0.597)	0.476 (0.469,0.483)	0.622 (0.620,0.624)	0.531 (0.524,0.538)	-	-
SpO ₂	256/270 Hz	0.785 (0.781,0.789)	0.786 (0.783,0.789)	0.749 (0.746,0.752)	0.766 (0.761,0.772)	0.764 (0.758,0.770)	0.775 (0.764,0.786)	0.800 (0.787,0.813)
	1 Hz	0.786 (0.782,0.790)	0.787 (0.781,0.793)	0.742 (0.733,0.751)	0.768 (0.756,0.780)	0.766 (0.761,0.771)	0.779 (0.772,0.786)	0.799 (0.788,0.810)
Gold	256/270, 256/270, 256/270 Hz	0.780 (0.777,0.783)	0.810 (0.807,0.814)	0.735 (0.728,0.742)	0.801 (0.800,0.802)	0.786 (0.782,0.790)	-	-
	4,4,1 Hz	0.778 (0.775,0.781)	0.807 (0.801,0.813)	0.725 (0.720,0.730)	0.796 (0.787,0.805)	0.778 (0.772,0.784)	-	-
	10,10,1 Hz	0.780 (0.776,0.784)	0.809 (0.806,0.812)	0.741 (0.731,0.751)	0.792 (0.782,0.802)	0.782 (0.780,0.784)	-	-

Supplementary Fig. 9 | Performance evaluation of macro F1 scores across different sampling rates and channel configurations. The figure includes results from four channels: Flow, Chest, SpO₂, and Gold. For each channel, macro F1 scores were evaluated under multiple sampling rates (256/270 Hz, 4 Hz, 10 Hz for Flow and Chest; 256/270 Hz and 1 Hz for SpO₂; and combined configurations for Gold). The solid lines represent the mean macro F1 scores, and the shaded regions indicate the 95% confidence intervals (CIs). The table below the plot provides the detailed mean macro F1 scores and 95% CIs for each channel, sampling rate, and dataset. Results are shown for seven datasets: SHHS1, SHHS2, MESA, MROS, CFS, FDU-HSH (R), and FDU-HSH (P), with corresponding sample sizes.

Supplementary Fig. 10 | Statistical evaluation of macro F1 scores across different sampling rates and channel configurations. The bar plots present the mean macro F1 scores for each dataset (SHHS1, SHHS2, MESA, MROS, CFS, FDU-HSH (R), and FDU-HSH (P)) under various sampling and channel configurations. The error bars represent the standard deviation obtained from five repeated experiments. The p-values, calculated using two-tailed paired t-tests, are displayed above the bars to assess statistical differences between configurations. Comparisons are made within each dataset for the same channel under different sampling rates.

Responses to Reviewer #3:

This revision and the answers provide important additions to the initial manuscript. The subject of addressing interactive sleep apnea assessment is of high importance in view of the high prevalence of the disorder.

To introduce transparent and interpretable AI is critical in view of the current discussion on using AI. The supplemental information is definitely needed in order to understand the details for a science reader.

Taken together, the work will be of high significance to the field of sleep medicine, sleep research, and consumer health. Flaws in the data analysis or interpretation can not be identified.

The supplemental material is rich and needed in order to allow the methods to be reproduced.

Response: Thank you very much for your thoughtful and encouraging feedback on our revised manuscript. We are truly heartened by your recognition of the importance of addressing interactive sleep apnea assessment and the critical role of transparent and interpretable AI in this field. Your acknowledgment of the significance of our work to sleep medicine, sleep research, and consumer health inspires us to continue pursuing high standards in both methodological transparency and impactful research.

In this revision, we carefully addressed the comments from Reviewer #1, including adding new analyses and reorganizing the supplemental figures and tables to further clarify the logical flow. These updates aim to make the methods and results more accessible and reproducible for future researchers.

Moreover, to support reproducibility and facilitate further advancements in the field, we have made the source code available on GitHub at the following link: <https://github.com/fdu-harry/Apnea-Interact-Explainer>. The repository includes detailed instructions on reproducing our results, as well as a packaged executable file for the AIX system, which enables users to directly run the software. We plan to maintain and update this repository to extend the impact of our work and provide a reliable resource for subsequent researchers to replicate and improve upon.

Thank you once again for your kind words and constructive review. We greatly appreciate your time and effort in evaluating our work and for helping us further refine and strengthen our manuscript.

Reference

1. Wang, E., Koprinska, I. & Jeffries, B. Sleep apnea prediction using deep learning. *IEEE Journal of Biomedical and Health Informatics* **27**, 5644-5654 (2023).
2. Levy, J., *et al.* Digital oximetry biomarkers for assessing respiratory function: standards of measurement, physiological interpretation, and clinical use. *NPJ digital medicine* **4**, 1 (2021).
3. Nakano, H., Tanigawa, T., Furukawa, T. & Nishima, S. Automatic detection of sleep-disordered breathing from a single-channel airflow record. *European Respiratory Journal* **29**, 728-736 (2007).